# ChemBOMAS: Accelerated BO for Scientific Discovery in Chemistry with LLM-Enhanced Multi-Agent System

## Abstract

Bayesian optimization (BO) is a powerful tool for scientific discovery in chemistry, yet its efficiency is often hampered by the sparse experimental data and vast search space. Here, we introduce **ChemBOMAS**: a large language model (LLM)-enhanced multi-agent system that accelerates BO through synergistic data- and knowledge-driven strategies. Firstly, the data-driven strategy involves an 8B-scale LLM regressor fine-tuned on a mere 1% labeled samples for pseudo-data generation, robustly initializing the optimization process. Secondly, the knowledge-driven strategy employs a hybrid Retrieval-Augmented Generation approach to guide LLM in dividing the search space while mitigating LLM hallucinations. An Upper Confidence Bound algorithm then identifies high-potential subspaces within this established partition. Across the LLM-refined subspaces and supported by LLM-generated data, BO achieves the improvement of effectiveness and efficiency. Comprehensive evaluations across multiple chemical benchmarks demonstrate that ChemBOMAS set a new state-of-the-art, accelerating optimization efficiency by up to 5-fold compared to baseline methods. Additionally, a real wet-lab campaign with strong early-round gains validated the practical relevance of ChemBOMAS.

## 1 Introduction

Manual experimentation and traditional control variable methods have long underpinned chemical discovery, yet they remain labor-intensive and time-consuming, slowing the generation of new scientific insights Xie et al. (2023); Tom et al. (2024). To address these constraints, automated or self-driving laboratories integrate robotic execution with AI algorithms, delivering high throughput, precision, and efficiency Seifrid et al. (2022); Chen et al. (2024); Ai et al. (2024a). Within these experimental platforms, Bayesian Optimization (BO) algorithms are widely recognized as a crucial decision-making tool for experiment design Guo et al. (2023); Abolhasani and Kumacheva (2023); Chen et al. (2023); Ai et al. (2024b). BO enables efficient navigation of complex experimental variable spaces and converges toward optimal reaction conditions or material compositions by integrating prior data, constructing probabilistic surrogate models, quantifying uncertainty, and iteratively selecting the most informative subsequent experiments Shields et al. (2021a).

Despite BO achieving remarkable success in complex scientific domains, particularly chemistry, it still contends with two major obstacles: (I) the scarcity and high cost of experimental observations during the early optimization stages, and (II) the multitude of reaction parameters that inflate the search into high-dimensional design spaces Shahriari et al. (2015); Wang et al. (2023). The two obstacles exacerbate the limitations of vanilla BO, also known as the "cold start" problem and the "curse of dimensionality", frequently leading to slow convergence Guo et al. (2023). Without effective acceleration strategies, the protracted optimization process may yield only marginal improvements, which could cause researchers to abandon the search before discovering the optimal conditions.

Several strategies have been proposed to accelerate BO, including search space partitioning Wang et al. (2020a), specialized encoding embeddings Tripp et al. (2020); Moriconi et al. (2020); Nayebi et al. (2019), pseudo-data generation Yin et al. (2023), and transfer across similar tasks Swersky et al. (2013). However, when these acceleration strategies are applied to the intricate chemical reactions, two critical shortcomings emerge. First, most approaches employ a single acceleration technique,

which might be insufficient for the chemical optimization problems with multiple demands, such as exploration of diverse reaction parameter combinations while overcoming data scarcity in the early-stage. Second, current acceleration methods are predominantly data-driven. Because chemical reaction pathways differ widely in their underlying kinetics and thermodynamics, a purely statistical BO framework frequently expends resources in chemically implausible regions of the search space, missing opportunities to leverage mechanistic insight that could guide the search more efficiently.

To overcome these limitations, we propose **ChemBOMAS**, an LLM-Enhanced **M**ulti-**A**gent **S**ystem specifically designed for accelerated **B**ayesian **O**ptimization in chemistry. ChemBOMAS synergistically integrates two LLM-powered modules: a **knowledge-driven search space decomposition module** and a **data-driven pseudo-data generation module**. The knowledge-driven module employs an LLM-powered agent to reason over existing chemical knowledge (e.g., literature, databases), intelligently decompose the vast search space and identify promising candidate regions, dynamically pruning the search space for better BO efficiency. Simultaneously, the data-driven module utilizes a fine-tuned LLM to generate informative pseudo-data points across the entire search space. These pseudo-data not only warm-start the BO process but also inform the knowledge-driven module's subspace selection. This closed-loop interaction enables ChemBOMAS to achieve superior optimization efficiency and convergence speed even under extreme data scarcity.

The effectiveness of ChemBOMAS was rigorously evaluated. We conducted extensive experiments on four chemical performance optimization benchmarks, demonstrating consistent improvements in optimal results, convergence speed, initialization performance, and robustness compared to various baseline methods. Crucially, ablation studies confirmed that the synergy between the knowledge-driven and data-driven strategies is essential for creating a highly efficient and robust optimization framework. Additionally, the practical utility and real-world applicability of ChemBOMAS were validated through a previously unreported wet-lab experiment.

Our main contributions are summarized as follows:

**1.** We systematically investigated how LLM-based approaches could address two key limitations in BO for scientific discovery in chemistry: data scarcity and inefficiency in vast search spaces.

**2.** We propose ChemBOMAS, a framework that synergistically leverages LLM-based knowledge and data modules to improve the sample efficiency of BO for chemical synthesis tasks. A knowledge-driven module partitions the variable space into chemically meaningful subspaces, while a data-driven module generates pseudo-data to identify the promising subspaces and warm-start the surrogate. Unlike prior approaches, these modules augment rather than replace the underlying BO components.

**3.** We show that ChemBOMAS consistently and substantially outperforms four relevant baselines when only 1% of the data is labeled across four chemical benchmarks with ten independent seeds.

**4.** We demonstrate the practical utility and scalability of ChemBOMAS through a real wet-lab campaign with a 10-combination variable space.

## 2 RELATED WORK

Large Language Models (LLMs) offer synergistic potential with Bayesian Optimization (BO) to address traditional BO limitations (e.g., sample inefficiency, cold starts) by providing prior knowledge Souza et al. (2021), enhancing surrogate models Liu et al. (2024); Nguyen et al. (2024); Ramos et al. (2023a), automating acquisition function design Austin et al. (2024), and enabling optimization in novel problem representations. Prior work has explored LLM-driven BO improvements in warm-starting, surrogate modeling, candidate generation, acquisition function design, and search space understanding.

However, directly replacing core BO modules with LLMs introduces significant challenges. LLM "hallucinations" can mislead optimization, compromising reliability. Furthermore, the direct suitability of LLMs as surrogates or acquisition functions is limited by concerns regarding uncertainty quantification, theoretical guarantees, computational cost, efficiency in low-data regimes, adaptability to specific numerical tasks, and interpretability.

On another front, some techniques such as LA-MCTS Wang et al. (2020a) was proposed, which employ tree structures to decompose the search space Wang et al. (2023; 2024; 2019). Some works

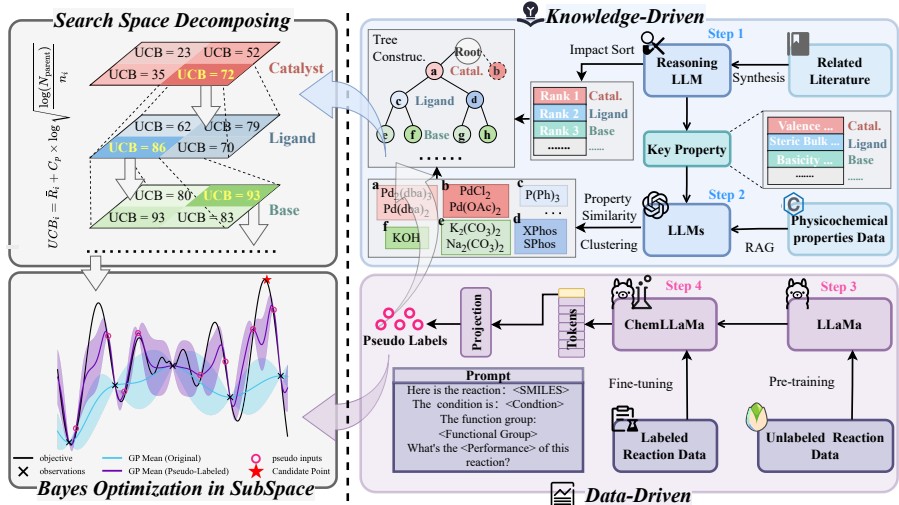

Figure 1: ChemBOMAS: A synergistic knowledge- and data-driven framework for efficient Bayesian Optimization. The framework operates as a closed-loop system: the **knowledge-driven module** decomposes the search space into subspaces using LLM-extracted chemical insights, followed by a UCB algorithm to select promising subspaces; the **data-driven module** generates pseudo-data to initialize both the subspace selection and the Bayesian Optimization process within the selected subspaces. The two modules interact iteratively, with real data from optimization feedback refining subsequent search directions.

propose hierarchical Bayesian optimization Moriconi et al. (2020); Reker et al. (2020). These approaches offer valuable strategies for managing and navigating complex optimization landscapes. Unlike previous works, we focus on robustly integrating LLM knowledge to enhance BO, leveraging their strengths as auxiliary tools while mitigating weaknesses such as hallucinations, to achieve this synergy over substitution.

## 3 METHODOLOGY

### 3.1 PROBLEM SETUP

This work aims to significantly improve the efficiency of searching a task's variable space for the optimal combination that maximizes the objective function.

### 3.2 THE FRAMEWORK OF CHEMBOMAS

As illustrated in Figure 1, we propose ChemBOMAS, an LLM-enhanced multi-agent optimization framework that systematically integrates data-driven and knowledge-driven strategies. First, the data-driven strategy utilizes a pre-trained and fine-tuned LLM regressor to generate pseudo-data, thereby robustly initializing the optimization process. Second, the knowledge-driven strategy employs a hybrid Retrieval-Augmented Generation (RAG) approach, which guides an LLM to partition the search space based on variable impact ranking and property similarity. Third, an Upper Confidence Bound (UCB) algorithm then identifies the most promising subspaces from this partition. Finally, BO is performed within the selected subspaces, supported by the LLM-generated pseudo-data, leading to enhanced effectiveness and efficiency. The complete algorithm process can be seen in Appendix D.The two strategies are detailed below.

### 3.3 DATA-DRIVEN STRATEGY: LLM-GENERATED PSEUDO DATA

An LLM-based regression model was constructed and utilized in three sequential steps to generate pseudo-data for optimization initialization.

**Step 1: Pre-training.** The base LLaMA 3.1 model Grattafiori et al. (2024) was pre-trained on the Pistachio dataset $\mathcal{D}_{\text{chem}}$ to enhance its representational ability for chemical reactions. The dataset was formatted as Q&A pairs where, given reactants $\mathbf{R}$ and products $\mathbf{P}$, the model learns to predict the corresponding reaction conditions $\mathbf{c} = (c_1, c_2, \ldots, c_T)$, thereby avoiding direct exposure to objective performance labels. Pre-training employed a Causal Language Modeling loss: $\mathcal{L}_{\text{pre-train}} = \mathbb{E}_{s \sim \mathcal{D}_{\text{chem}}} \left[ \sum t = 1^T \log p(w_t | s_{<t}) \right]$, where $t$ denotes the token index, $s = (w_1, ..., w_t)$ denotes a token sequence, $w_t$ represents the $t$-th token, and $T$ is the sequence length.

**Step 2: Fine-tuning.** The pre-trained model was subsequently fine-tuned on a small labeled dataset $\mathcal{D}_{\text{labeled}} = \{(\mathbf{x}_i, y_i)\}_{i=1}^N$, which only comprises 1% of the data, by integrating a regression head. A reaction configuration $\mathbf{x}$ (including $\mathbf{R}$, $\mathbf{P}$, and $\mathbf{c}$) is fed into the LLM via prompt engineering. The final hidden state $\mathbf{h}T = \text{LLM}(\mathbf{x})^{[T]}$ is then projected to a reaction performance prediction $\hat{y}$ via an MLP: $\hat{y} = f\theta_{\text{MLP}}(\mathbf{h}_T) = f_{\theta_{\text{MLP}}}(\text{LLM}(\mathbf{x})^{[T]})$. Fine-tuning used Low-Rank Adaptation (LoRA) Hu et al. (2022) with rank $r = 8$, introducing adaptable parameters $\phi_{\text{LoRA}}$ alongside the frozen pre-trained weights $\theta_{\text{LLM}}$. The MLP parameters $\theta_{\text{MLP}}$ were fully trained to minimize an L2-loss with regularization:

$$\mathcal{L}_{\text{fine-tune}} = \frac{1}{|\mathcal{D}_{\text{labeled}}|} \sum_{(\mathbf{x},y) \in \mathcal{D}_{\text{labeled}}} \left\| f_{\theta_{\text{MLP}}}(\text{LLM}_{\theta_{\text{LLM}}, \phi_{\text{LoRA}}}^{[T]}(\mathbf{x})) - y \right\|_2^2 + \lambda \|\theta_{\text{MLP}}\|_2^2 \tag{1}$$

**Step 3: Pseudo-data Generation and Utilization.** The fine-tuned LLM regressor was used to generate pseudo-data for all unsampled data points, forming a pseudo-dataset $\mathcal{D}_{\text{pseudo}} = \{(\mathbf{x}_k, \hat{y}_k)\}_{k=1}^M$, where $\hat{y}_k = f_{\theta_{\text{MLP}}}(\text{LLM}_{\theta_{\text{LLM}}, \phi_{\text{LoRA}}}^{[T]}(\mathbf{x}_k))$ was used to initialize a UCB algorithm. The UCB algorithm (Section 3.4) then identifies the high-potential subspaces. BO is then conducted within these subspaces (Section 3.5), leveraging both the selected pseudo-data and limited real data to accelerate surrogate model fitting. To mitigate the influence of the noise in the pseudo-data, a refinement strategy based on data similarity and reverse-order removal is applied (see Appendix C for details).

### 3.4 KNOWLEDGE-DRIVEN STRATEGY: LLM-DIVIDED SEARCH SPACE

To efficiently identify high-potential regions in the vast chemical reaction parameter space, we construct a Subspace Tree Search module using the GPT-o3 API in three steps.

**Step 1: LLM-Guided Space Partitioning.** The $n$-dimensional variable space is defined as $\mathcal{X} = \{C_i\}_{i=1}^n$, containing $n$ categories of chemicals involved in the reaction, where each $C_i = \{x_{i,1}, \ldots, x_{i,k}\}$ represented a category including $k$ candidate substances. A hybrid Retrieval-Augmented Generation (RAG) approach integrates multi-source information (literature, professional databases, web search) to facilitate the LLM's decisions and minimize hallucination. The LLM first ranks the chemical categories $C_i$ by their importance to the chemical reaction, generating an ordered sequence $\mathcal{O} = (o_1, \ldots, o_n)$. Subsequently, for each chemical category $C_i$, the LLM identifies key influencing physicochemical properties $p_{i,1}, p_{i,2}, \ldots$ and clusters the candidates based on these property values. This partitions each $C_i$ into a collection of seperate subspaces $\Pi_i = \{S_{i,1}, \ldots, S_{i,q_i}\}$, where candidate substances within each subspace $S_{i,l}$ share similar properties.

**Step 2: Hierarchical Search Tree Construction.** A hierarchical tree is built based on the category importance order $\mathcal{O}$ and the clustering results $\Pi_i$. The $l$-th layer of the tree corresponds to the $l$-th most important category $C_l$ and contains nodes representing its $q_l$ clusters. Each path from the root to a leaf node defines a unique search subspace as the Cartesian product of $n$ clusters, resulting in a total of $\prod_{i=1}^n q_i$ disjoint subspaces that comprehensively partition the original space.

**Step 3: UCB-based Subspace Selection.** A UCB algorithm is employed to explore the tree and identify promising subspaces. Starting from the root, UCB selects child nodes layer-by-layer until reaching a leaf node. The UCB value for a child node $i$ is computed as: $\text{UCB}_i = \bar{R}_i + C_p \times \log \sqrt{\frac{log(N_{parent})}{n_i}}$, where $\bar{R}_i$ is the average performance value (exploitation), $N_{parent}$ is the parent's visit count, $n_i$ is the child's visit count, and $C_p$ is an exploration constant. At each layer, the top-5

nodes by UCB value are selected for further exploration. This path traversal pinpoints high-value subspaces for subsequent BO. The UCB values are updated dynamically as BO progresses and new samples are acquired.

## 3.5 Bayesian Optimization in ChemBOMAS

BO is performed within the promising subspaces identified by the preceding modules, leveraging the LLM-generated pseudo-data for initialization. The procedure consists of two main steps.

**Step 1: Surrogate Modeling.** A Gaussian Process (GP) surrogate model, using a Matérn kernel with constant scaling and a white noise kernel, is fitted to the combined set of actual observations and pseudo-data points, which serve as an informative prior. This model provides, for any unsampled point $\mathbf{x}$ in the target subspaces, a posterior distribution over the performance value, characterized by a mean function $\mu(\mathbf{x})$ and a variance function $\sigma^2(\mathbf{x})$.

**Step 2: Acquisition Function Optimization.** An acquisition function $\alpha(\mathbf{x})$, such as Expected Improvement (EI), is used to recommend the next sample by balancing exploration (high uncertainty) and exploitation (high predicted mean). The next query point is selected by maximizing $\alpha(\mathbf{x})$ over the unsampled points within the high-potential subspaces: $\mathbf{x}_{next} = \arg\max \mathbf{x} \in \mathcal{X}_{\text{sub}} \alpha(\mathbf{x})$. This point is then evaluated to obtain a new real observation, which updates the GP surrogate model for the next iteration.

# 4 Experiment

## 4.1 Data

The pre-training phase employed a subset of the Pistachio dataset containing approximately 50,000 chemical reaction entries, none of which contained objective performance labels.

For the fine-tuning and Bayesian Optimization (BO) phases, three benchmark datasets were used: Suzuki Perera et al. (2018), Arylation Shields et al. (2021b), and Buchwald Ahneman et al. (2018). In each case, only 1% of the labeled data was randomly selected for fine-tuning the LLM regressor; the effectiveness of this data volume is analyzed in Appendix J.2. For the Buchwald dataset, which exhibits inconsistencies in reactants and products across entries, two consistent subsets were constructed, denoted Buchwald_sub1 and Buchwald_sub2, to serve as rational benchmarks for the optimization task. Detailed statistics for all benchmarks are provided in Appendix F.

## 4.2 Experiment Setup

The LLM regressor in data-driven module was trained on $2\times$ NVIDIA A800 GPUs. To facilitate parameter-efficient fine-tuning, we adopted the Low-Rank Adaptation (LoRA) technique, configuring the rank $r = 8$, the scaling factor $\alpha = 16$, and the LoRA dropout rate to $0.1$. For fine-tuning the LLM regressor, the hyperparameters were set as follows: learning rate of $1 \times 10^{-4}$, batch size of 24, and 100 training epochs.

In the knowledge-driven module, the search tree was constructed using a UCB policy with an exploration constant $\kappa = 1$. The BO process was run for 40 iterations, initialized with 1% of the data as the prior and sampling 0.1% of the dataset in each iteration. It employed a Single-task Gaussian Process as the surrogate model and utilized EI as the acquisition function. Each optimization experiment was independently repeated 10 times with different random seeds, and the average performance across these runs is reported as the final result. The prompts for LLM clustering and further implementation details are provided in the Appendix E. Additionally, the specific hyperparameter settings for all baseline methods are detailed in Appendix F.4.

## 4.3 Performance Comparsion

### 4.3.1 Regression Models

The quality of the pseudo-data is directly influenced by the prediction accuracy of the regression model. We evaluated the performance prediction accuracy of ChemBOMAS against four categories

of existing regression models on three chemical datasets: 1) traditional machine learning models fitted on 1% labeled data; 2) general-purpose LLMs (GPT series) with zero-shot inference; 3) open LLMs fine-tuned on 1% labeled data; and 4) scientific LLMs with molecule pre-training fine-tuned on 1% labeled data . The prediction metrics for each model are summarized in Table 1, from which two key observations can be drawn.

Table 1: Comparative performance of various LLM-based regression models on the chemical performance prediction task.

| Model | Suzuki | | | Arylation | | | Buchwald | | |
|---|---|---|---|---|---|---|---|---|---|
| | MSE↓ | MAE↓ | $R^2$ ↑ | MSE↓ | MAE↓ | $R^2$ ↑ | MSE↓ | MAE↓ | $R^2$ ↑ |
| *Traditional Machine Learning Models fitted on 1% labeled data* | | | | | | | | | |
| MLP | 737.88 | 23.93 | 0.04 | 596.30 | 22.62 | 0.03 | 612.57 | 22.51 | 0.06 |
| DecisionTree | 749.86 | 20.80 | 0.05 | 927.01 | 23.38 | -0.24 | 1003.35 | 25.19 | -0.35 |
| RandomForest | 693.08 | 22.25 | 0.12 | 735.65 | 22.78 | 0.01 | 761.55 | 23.50 | -0.02 |
| XGBoost | 643.03 | 21.89 | 0.18 | 653.86 | 21.25 | **0.13** | 667.22 | 21.63 | 0.10 |
| *General-purpose LLMs with zero-shot inference* | | | | | | | | | |
| GPT-4o | 2207.17 | 40.02 | -1.80 | 2702.58 | 44.86 | -2.63 | 1512.44 | 33.60 | -1.03 |
| GPT-5 | 1218.93 | 30.34 | -0.55 | 1515.81 | 33.68 | -1.04 | 1516.62 | 33.55 | -1.04 |
| *Open LLMs fine-tuned on 1% labeled data* | | | | | | | | | |
| Bert | 808.12 | 24.04 | -0.03 | 746.78 | 23.18 | -0.00 | 747.05 | 23.19 | -0.00 |
| Qwen3-7B | 820.48 | 22.10 | -0.04 | 848.51 | 22.52 | -0.14 | 998.22 | 25.25 | -0.34 |
| GLM4-9B | **593.49** | **18.94** | **0.25** | 739.20 | 20.78 | 0.01 | 719.72 | 20.77 | 0.00 |
| LLaMa-3.1-8B | 685.55 | 20.50 | 0.13 | 679.72 | 19.57 | 0.09 | 739.27 | 20.57 | 0.01 |
| *Scientific LLMs with molecule pre-training and fine-tuned on 1% labeled data* | | | | | | | | | |
| MolFormer | 788.57 | 24.04 | -0.00 | 746.85 | 23.17 | -0.00 | 744.97 | 23.19 | -0.00 |
| MolT5-Large | 1081.23 | 25.13 | -0.37 | 1094.86 | 25.40 | -0.47 | 1098.16 | 25.37 | -0.47 |
| Chem-T5 | 1551.12 | 29.79 | -0.96 | 1189.38 | 26.14 | -0.60 | 1184.85 | 26.09 | -0.59 |
| Galactica-1.3B | 727.18 | 22.23 | 0.08 | 785.01 | 21.83 | -0.05 | 857.79 | 22.54 | -0.15 |
| **ChemBOMAS** | 633.68 | 19.47 | 0.20 | **650.00** | **19.55** | **0.13** | **593.76** | **18.52** | **0.20** |

First, ChemBOMAS demonstrated superior effectiveness and versatility in chemical performance regression. As shown in Table 1, ChemBOMAS achieved the highest prediction accuracy on the Arylation and Buchwald datasets, with $R^2$ scores exceeding the second-best model by 200% and 140%, respectively. On the Suzuki dataset, ChemBOMAS outperformed thirteen of the fourteen compared models, ranking second only to GLM4-9B GLM et al. (2024). However, the poor generality of GLM4-9B is evident from its near-zero $R^2$ scores on the other two datasets.

Second, task-specific fine-tuning proved essential. Despite their general capabilities, the off-the-shelf general-purpose LLMs GPT-4o OpenAI et al. (2024) and GPT-5 OpenAI (2025) performed poorly on this specialized regression task, consistently yielding strongly negative $R^2$ scores, lower than most fine-tuned models, which also confirms that these chemical datasets were not part of their training data. In contrast, traditional machine learning models, while computationally efficient, exhibit limited capability in capturing complex structure-activity relationships. Their $R^2$ values consistently remained below 0.20, with regression performance fluctuating significantly across different datasets. Among the open-source models, LLaMa-3.1-8B Grattafiori et al. (2024) exhibited a favorable balance of prediction accuracy and generalization, justifying its selection as the base model for the data-driven module of ChemBOMAS.

Furthermore, we investigated the impact of fine-tuning data volume on pseudo-data quality and BO performance in Appendix J.2. The predictive performance improved gradually as data volume increased from 0.00% to 32.00%. Notably, the $R^2$ value first turned positive and consistently exceeded 0.1 across all datasets at the 1% volume. Table 10 indicates that BO's optimization performance does not linearly correlate with the regression model's $R^2$; a value between 0.1 and 0.2 is sufficient for BO to identify high-performing conditions. Therefore, using 1% data volume for fine-tuning represents a rational and effective trade-off between cost and performance.

### 4.3.2 CLUSTER METHODS

We evaluated the impact of the search tree structure on BO by comparing scenarios with and without a tree, as well as trees constructed using three distinct strategies: expert guidance (Expert), data-driven approach (D-d), and knowledge-driven approach (K-d). All methods were initialized without pseudo-data to ensure a fair comparison.

Table 2: Comparison of Bayesian Optimization performance using different search space decomposition strategies: expert-guided clustering (Expert), data-driven clustering (D-d) and knowledge-driven clustering (K-d). 95% Iter and Best Iter are rounded to the nearest integer.

| Method | Best Found | | | Initial | | | 95% Iter | | | Best Iter | | |
|---|---|---|---|---|---|---|---|---|---|---|---|---|
| | Mean ± Std | 95% CI | $p$-val | Mean ± Std | 95% CI | $p$-val | Mean ± Std | 95% CI | $p$-val | Mean ± Std | 95% CI | $p$-val |
| **Suzuki** | | | | | | | | | | | | |
| Expert | 87.85 ± 6.44 | [83.25, 92.46] | - | 61.82 ± 14.78 | [51.24, 72.39] | - | 17 ± 13 | [7, 27] | - | 28 ± 11 | [19, 36] | - |
| D-d | 84.94 ± 7.74 | [79.40, 90.48] | > 0.05 | 58.29 ± 12.14 | [49.61, 66.98] | > 0.05 | 13 ± 14 | [3, 22] | > 0.05 | 31 ± 14 | [20, 41] | > 0.05 |
| K-d | 82.04 ± 4.49 | [78.83, 85.26] | > 0.05 | 66.94 ± 8.02 | [61.21, 72.68] | > 0.05 | 8 ± 11 | [0, 16] | > 0.05 | 18 ± 14 | [8, 28] | > 0.05 |
| **Arylation** | | | | | | | | | | | | |
| Expert | 82.05 ± 2.32 | [80.39, 83.71] | - | 50.07 ± 19.15 | [36.37, 63.77] | - | 16 ± 12 | [7, 25] | - | 26 ± 13 | [16, 35] | - |
| D-d | 82.20 ± 2.58 | [80.35, 84.05] | > 0.05 | 43.82 ± 27.57 | [24.09, 63.54] | > 0.05 | 14 ± 10 | [7, 22] | > 0.05 | 29 ± 8 | [23, 35] | > 0.05 |
| K-d | 81.28 ± 2.12 | [79.76, 82.80] | > 0.05 | 59.38 ± 17.79 | [46.66, 72.11] | > 0.05 | 11 ± 13 | [2, 21] | > 0.05 | 20 ± 14 | [10, 30] | > 0.05 |
| **Buchwald_sub1** | | | | | | | | | | | | |
| Expert | 79.37 ± 1.00 | [78.65, 80.09] | - | 35.63 ± 28.22 | [15.44, 55.81] | - | 7 ± 3 | [5, 9] | - | 23 ± 11 | [15, 31] | - |
| D-d | 79.27 ± 0.48 | [78.93, 79.61] | > 0.05 | 47.98 ± 30.69 | [26.02, 69.93] | > 0.05 | 6 ± 6 | [1, 10] | > 0.05 | 24 ± 14 | [14, 34] | > 0.05 |
| K-d | 80.25 ± 2.22 | [78.66, 81.83] | > 0.05 | 44.08 ± 26.71 | [24.98, 63.19] | > 0.05 | 4 ± 2 | [2, 6] | 0.0373 | 11 ± 7 | [5, 16] | 0.0062 |
| **Buchwald_sub2** | | | | | | | | | | | | |
| Expert | 53.01 ± 0.87 | [52.39, 53.63] | - | 12.23 ± 16.70 | [0.29, 24.18] | - | 14 ± 7 | [9, 19] | - | 30 ± 11 | [22, 39] | - |
| D-d | 52.67 ± 1.89 | [51.32, 54.02] | > 0.05 | 18.99 ± 18.12 | [6.03, 31.95] | > 0.05 | 15 ± 10 | [7, 22] | > 0.05 | 15 ± 10 | [8, 22] | 0.0010 |
| K-d | 53.23 ± 0.38 | [52.95, 53.50] | > 0.05 | 18.19 ± 22.47 | [2.12, 34.26] | > 0.05 | 18 ± 9 | [12, 25] | > 0.05 | 26 ± 7 | [22, 31] | > 0.05 |

The results, summarized in Table 2, demonstrate that the clustering methods derived from ChemBO-MAS—both D-d and K-d—consistently matched or surpassed the performance of the expert-guided approach across all benchmarks. This underscores the robustness and reliability of our automated framework for variable space decomposition. Furthermore, the knowledge-driven clustering strategy achieved superior optimization performance on more benchmarks compared to its data-driven counterpart, highlighting the value of incorporating structured chemical knowledge.

### 4.3.3 OPTIMIZATION

The analysis focuses on four key metrics: the optimal yield found (**Best Found**), the starting yield (**Initial**), the number of iterations required to reach 95% of the optimum (**95% Iter**), and the iteration where the best result was identified (**Best Iter**).

As shown in Figure 2, ChemBOMAS demonstrates consistent and superior performance over all baseline methods across the four benchmark datasets in terms of optimal result, convergence rate, initialization performance, and robustness.

In terms of final performance and convergence speed, ChemBOMAS identified the highest objective values—96.15% (Suzuki), 82.83% (Arylation), 79.97% (Buchwald_sub1) and 56.81% (Buchwald_sub2)—achieving convergence in just 3, 4, 23, and 2 iterations, respectively.

Regarding initialization performance, ChemBOMAS attained the highest initial performance on the Suzuki, Arylation, Buchwald_sub1, and Buchwald_sub2 datasets. It surpassed all baselines by the first iteration and proceeded to converge, highlighting its strong optimization capability.

Two additional observations further underscore the robustness of ChemBOMAS. First, it exhibited the lowest variance across ten independent optimization runs, indicating high stability. Second, its performance remained consistently effective and was largely unaffected by the subspace partition (see Appendix J.1 for details). These findings collectively confirm the reliability of our method.

To evaluate the generality of ChemBOMAS beyond chemistry, we assessed its optimization performance on a materials science benchmark. As shown in Table 13 (see Appendix K for dataset details), ChemBOMAS maintains competitive performance, demonstrating its applicability to other scientific domains.

To further validate the practical applicability of ChemBOMAS and preclude the possibility of knowledge leakage in the LLM, we conducted wet-lab optimization for a previously unreported

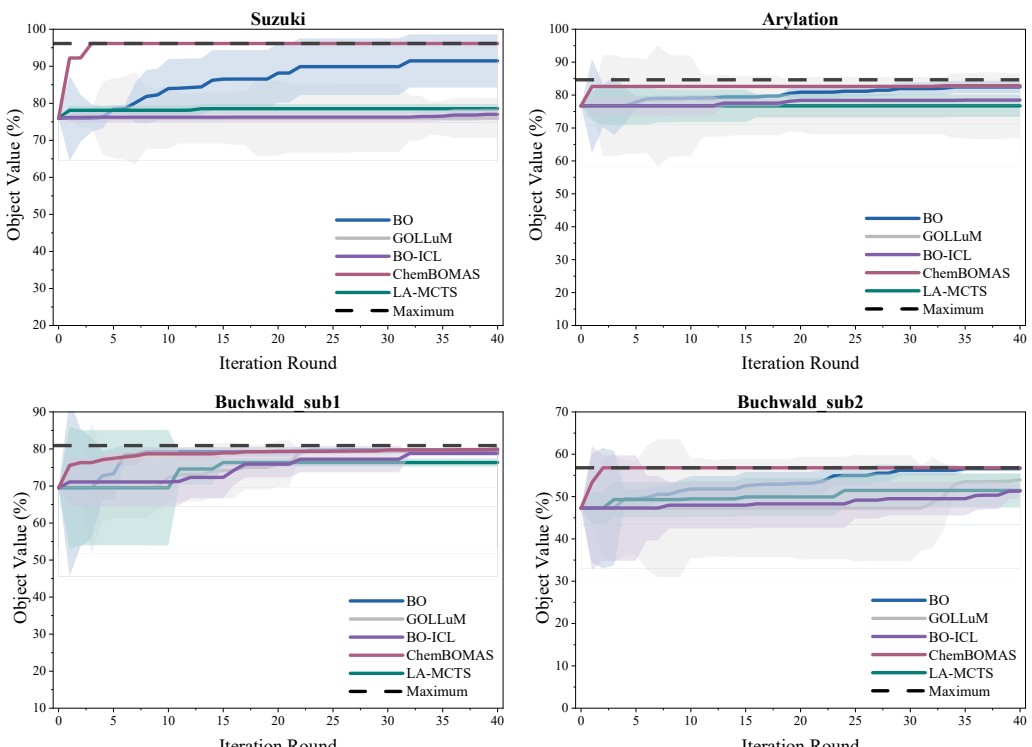

Figure 2: Optimization performance comparison between ChemBOMAS and baseline methods on the four benchmark datasets: (a) Suzuki, (b) Arylation, (c) Buchwald_sub1, and (d) Buchwald_sub2. ChemBOMAS exhibits accelerated convergence and achieves superior final performance with lower variance across all tasks, demonstrating its enhanced efficiency and robustness. Detailed data can be found in Appendix H.

chemical reaction (see Appendix I for details). As shown in Figure **??**, ChemBOMAS identified the optimal reaction condition with a yield of 96% after evaluating only 43 samples in 2 iterations. This result markedly outperforms the 15% yield obtained by a chemist using the traditional control variable method, demonstrating the framework's effectiveness in real-world scenarios.

## 4.4 ABLATION STUDY

We first evaluate the impact of pre-training and fine-tuning on the regression performance of Chem-BOMAS. The prediction accuracy is measured on the Suzuki, Arylation, and Buchwald datasets using Mean Squared Error (MSE), Mean Absolute Error (MAE), and the coefficient of determination ($R^2$)

Table 3: Impact of pre-training and fine-tuning strategies on the regression performance of ChemBO-MAS across the Suzuki, Arylation, and Buchwald datasets.

| Model Configuration | Suzuki | | | Arylation | | | Buchwald | | |
|---|---|---|---|---|---|---|---|---|---|
| | MSE↓ | MAE↓ | $R^2$ | MSE↓ | MAE↓ | $R^2$ | MSE↓ | MAE↓ | $R^2$ |
| w/o Pre & SFT | 2338.02 | 39.43 | -1.966 | 1797.88 | 32.54 | -1.413 | 1881.96 | 33.73 | -1.527 |
| Pre-train Only (w/o SFT) | 2407.22 | 40.27 | -2.054 | 1853.70 | 33.24 | -1.488 | 1795.72 | 32.52 | -1.411 |
| SFT Only (w/o Pre-train) | 685.55 | 20.50 | 0.130 | 679.72 | 19.57 | 0.088 | 739.27 | 20.57 | 0.007 |
| **Pre-train & SFT** | **633.68** | **19.47** | **0.196** | **650.00** | **19.55** | **0.128** | **667.16** | **19.51** | **0.104** |

The results in Table 3 indicate that the combined use of pre-training and supervised fine-tuning (SFT) yields the best predictive performance across all benchmarks. Notably, SFT alone (without pre-training) achieves the second-best performance and substantially outperforms models using only

pre-training or those without any training. This strongly suggests that supervised fine-tuning is the most critical component for adapting large models to chemical performance prediction tasks.

We further evaluate the contribution of each module by comparing the complete ChemBOMAS framework against three ablated versions: (i) without the data-driven module, (ii) without the knowledge-driven module, and (iii) without both modules. The results in Table 4 demonstrate that both modules are critical to the framework's performance. Ablating either module leads to a significant degradation in both optimization efficiency and final effectiveness.

For instance, on the Suzuki dataset, the full ChemBOMAS achieves the optimal value of 96.15% within only three iterations. In contrast, removing the data-driven module reduces the optimum to 83.26%. The performance of the single-module ablations is comparable to or only marginally better than the version lacking both modules, indicating that neither strategy alone is sufficient. These results underscore that the synergy between the knowledge-driven and data-driven strategies is essential for creating a highly efficient and robust optimization framework.

Table 4: Optimization performance of the full ChemBOMAS framework compared to its ablated variants. 95% Iter and Best Iter are rounded to the nearest integer.

| Method | Best Found | | | Initial | | | 95% Iter | | | Best Iter | | |
|---|---|---|---|---|---|---|---|---|---|---|---|---|
| | Mean ± Std | 95% CI | $p$-val | Mean ± Std | 95% CI | $p$-val | Mean ± Std | 95% CI | $p$-val | Mean ± Std | 95% CI | $p$-val |
| **Suzuki 1%** | | | | | | | | | | | | |
| full | 96.15 ± 0 | [96.15, 96.15] | - | 92.24 ± 0 | [92.24, 92.24] | - | 1 ± 0 | [1, 1] | - | 3 ± 0 | [3, 3] | - |
| w/o d-d | 83.26 ± 5.4 | [79.40, 87.12] | 0.0000 | 65.09 ± 9.88 | [58.02, 72.16] | 0.0000 | 10 ± 12 | [1, 18] | 0.0418 | 20 ± 13 | [11, 30] | 0.0022 |
| w/o k-d | 96.15 ± 0 | [96.15, 96.15] | ∞ | 58.91 ± 12.14 | [50.23, 67.60] | 0.0000 | 8 ± 5 | [5, 12] | 0.0016 | 9 ± 5 | [5, 12] | 0.0101 |
| w/o both | 91.44 ± 7.58 | [86.02, 96.87] | 0.0812 | 58.91 ± 12.14 | [50.23, 67.60] | 0.0000 | 12 ± 10 | [5, 19] | 0.0050 | 16 ± 7 | [10, 21] | 0.0004 |
| **Arylation 1%** | | | | | | | | | | | | |
| full | 82.83 ± 0.64 | [82.38, 83.29] | - | 82.63 ± 0 | [82.63, 82.63] | - | 1 ± 0 | [1, 1] | - | 4 ± 10 | [1, 11] | - |
| w/o d-d | 81.28 ± 2.12 | [79.76, 82.80] | 0.0680 | 59.38 ± 17.79 | [46.66, 72.11] | 0.0025 | 12 ± 13 | [2, 21] | 0.0347 | 20 ± 14 | [10, 30] | 0.0390 |
| w/o k-d | 79.76 ± 0.11 | [79.68, 79.84] | 0.0000 | 49.59 ± 15.1 | [38.79, 60.40] | 0.0001 | 8 ± 3 | [5, 10] | 0.0001 | 24 ± 11 | [16, 32] | 0.0020 |
| w/o both | 82.83 ± 1.77 | [81.57, 84.10] | 0.9969 | 49.59 ± 15.1 | [38.79, 60.40] | 0.0001 | 12 ± 9 | [5, 18] | 0.0036 | 27 ± 8 | [21, 33] | 0.0004 |
| **Buchwald_sub1 4%** | | | | | | | | | | | | |
| full | 80.45 ± 0.59 | [80.03, 80.87] | - | 79.52 ± 0 | [79.53, 79.53] | - | 1 ± 1 | [1, 1] | - | 10 ± 7 | [5, 15] | - |
| w/o d-d | 80.3 ± 2.18 | [78.74, 81.86] | 0.8218 | 50.35 ± 28.93 | [29.66, 71.05] | 0.0110 | 4 ± 2 | [2, 6] | 0.0030 | 14 ± 8 | [8, 20] | 0.0341 |
| w/o k-d | 80.66 ± 0.53 | [80.28, 81.04] | 0.3983 | 49.67 ± 18.48 | [36.45, 62.89] | 0.0006 | 8 ± 5 | [5, 12] | 0.0018 | 20 ± 10 | [13, 27] | 0.0736 |
| w/o both | 79.74 ± 0.42 | [79.44, 80.04] | 0.0064 | 53.57 ± 25.22 | [35.54, 71.61] | 0.0099 | 5 ± 2 | [3, 6] | 0.0013 | 12 ± 10 | [5, 19] | 0.5027 |
| **Buchwald_sub2 1%** | | | | | | | | | | | | |
| full | 56.81 ± 0 | [56.81, 56.81] | - | 53.33 ± 0 | [53.34, 53.34] | - | 2 ± 0 | [2, 2] | - | 2 ± 0 | [2, 2] | - |
| w/o d-d | 53.22 ± 0.38 | [52.95, 53.49] | 0.0000 | 16.58 ± 22.98 | [0.14, 33.02] | 0.0007 | 16 ± 8 | [10, 21] | 0.0004 | 20 ± 5 | [16, 23] | 0.0000 |
| w/o k-d | 56.81 ± 0 | [56.81, 56.81] | ∞ | 31.62 ± 13.55 | [21.92, 41.31] | 0.0007 | 8 ± 3 | [6, 10] | 0.0000 | 8 ± 3 | [6, 10] | 0.0000 |
| w/o both | 56.61 ± 0.61 | [56.18, 57.05] | 0.3434 | 31.62 ± 13.55 | [21.92, 41.31] | 0.0007 | 20 ± 9 | [13, 27] | 0.0002 | 22 ± 9 | [16, 28] | 0.0000 |

## 4.5 Wet Experiments

To further validate the practical value of our proposed method, an algorithm-driven **wet laboratory experiment** was conducted. Guided by ChemBOMAS, this study aimed to maximize product yield in a challenging chemical reaction optimization—the palladium-catalyzed cross-coupling of boronic esters with aryl chlorides. The details are shown in Appendix I.

This optimization task, provided by a pharmaceutical enterprise, was subject to four stringent constraints: (1) a **previously-unreported** chemical reaction, resulting in the complete absence of reference data; (2) a six-dimensional process parameter space, reportedly **exceeding seventy times** the scale of those in comparable published studies , making exploration highly challenging; (3) a cost-saving imperative requiring **a tenfold reduction** in catalyst loading relative to conventional levels, substantially hampering product formation; and (4) a hard budget of approximately **60 experimental runs** to curtail labor intensity.

As shown in Figure 3, during the wet experiment task, ChemBOMAS successfully identified the optimal reaction condition with a yield of 96%, markedly outperforming the 15% yield achieved by a chemist employing the traditional control variable method. Additionally, three noteworthy phenomena emerged. First, in the initial round, ChemBOMAS had attained the product yield of 90%, surpassing the target threshold of 75%. Second, the optimal reaction condition yielding 96% was discovered in the early stage of the optimization process, specifically in the second iteration. Third, as the optimization progressed, ChemBOMAS increasingly recommended reaction conditions with yields exceeding the 75% target threshold, indicating a continuous refinement of the surrogate model. The number of high-yielding conditions ($\geq 75\%$) identified in rounds one through five was one, two, three, four, and five, respectively. The strong initialization performance, rapid convergence,

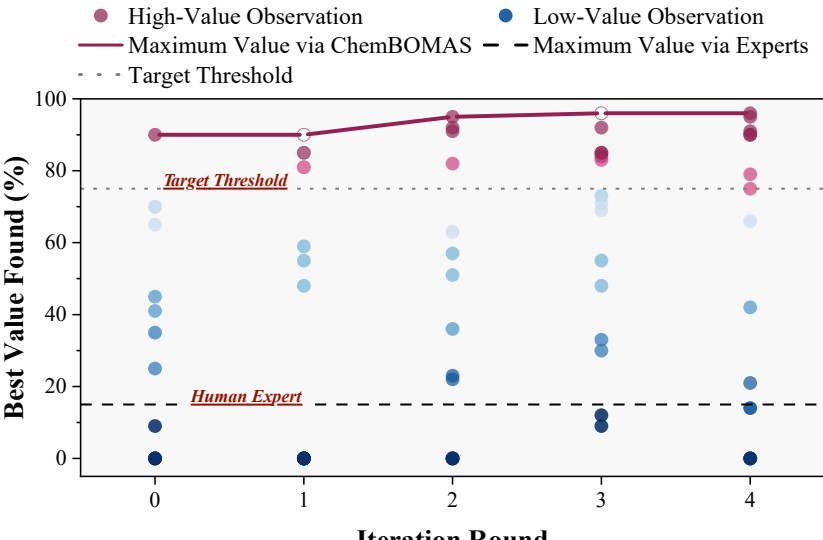

Figure 3: Wet laboratory experiment result. Comparison of Best Value Found (%) over Iteration Rounds', showing individual high and low-value observations. Lines indicate maximum values achieved via ChemBOMAS, human experts, and a target threshold.

and progressive model improvement collectively demonstrate the potential utility of the proposed method in real-world chemical optimization tasks.

## 5 LIMITATIONS

While ChemBOMAS represents a significant advancement in accelerating Bayesian Optimization for chemical reactions, its performance remains constrained by several factors. Most notably, the framework is inherently dependent on the accuracy and scope of the underlying LLM and its knowledge base; inference errors in literature parsing or incomplete corpora can lead to suboptimal search-space decompositions. In addition, the absence of explicit safety and feasibility constraints raises the risk of recommending theoretically optimal yet practically hazardous or infeasible conditions, underscoring the need for expert oversight or integration of safety-aware modules in future implementations.

## 6 CONCLUSION

ChemBOMAS presents an LLM-enhanced multi-agent framework designed to accelerate Bayesian Optimization in the context of chemical reactions. Through a synergistic combination of knowledge-driven search space decomposition and data-driven pseudo-data generation, this approach seeks to mitigate common challenges like data scarcity and complex reaction mechanisms. Results from benchmark evaluations, along with encouraging outcomes from wet-lab validation on a demanding, previously unreported industrial reaction—where ChemBOMAS showed improved performance compared to domain expert methods—suggest its potential for practical application. ChemBOMAS offers a promising direction for facilitating chemical discovery and enhancing the optimization of complex chemical processes.

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

## A  ILLUSTRATIVE EXAMPLE OF LLM ASSISTED CONSTRUCTION OF A REACTION OPTIMIZATION TREE

In the following illustrative example, we demonstrate how an LLM can assist in constructing an optimization tree for the reaction A + B → C in two steps, enabling efficient optimization of reaction conditions (e.g., catalyst, ligand, solvent, base), given that reactants A and B are fixed. In the first step, an LLM is used to infer the possible reaction type for A + B → C. Based on the inferred reaction type and specific optimization objectives (e.g., improving yield or selectivity), relevant scientific literature is retrieved. Literature acquisition can be done through manual downloads or by using publisher-provided APIs (noting that not all APIs are openly accessible). The collected literature is then used to construct a vector database to support the subsequent retrieval process. Using the information from the literature in the vector database, the LLM is queried via analyzing literature to determine the relative importance of different reaction conditions (variables) on the reaction objects, generating a ranked list. For instance, the LLM might determine the order of influence as: Catalyst > Ligand > Solvent > Base. Further queries to the LLM identify the key physicochemical properties within each category that significantly influence the chemical reaction performance. For example, within the ligand category, the LLM may highlight "steric and electronic effects" as crucial physicochemical properties. Subsequently, detailed information regarding the key physicochemical properties of each ligand candidate is retrieved from online databases, after which the LLM clusters these ligand candidates into subsets based on similarities in "steric and electronic effects".

In the second step, the optimization tree is constructed based on the variable importance ranking and clustering results. The first level of the tree corresponds to the most important variable—the catalyst. At the first level, several child nodes can be established, representing different subsets of catalyst candidates clustered by property similarity. The second level of the tree corresponds to the next most important variable—the ligand. Under each catalyst subset node at the first level, additional child nodes branch out, representing various subsets of ligand candidates categorized by their physicochemical properties. This process continues iteratively, layer by layer, incorporating additional variables (e.g., solvent, base) until the complete optimization tree is constructed.

## B  UPDATE ON CHEMBOMAS DURING OPTIMIZATION

After receiving the observation feedback on each round of the experiment, ChemBOMAS would update. First, the data module would be retrained with the prior and newly acquired data points, and then infer the unsampled data points to generate pseudo-labels. Second, the optimization tree would recount the visit number and value of each node to refine the identified hot regions. Third, with the updated observations, pseudo-labels, and refined hot regions, the BO module would recommend next-round reaction conditions, targeting potentially higher object values.

## C  DUAL-STRATEGY REFINEMENT FOR ENHANCED OPTIMIZATION

To mitigate the detrimental influence of noise and redundancy inherent in generated pseudo-data, we introduced a dual-pronged refinement strategy. This approach was designed to dynamically curate the pseudo-dataset, ensuring its quality and diversity throughout the optimization process. The strategy combined a local, similarity-based removal mechanism with a global, performance-driven pruning policy. This ensured that the pseudo-dataset remained a reliable and informative asset for guiding the optimization, particularly in complex search spaces.

**Data Similarity (Local Removal):** We utilized the final token embedding, $\mathbf{e}(\mathbf{x}) = \text{LLM}^{[T]}_{\theta_{\text{LLM}},\phi_{\text{LoRA}}}(\mathbf{x})$, to calculate cosine similarity. Upon acquiring a new real data point $(\mathbf{x}_{\text{new}}, y_{\text{new}})$, the pseudo-dataset was updated by removing points that were too similar:

$$\mathcal{D}_{\text{pseudo}} \leftarrow \mathcal{D}_{\text{pseudo}} \setminus \left\{ (\mathbf{x}_j, \hat{y}_j) \in \mathcal{D}_{\text{pseudo}} \mid \frac{\mathbf{e}(\mathbf{x}_j) \cdot \mathbf{e}(\mathbf{x}_{\text{new}})}{\|\mathbf{e}(\mathbf{x}_j)\|\|\mathbf{e}(\mathbf{x}_{\text{new}})\|} > \tau \right\} \tag{2}$$

where $\tau$ was a predefined similarity threshold.

**Observed Performance (Global Removal):** As the optimization progresses, the model should be encouraged to explore more broadly. Therefore, based on the predicted performance values $\hat{y}$ of

the pseudo-points, we randomly discarded a proportion of pseudo-data, starting from those with high predicted performance downwards. The probability of discarding a pseudo-point $(\mathbf{x}_j, \hat{y}_j)$ was a monotonically increasing function of its predicted performance.

These generated pseudo-points could also provide further support for the construction of the knowledge-guided optimization tree in Stage 1. By adjusting the LLM's temperature parameter during generation, we could produce a set of candidate tree structures, $\{\mathcal{T}_1, \mathcal{T}_2, \ldots, \mathcal{T}_K\}$. Using the pseudo-points, we quantitatively evaluated these candidates. Let $\mathcal{N}(\mathcal{T}_k)$ be the set of leaf nodes of tree $\mathcal{T}_k$, and let $\mathcal{D}_{\text{pseudo}}^{(j)}$ be the subset of pseudo-points belonging to node $j \in \mathcal{N}(\mathcal{T}_k)$. The tree structure that minimized the weighted average of intra-node variances was selected as the optimal one:

$$\mathcal{T}^* = \arg\min_{\mathcal{T}_k} \sum_{j \in \mathcal{N}(\mathcal{T}_k)} \frac{|\mathcal{D}_{\text{pseudo}}^{(j)}|}{|\mathcal{D}_{\text{pseudo}}|} \text{Var}(\{\hat{y} \mid (\mathbf{x}, \hat{y}) \in \mathcal{D}_{\text{pseudo}}^{(j)}\}) \tag{3}$$

This ensured the selection of a tree that best partitions the search space into regions of homogeneous performance, guiding the subsequent optimization more effectively.

## D  COMPLETE ALGORITHM PROCESS

To provide a comprehensive and formal description of the ChemBOMAS framework, we present its complete algorithmic process in Algorithm D. This pseudocode encapsulates the synergistic, two-stage optimization strategy detailed in Section 3.

The algorithm begins by initializing a hierarchical search tree using the LLM-guided knowledge-driven approach as shown in Section 3.4. The main loop then iterates through the coarse-grained optimization phase, where a UCB policy navigates the tree to select a promising subspace. Within this selected subspace, the algorithm transitions to the fine-grained, data-driven optimization phase. Here, a standard Bayesian Optimization procedure is executed, but it is significantly accelerated by an informative prior constructed from both real experimental data and pseudo-data generated by the fine-tuned LLM regressor (Section 3.3).

After each experimental evaluation, the results are backpropagated to update the value estimates of the nodes in the search tree, refining the knowledge-driven search for subsequent iterations. This process continues until the predefined budget of evaluations is exhausted, ultimately returning the best-performing experimental configuration found.

## E  DETAILS OF PROMPTS

As outlined in the main text, our methodology leverages LLMs to support several critical tasks in reaction optimization, such as analyzing literature, assessing parameter significance, and understanding physicochemical properties to inform the construction of a hierarchical optimization tree. This appendix section presents a detailed overview of the specific prompts designed to guide the LLM in executing these crucial Tasks.

### E.1  PROMPT OF DATA MODULE

As detailed in the Section 3.3, our pre-training phase employs a conditional prediction task. Given the reactants and products, the model's objective is to predict the corresponding reaction conditions. This process utilizes a Causal Language Modeling (CLM) loss, where the model learns to predict the next token in the sequence of reaction conditions.

To provide concrete examples of the input format for this task, this appendix section presents a selection of prompts utilized during the pre-training phase. These prompts typically consist of the reactants, products, and the target reaction condition sequence that the model is trained to predict. Furthermore, in line with the methodology described in the main text, these input sequences are augmented with functional group annotations (generated via RDKit) to enhance the model's chemical awareness; the augmentation of the prompt is also reflected in the examples provided below.

---

**Algorithm 1** The Complete Algorithm Process of ChemBOMAS

---

**Input:** Search space $\mathcal{X}$, black-box objective function $h(\cdot)$, coarse iterations $N_{coarse}$, fine iterations per evaluation $N_{fine}$, exploration constant $C_p$, fine-tuned LLM regressor $f_{\theta_{\text{MLP}}}(\text{LLM}_{\theta_{\text{LLM}}, \phi_{\text{LoRA}}}(\cdot))$.

**Initialize:**

Construct hierarchical search tree via LLM-guided space partitioning (see Section 3.4).

Construct hierarchical search tree $\mathcal{T}$ by partitioning $\mathcal{X}$ via LLM-driven analysis.

Initialize value estimate $Q_v \leftarrow 0$, visit count $n_v \leftarrow 0$ for all nodes $v \in \mathcal{T}$.

Initialize global set of real experimental data $\mathcal{D}_{\text{real}} \leftarrow \emptyset$.

**procedure** MAIN LOOP
    **for** $i = 1$ to $N_{coarse}$ **do**
        $v_{\text{current}} \leftarrow \text{root}(\mathcal{T})$
        $\text{path} \leftarrow [v_{\text{current}}]$

        **// Stage 1: Knowledge-driven Strategy**
        **while** $v_{\text{current}}$ is not a leaf node **do**
$$v_{\text{current}} \leftarrow \underset{v_k \in \text{children}(v_{\text{current}})}{\arg\max} \left( \frac{Q_{v_k}}{n_{v_k}} + C_p \sqrt{\frac{\ln n_{v_{\text{current}}}}{n_{v_k}}} \right)$$
            Append $v_{\text{current}}$ to path.
        **end while**
        Let $\mathcal{S}_j$ be the promising subspace corresponding to the leaf node $v_{\text{current}}$.

        **// Stage 2: Data-driven Strategy**
        $(y_{\text{new}}, \mathbf{x}_{\text{new}}) \leftarrow \text{BO}(\mathcal{S}_j, N_{fine}, \mathcal{D}_{\text{real}}, f_{\theta_{\text{MLP}}}(\text{LLM}_{\theta_{\text{LLM}}, \phi_{\text{LoRA}}}(\cdot)))$
        $\mathcal{D}_{\text{real}} \leftarrow \mathcal{D}_{\text{real}} \cup \{(\mathbf{x}_{\text{new}}, y_{\text{new}})\}$.
        **for** $v$ in path **do**                            ▷ Backpropagation
            $n_v \leftarrow n_v + 1$
            $Q_v \leftarrow Q_v + y_{\text{new}}$
        **end for**
    **end for**
**end procedure**

**function** BO($\mathcal{S}_j, N_{fine}, \mathcal{D}_{\text{real}}, \text{LLM\_regressor}$)
    //Initialize surrogate model with LLM-generated pseudo-data.
    Generate pseudo-dataset $\mathcal{D}\text{pseudo} = \{(\mathbf{x}_k, \hat{y}_k)\}_{k=1}^{M}$ for $\mathbf{x}_k \in \mathcal{S}_j$ using LLM\_regressor.
    Let $\mathcal{D}\text{real}^{(j)} = \{(\mathbf{x}, y) \in \mathcal{D}_{\text{real}} \mid \mathbf{x} \in \mathcal{S}j\}$.
                          ▷ Fit Gaussian Process (GP) on combined data to serve as an informative prior.
    Initialize GP surrogate model $\mathcal{M}$ on $\mathcal{D}\text{pseudo} \cup \mathcal{D}_{\text{real}}^{(j)}$.
    **for** $k = 1$ to $Nfine$ **do**
                         ▷ Select next point by maximizing the acquisition function $\alpha(\cdot)$.
        $x\text{next} \leftarrow \arg\max_{\mathbf{x} \in \mathcal{S}j} \alpha(\mathbf{x} | \mathcal{M})$
        $y\text{next} \leftarrow h(\mathbf{x}\text{next})$         ▷ Perform real experiment to get objective value.
        $\mathcal{D}\text{real}^{(j)} \leftarrow \mathcal{D}_{\text{real}}^{(j)} \cup \{(\mathbf{x}_{\text{next}}, y_{\text{next}})\}$

        **// Apply refinement strategy**
        Update $\mathcal{D}\text{pseudo}$ by removing points based on similarity and performance rules.
        Update GP surrogate model $\mathcal{M}$ with new data $\{(\mathbf{x}\text{next}, y_{\text{next}})\}$ and pruned $\mathcal{D}\text{pseudo}$.
    **end for**
    **return** $(y\text{next}, \mathbf{x}_{\text{next}})$         ▷ Return the result of the last experiment.
**end function**

**Output:** The configuration $\mathbf{x}^*$ with the highest observed objective value $h(\mathbf{x}^*)$ from $\mathcal{D}_{\text{real}}$.

---

**Prompt of Condition Prediction Pretraining:**   For the condition prediction pre-training, the input prompts are structured to provide the model with comprehensive reaction information. Typically, a prompt is formatted as: [Reactants_SMILES]; [Products_SMILES]; [Reaction Type];[Target_Reaction_Conditions]. Prior to constructing these prompts, the SMILES strings for both reactants and products are canonicalized using RDKit. This normalization step ensures a standardized and consistent representation of molecular structures, which is vital for robust model training. The model then processes this complete sequence, aiming to predict the [Target_Reaction_Conditions] segment token by token, guided by the Causal Language Modeling objective and conditioned on the preceding reaction type, reactants, and products.

To further clarify the input structure for this prediction task, the following examples demonstrate the format used:

---

**Condition Prediction Pre-training Prompts**

"reaction": "Here is a chemical reaction.
Reactants are: C1=CC=CC=2C3=CC=CC=C3N(C12)CC#C,BrC#CCCCCO.
Product is: C1=CC=CC=2C3=CC=CC=C3N(C12)CC#CC#CCCCCO.
Reaction type is Cadiot-Chodkiewicz coupling.",
"condition": "The reaction conditions of this reaction are:
Solvent: O,CN(C=O)C,CN(C=O)C. Catalyst: Cl[Cu]. Atmosphere: N#N. Additive: C(C)N,[Na]Cl,Cl.NO. ", "reaction_type": "Cadiot-Chodkiewicz coupling",

---

**Prompt of Yield Prediction Fine-tuning:**   To fine-tune LLM for precise prediction of chemical reaction yields, we combine key chemical information—including reaction type, products, reactants, and reaction conditions—into structured prompts. This approach guides the model to learn the complex relationships between these variables and reaction outcomes, enabling it to output a specific numerical prediction.

Below is an example prompt for yield prediction fine-tuning from the reactants in the Suzuki coupling dataset.

---

**An example prompt for yield prediction fine-tuning**

Here is a chemical reaction:
Reactants are: CCc1cccc(CC)c1.Clc1ccc2ncccc2c1, Cc1ccc2c(cnn2C2CCCCO2)c1B(O)O.
Product is: Cc1ccc2c(cnn2C2CCCCO2)c1-c1ccc2ncccc2c1.
Reaction type is Suzuki Miyaura.
The reaction conditions of this reaction are:
Solvent: $CC\equiv N \cdot O$
Ligand: CC(C)(C)P(C(C)(C)C)C(C)(C)C
Base: $[Na^+] \cdot [OH-]$
What is the yield of this reaction?

---

E.2   PROMPTS OF KNOWLEDGE MODULE

The Knowledge Module, as described in Section 3.4, employs the LLM to systematically analyze chemical literature and physicochemical data. This involves ranking the impact of various reaction parameters and classifying components based on their physicochemical properties.

**Variable Candidates Clustering Prompt:** The prompt guides the LLM to identify key physicochemical properties of each variable and cluster variable candidates based on their similarity in the physicochemical properties. Below is an example of the prompt for variable candidates classification.

> **Prompt for Variable Candidates Classification Based on Physicochemical Data**
>
> **Objective:**
> Classify the provided list of candidate chemical substances into NO MORE THAN THREE groups according to the [Specified_physicochemical_Properties], or place them all in ONE class if justified.. Your primary method for classification must be the utilization of quantitative data that would typically be found in a comprehensive physicochemical property database.
>
> **Crucial Instructions:**
> **Prioritize Quantitative Data:** For each substance and property, you should first attempt to classify it based on specific, measurable, quantitative values (e.g., pKa for basicity/acidity, dielectric constant for polarity, boiling point for volatility, specific functional group counts).
> **Minimize General Knowledge/Intuition:** Avoid relying on your general, unquantified chemical knowledge or intuition. If a quantitative value from the "database" directly supports a classification, state that. If a direct value isn't typically used for a category but strong structural indicators (which could be quantified, e.g., number of H-bond donors) point to it, explain this as an inference based on data-like principles.
> **Adhere to Provided Categories:** Classify substances strictly into the categories provided for each property. If a substance does not clearly fit or straddles categories based on (assumed) data, note this ambiguity.
>
> **Candidate Substances to Classify:**
>     [TYPE] : [CANDIDATE_SUBSTANCES_LIST]
>
> **Provided Literature:**
>     [LITERATURE_1]
>     [LITERATURE_2]
>     ...
>
> **Available Tools:**
>     [PubMedToolkit], [PubChemToolkit], [GoogleSearchToolkit]

## F    BENCHMARK DETAIL

This section provides further details on the benchmark datasets used for evaluating ChemBOMAS.

### F.1    DATASET FOR LLM PRE-TRAIN

The Pistachio dataset employed during the pre-training phase is a large-scale reaction information repository. Its core data was systematically extracted from the full texts of US patents and European patents through automated text mining techniques. To enhance data diversity and accuracy, the dataset integrates information from multiple sources, including: - Structured data parsed from ChemDraw files embedded directly within patent documents - Records sourced from specialized chemical databases such as Reaxys - Exported data from select electronic laboratory notebooks. The dataset contains a total of 19.17 million chemical reactions. In this project, we primarily utilize the reaction SMILES strings for model pre-training.

### F.2    DATASET FOR LLM FINE-TUNE

To conduct a rigorous and unbiased evaluation of model performance, we selected a series of publicly available benchmark datasets widely used in the field of chemical reaction optimization. The core strength of these datasets lies in their completeness: all were generated via high-throughput automated experimental platforms and encompass experimental results for every variable combination within a clearly defined chemical space (full factorial design). This exhaustive coverage effectively eliminates sampling bias, enabling deterministic quantitative evaluation of algorithmic recommendation performance against known experimental ground truth.

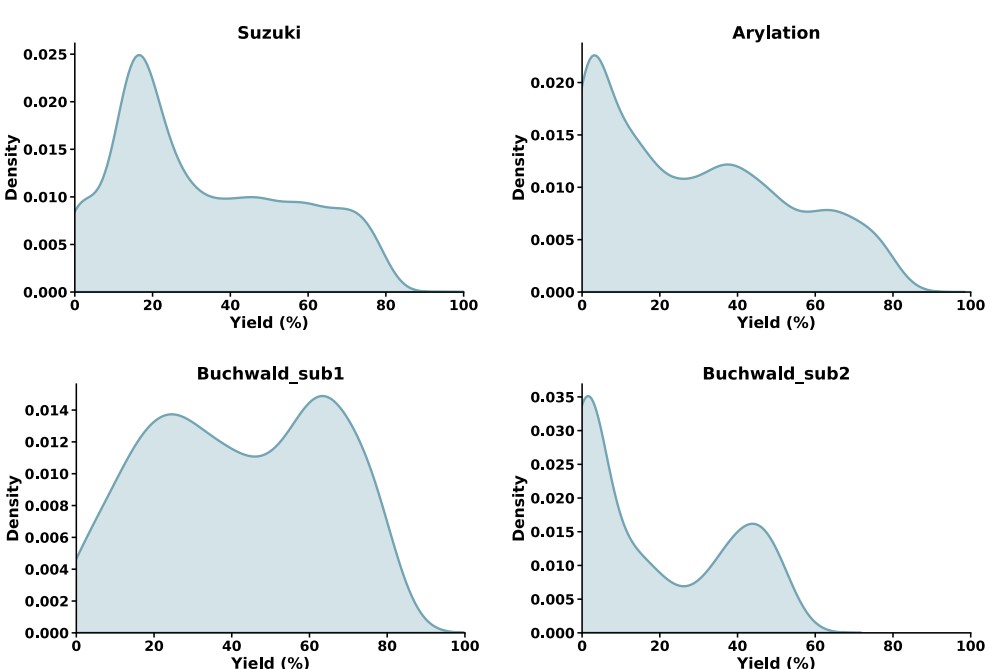

Figure 4: KDE plots illustrating the yield distributions for the four benchmark datasets.

Specifically, we employed three recognized benchmark datasets: Suzuki, Arylation, and Buchwald reactions. During fine-tuning, we randomly sampled 1% of data from each dataset as training samples to adjust the pre-trained model.

**Suzuki** originates from the automated nanomolar-scale flow screening study reported by Perera et al. in 2018. The chemical space of the experiments comprised a full factorial combination of 4 halogenated quinolines, 3 boronic acid derivatives, 11 phosphine ligands, 7 bases, and 4 solvents. All reactions were conducted under uniform conditions (100 °C, 1-minute residence time, 9:1 organic/aqueous phase). Reaction yields were detected via dual UPLC-MS online detection and uniformly calibrated. The data is comprehensive and highly consistent, making it one of the widely adopted validation standards in the field.

**Arylation** was reported by Shields et al. in 2021 for Bayesian optimization studies. Its chemical space was generated via a full factorial design comprising 12 phosphine ligands, 4 bases, 4 solvents, 3 temperature gradients, and 3 concentration gradients. All experiments were conducted at high throughput in 96-well plates, with yields precisely quantified via UHPLC-MS coupled with internal standard methods. This dataset features no duplicates or missing data, exhibits uniform variable distribution, and has been validated by 50 practicing chemists, establishing it as a critical benchmark for optimizing C-H functionalization reactions.

**Buchwald** was published by Ahneman et al. in 2018, this dataset aims to predict yields of C-N coupling reactions via machine learning. Experiments were conducted in nanomolar-scale high-throughput format using 1536-well plates, systematically examining all combinations of 15 aryl halides, 4 ligands, 3 bases, and 23 isoxazole additives. All reactions proceeded under standard conditions (60 °C, DMSO, 16 hours), with yields quantified by LC-MS. This dataset is complete with no missing values, serving as an authoritative open-access resource for studying additive effects and modeling complex reaction systems.

Table 5: Descriptive statistics of the four reaction datasets. The table summarizes key statistical measures for the reaction yields, including measures of central tendency, dispersion, and distribution shape.

| Statistic | Suzuki | Arylation | Buchwald_sub1 | Buchwald_sub2 |
|---|---|---|---|---|
| Total data points (N) | 5030 | 3678 | 629 | 765 |
| Maximum Yield (%) | 96.15 | 84.65 | 80.91 | 56.81 |
| Minimum Yield (%) | 0.00 | 0.00 | 0.00 | 0.00 |
| Mean (%) | 33.04 | 29.05 | 42.24 | 18.71 |
| Median (%) | 26.86 | 25.53 | 42.21 | 11.34 |
| Standard Deviation (%) | 22.47 | 23.79 | 22.86 | 18.98 |
| 25% Quantile (%) | 15.26 | 6.87 | 23.14 | 0.72 |
| 75% Quantile (%) | 51.27 | 47.14 | 63.01 | 38.77 |

### F.3 DATASET FOR BAYESIAN OPTIMIZATION

For Bayesian optimization tasks, we employed four benchmark datasets: Suzuki, Arylation, Buchwald_sub1, and Buchwald_sub2. The latter two originate from partitions of the aforementioned Buchwald-Hartwig dataset. To ensure consistency of target products within the optimization space, the original dataset was first divided into five independent subsets based on product molecular structures. We observed distinct high-yield and low-yield patterns in the reaction yields of these subsets. To ensure comprehensive evaluation, we selected one representative subset from each category, naming them Buchwald_sub1 and Buchwald_sub2, respectively. Table 5 summarizes key descriptive statistics for these four datasets, while Figure 4 visually depicts their respective yield distributions via kernel density estimation (KDE) plots. These datasets exhibit distinct statistical characteristics, with average yields ranging from 18.71% to 42.24% and diverse distribution shapes. Collectively, they form a challenging optimization problem that effectively tests algorithm performance across varying data environments.

### F.4 COMPARATIVE ALGORITHMS

To evaluate the efficacy of our proposed method, we benchmark it against four algorithms representing diverse approaches to black-box optimization. These baselines were strategically selected to strictly validate specific components of the ChemBOMAS framework: (1) **BO** serves as the classical standard. (2) **BO-ICL** relies entirely on LLM inference, serving as a reference for our data-driven module. (3) **LA-MCTS** employs a tree-structured partitioning mechanism analogous to our knowledge-driven module. (4) **GOLLuM** operates in the latent space, sharing the core philosophy of Latent Bayesian Optimization (LBO).

To ensure a fair comparison, we standardized the experimental protocol across all methods involving Bayesian optimization components. The detailed parameters and computational costs across all methods are shown in Table 6 The specific configurations and selection rationale for each baseline are detailed below:

- Traditional Bayesian Optimization (**BO**): This serves as the classical fundamental baseline. It utilizes a Gaussian Process (GP) with a Matérn kernel as the surrogate model to approximate the objective function. Consistent with the general protocol, it employs EI to guide sequential sampling. Input features are processed using one-hot encoding, as prior research indicates that more complex encoding schemes yield negligible benefits in this context Taylor et al. (2023); Shields et al. (2021a).

- Bayesian Optimization with In-Context Learning (**BO-ICL**): This method integrates a frozen LLM with BO, as proposed by Ramos et al. (2023b). Instead of fine-tuning, it leverages the in-context learning capability of the LLM to function as a surrogate model. Since this approach relies entirely on the LLM to drive optimization, it serves as a direct comparator to evaluate the effectiveness of the LLM-based regression strategy in our data-driven module. Our implementation utilizes gpt-3.5-turbo with a temperature setting of 0.7. The model predicts outcomes and uncertainty by retrieving the $k = 3$ nearest neighbors from the optimization history to construct the textual context prompts.

Table 6: Detailed Experimental Settings and Computational Cost.

|  | BO | BO-ICL | GoLLum | LA-MCTS | ChemBOMAS |
|---|---|---|---|---|---|
| *Experimental Settings* | | | | | |
| Initialization | | | 1% of dataset | | |
| BO Batch Size | | | 0.1% of dataset | | |
| Acq. Function | | | Types of Expected Improvement | | |
| Iterations | | | 40 | | |
| Repeat Campaigns | | | 10 | | |
| Kernel Function | | | Matérn Family ($\nu = 3/2, 5/2,$ or $\infty$) | | |
| *Computational Cost per campaign (seconds)* | | | | | |
| Suzuki | 144 | 19 | 22 | 2575 | 212 |
| Arylation | 88 | 20 | 21 | 1957 | 100 |
| Buchwald_sub1 | 21 | 22 | 13 | 1734 | 28 |
| Buchwald_sub2 | 17 | 19 | 13 | 1767 | 22 |

- Latent Action Monte Carlo Tree Search (**LA-MCTS**): This is a meta-algorithm designed for high-dimensional optimization Wang et al. (2020b). It employs Monte Carlo Tree Search to dynamically partition the search space into high- and low-performance regions. This hierarchical partitioning strategy closely mirrors the tree-structured search logic of our knowledge-driven module, making it an ideal baseline to assess the efficiency of our LLM-guided space decomposition. The tree search policy utilizes the Upper Confidence Bound (UCB) as the acquisition function, with a dynamically adjusted exploration parameter $\kappa$. A local optimizer is subsequently deployed within the promising subregions identified by the tree.

- Gaussian Process Optimized LLMs (**GOLLuM**): Representing a deeper fusion of LLMs and Bayesian optimization Ranković and Schwaller (2025), this method utilizes the LLM's embedding space as a deep kernel for the GP, jointly optimizing the embedding and GP hyperparameters. Its core mechanism of performing BO within a latent space shares the fundamental philosophy of LBO, providing a benchmark for latent-space-based strategies. We employed T5-base as the featurizer to extract representations, utilizing a random initialization method for the embedding space optimization.

## G  QUALITATIVE COMPARISON OF OPTIMIZATION TRAJECTORIES

To qualitatively assess how well the automated clustering strategies of ChemBOMAS emulate expert-level reasoning, we visualized the optimization progress. Figure 5 provides a comparative heatmap of the "Best Found" objective value over 40 iterations for three different search tree configurations: one guided by human experts, one by our knowledge-driven module (K-d), and one by our data-driven module (D-d).

The visual evidence strongly suggests that both automated ChemBOMAS strategies produce optimization trajectories that are remarkably consistent with the expert-guided approach. The color progression—from blue (lower values) to red (higher values)—is highly similar across all three methods. This indicates that the subspaces identified as promising by the LLM-driven modules align well with those selected by human domain experts. The ability of both the knowledge-driven and data-driven variants to rapidly progress towards high-yield regions in a manner analogous to the expert baseline underscores the effectiveness of our framework in automatically structuring the search space in a chemically meaningful way. This qualitative alignment provides further confidence in the robustness and practical utility of ChemBOMAS for real-world chemical optimization tasks.

## H  COMPLETE DATA FOR OPTIMIZATION TASK EXPERIMENTS

Table 7 presents the comprehensive performance metrics for ChemBOMAS and four baseline methods across the Suzuki, Arylation, Buchwald_sub1, and Buchwald_sub2 datasets. Results are averaged

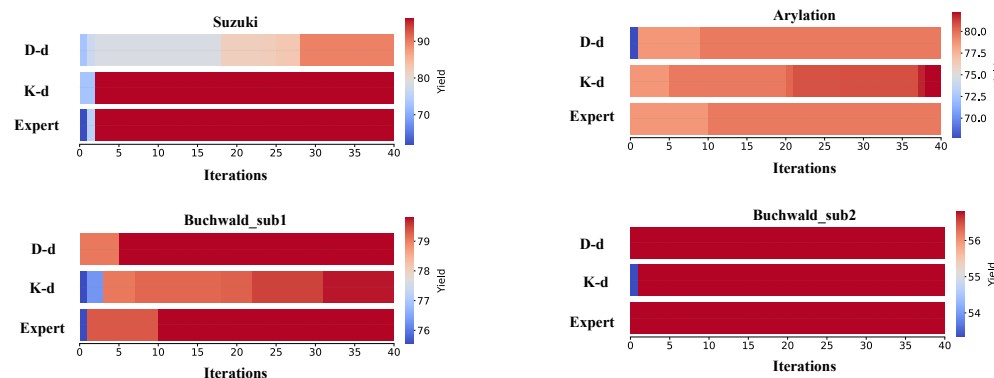

Figure 5: Heatmap of the best-found objective value over 40 iterations on the Suzuki dataset for three different tree-building strategies. Each colored block represents the highest value discovered up to that iteration, with the color scale progressing from blue (low) to red (high). The visual similarity in the optimization trajectories demonstrates that both the knowledge-driven (K-d) and data-driven (D-d) methods closely mirror the performance progression of the expert-guided approach.

over 10 independent runs. The $p$-values are calculated using a one-sided $t$-test comparing the "Best Found" and "Initial" performance of each baseline against ChemBOMAS.

Table 7: Comparison of Methods across Datasets. 95% Iter and Best Iter are rounded to the nearest integer.

| Method | Best Found | | | Initial | | | 95% Iter | | | Best Iter | | |
|---|---|---|---|---|---|---|---|---|---|---|---|---|
| | Mean ± Std | 95% CI | $p$-val | Mean ± Std | 95% CI | $p$-val | Mean ± Std | 95% CI | $p$-val | Mean ± Std | 95% CI | $p$-val |
| **Suzuki** | | | | | | | | | | | | |
| BO | 91.45 ± 7.58 | [86.02, 96.87] | 0.0812 | 58.91 ± 12.14 | [50.23, 67.60] | 0.0000 | 12 ± 10 | [5, 19] | 0.0050 | 16 ± 7 | [10, 21] | 0.0004 |
| BO-ICL | 80.37 ± 5.93 | [74.14, 86.59] | 0.0013 | 76.02 ± 0.07 | [75.94, 76.10] | 0.0000 | 6 ± 12 | [1, 18] | 0.3632 | 21 ± 13 | [7, 34] | 0.0197 |
| LA-MCTS | 78.43 ± 1.15 | [77.60, 79.25] | 0.0000 | 77.70 ± 1.67 | [76.51, 78.90] | 0.0000 | 1 ± 0 | [1, 1] | ∞ | 3 ± 4 | [0, 6] | 0.8751 |
| GOLLuM | 78.07 ± 6.67 | [73.30, 82.85] | 0.0000 | 79.10 ± 26.53 | [26.27, 26.79] | 0.0000 | 25 ± 9 | [18, 31] | 0.0000 | 31 ± 7 | [26, 36] | 0.0000 |
| ChemBOMAS | 96.15 ± 0.00 | [96.15, 96.15] | - | 92.24 ± 0.00 | [92.24, 92.24] | - | 1 ± 0 | [1, 1] | - | 3 ± 0 | [3, 3] | - |
| **Arylation** | | | | | | | | | | | | |
| BO | 82.83 ± 1.77 | [81.57, 84.10] | 0.9969 | 49.59 ± 15.10 | [38.79, 60.40] | 0.0000 | 12 ± 9 | [0, 18] | 0.0036 | 27 ± 8 | [21, 33] | 0.0004 |
| BO-ICL | 78.63 ± 1.21 | [77.36, 79.91] | 0.0005 | 76.43 ± 0.77 | [5.62, 77.23] | 0.0000 | 1 ± 0 | [1, 1] | ∞ | 27 ± 13 | [13, 41] | 0.0047 |
| LA-MCTS | 74.12 ± 4.03 | [71.23, 77.01] | 0.0001 | 67.85 ± 7.53 | [2.46, 73.24] | 0.0002 | 6 ± 8 | [1, 11] | 0.0858 | 13 ± 11 | [5, 21] | 0.0976 |
| GOLLuM | 76.99 ± 8.39 | [70.99, 83.00] | 0.0557 | 23.77 ± 1.17 | [22.93, 24.60] | 0.0000 | 21 ± 11 | [13, 29] | 0.0003 | 29 ± 11 | [21, 36] | 0.0002 |
| ChemBOMAS | 82.83 ± 0.64 | [82.38, 83.29] | - | 82.63 ± 0.00 | [82.63, 82.63] | - | 1 ± 0 | [1, 1] | - | 4 ± 10 | [1, 11] | - |
| **Buchwald_sub1** | | | | | | | | | | | | |
| BO | 79.74 ± 0.42 | [79.44, 80.04] | 0.3216 | 53.57 ± 25.22 | [35.54, 71.61] | 0.0222 | 5 ± 2 | [3, 6] | 0.7811 | 12 ± 10 | [5, 19] | 0.0281 |
| BO-ICL | 78.26 ± 2.53 | [75.61, 80.91] | 0.1692 | 69.78 ± 0.64 | [69.11, 70.45] | 0.0000 | 7 ± 6 | [1, 13] | 0.4604 | 19 ± 13 | [5, 33] | 0.5129 |
| LA-MCTS | 75.52 ± 3.95 | [72.69, 78.34] | 0.0056 | 71.10 ± 13.22 | [1.64, 80.55] | 0.3141 | 2 ± 4 | [1, 6] | 0.4945 | 3 ± 5 | [1, 7] | 0.0016 |
| GOLLuM | 79.77 ± 0.72 | [79.25, 80.28] | 0.3980 | 36.32 ± 4.02 | [3.45, 39.19] | 0.0000 | 13 ± 7 | [8, 18] | 0.0042 | 25 ± 12 | [17, 34] | 0.7381 |
| ChemBOMAS | 79.97 ± 0.50 | [79.62, 80.33] | - | 75.55 ± 0.00 | [5.55, 75.55] | - | 4 ± 5 | [0, 8] | - | 23 ± 13 | [4, 33] | - |
| **Buchwald_sub2** | | | | | | | | | | | | |
| BO | 56.61 ± 0.61 | [56.18, 57.05] | 0.3434 | 31.62 ± 13.55 | [21.92, 41.31] | 0.0016 | 20 ± 9 | [3, 27] | 0.0002 | 22 ± 9 | [16, 28] | 0.0000 |
| BO-ICL | 53.14 ± 1.93 | [51.12, 55.16] | 0.0055 | 46.12 ± 2.85 | [43.12, 49.11] | 0.0016 | 23 ± 12 | [1, 35] | 0.0069 | 24 ± 10 | [20, 40] | 0.0008 |
| LA-MCTS | 51.63 ± 3.40 | [49.19, 54.06] | 0.0010 | 48.25 ± 4.00 | [45.39, 51.11] | 0.0030 | 4 ± 7 | [0, 9] | 0.4723 | 6 ± 8 | [0, 12] | 0.1439 |
| GOLLuM | 54.99 ± 1.99 | [53.57, 56.41] | 0.0177 | 7.46 ± 3.08 | [5.25, 9.66] | 0.0000 | 24 ± 11 | [16, 31] | 0.0001 | 28 ± 12 | [19, 37] | 0.0001 |
| ChemBOMAS | 56.81 ± 0.00 | [56.81, 56.81] | - | 53.33 ± 0.00 | [53.34, 53.34] | - | 2 ± 0 | [2, 2] | - | 2 ± 0 | [2, 2] | - |

**Definition of Four Optimization Metrics.** Initial: The best objective value observed in the first "search", reflecting the quality of the initial design or warm start; Best Found: The final best objective value achieved by a method over all iterations, showing the optimization performance under the same evaluation budget; 95% Iter: The smallest iteration index for which the best-so-far value reaches at least 95% of the final best value, measuring the convergence speed in terms of the number of iterations needed to get "close enough" to the final optimum; Best Iter: The earliest iteration at which the final best value is first achieved, together with 95% Max Iteration, gives a more detailed view of the convergence trajectory.

**Final Performance and Statistical Significance.** ChemBOMAS consistently achieves the highest mean objective values across all four benchmark datasets: 96.15% (Suzuki), 82.83% (Arylation), 79.97% (Buchwald_sub1), and 56.81% (Buchwald_sub2). As shown in Table 7, our method demon-

strates statistically significant superiority ($p < 0.05$) against baselines such as LA-MCTS and GOLLuM across most tasks.

**Initialization and Cold-Start Capability.** A distinguishing feature of ChemBOMAS is its exceptional cold-start performance. The Initial column in Table 7 reveals that ChemBOMAS surpasses all baselines by a large margin in the very first iteration. For instance, in the Suzuki task, ChemBOMAS starts at 92.24%, whereas the strongest baseline GOLLuM starts at 79.10% and BO at only 58.91%.

**Convergence Efficiency and Stability.** ChemBOMAS exhibits the fastest convergence rates among all compared methods. The 95% Iter metric indicates that our method reaches near-optimal solutions within merely 1 to 4 iterations across all tasks. In contrast, baseline methods often require significantly more iterations.

## I  WET EXPERIMENTS

### I.1  WET EXPERIMENT DETAIL PROTOCOL

To validate the robustness of ChemBOMAS's initialization performance, the initial-round sampling was repeated ten times with the fixed experimental configurations. In the ten repeated initialization tests using ChemBOMAS, each run consistently identified at least two reaction conditions with yields exceeding 60%. Moreover, reaction conditions achieving yields above 80% appeared in 70% of the validation tests, totaling 11 such high-yield conditions across all trials. These results demonstrate that ChemBOMAS reliably mitigates the "cold-start" problem inherent to BO optimization.

**General Procedure for Reaction Optimization**  For the wet experiment involving palladium-catalyzed coupling of boronic esters with aryl chlorides, first, an oven-dried 10 mL Schlenk tube fitted with a Teflon-coated magnetic stir bar was charged inside an $N_2$-filled glovebox with Pd-catalyst (0.002 mmol), Phosphine ligand (0.008 mmol), and base (0.30 mmol, 1.5 equiv). Then, the tube was sealed with a septum, removed from the glovebox, and placed under a positive flow of $N_2$. The Mixture of organic solvent and water (2 mL) was introduced via a syringe. Next, pinacol boronic ester 2 (Reactant 1, 0.20 mmol, 1 equiv) and Aryl chloride 1 (Reactant 1, 0.25 mmol, 1.25 equiv) were added sequentially by syringe. The tube was capped tightly, placed in a pre-heated aluminum heating block maintained at 80 °C, 100 °C, or 120 °C, and the mixture was stirred (approximately 1500 rpm) for 24 hours. After cooling to room temperature, the mixture was diluted with ethyl acetate (3 mL) and quenched with water (3 mL). Finally, GC yields were determined directly from the crude mixture against the n-dodecane standard.

**ChemBOMAS Configuration**  Some configurations of ChemBOMAS described in the Experiment Section of the main text were adjusted for the wet experiment task. First, in the Knowledge Module, the additional process parameters (here, water usage and temperature) were divided into multiple subsets automatically by the LLM using RAG, and these subsets were grouped by the similarity of physical properties, which is the same as the category variables. For instance, temperature conditions were categorized into three distinct subsets corresponding to low, intermediate, and high activation energy levels. Moreover, during the Bayesian Optimization (BO), considering the relatively high experimental throughput, multiple acquisition functions (here, EI and UCB) were applied to generate fourteen samples per round. Apart from the aforementioned adjustments, all other configurations within ChemBOMAS remained consistent with those used in the dry-lab experiments.

**Sample in The Initial Round**  The initial experiment was only designed by Knowledge module due to the lack of prior data. Specifically, after the Knowledge Module partitioned the variables into subsets, a sampling function that can select variables from different subsets evenly was applied to generate fourteen diverse reaction conditions. The generated reaction conditions were then sent to the experiment operators for actual observation, which facilitated providing data to inform the experimental design in the next round.

**Sample in The Iterated Round**  As illustrated in Section B of the Supplementary Material, after receiving the observation feedback on each round of the wet experiment, all ChemBOMAS modules would update based on the feedback from each round of the wet-lab experiments. Following the

update of ChemBOMAS, the BO module would recommend fourteen reaction conditions with potentially higher yields for the subsequent round.

## J    ADDITIONAL RESULTS ANALYSIS

### J.1    DETAILED IMPLEMENTATION AND ROBUSTNESS ASSESSMENT OF THE KNOWLEDGE-DRIVEN MODULE

In this section, we provide an in-depth exposition of the knowledge-driven components within ChemBOMAS. We first detail the hierarchical information retrieval protocol that grounds the LLM's reasoning. Subsequently, we present concrete examples of the resulting subspace partitions and analyze the robustness of our framework against the inherent stochasticity of LLMs.

#### J.1.1    HYBRID RETRIEVAL-AUGMENTED GENERATION ARCHITECTURE

To ensure the LLM partitions the chemical search space based on grounded scientific principles rather than hallucinated correlations, we implemented a three-tier "Hybrid RAG" architecture. This prioritized pipeline orchestrates data retrieval from sources of varying structure and specificity:

- **Tier 1: Specialized Literature Repository.** The system first queries a curated local repository comprising peer-reviewed publications. Using reaction-specific keywords (e.g., "Suzuki Coupling mechanism," "ligand steric effects"), the retriever extracts the top-$k$ unstructured text passages. This tier prioritizes expert consensus on reaction mechanisms and reagent interactions.
- **Tier 2: Structured Chemical Databases.** If the unstructured text yields insufficient context for specific molecular properties, the pipeline queries structured databases, including RDKit and PubChem. By utilizing exact molecular identifiers (SMILES strings or IUPAC names), the system retrieves precise quantitative data, such as molecular fingerprints and physicochemical descriptors, to substantiate the clustering process.
- **Tier 3: Constrained Web Search.** As a final fallback mechanism, a web search API is employed to access encyclopedic or handbook-style chemical websites. This tier is strictly constrained to factual verification and obtaining short descriptions for less common reagents that may be absent from the local repository.

#### J.1.2    QUALITATIVE ANALYSIS OF SUBSPACE PARTITIONING

The effectiveness of the ChemBOMAS framework relies on the logical partitioning of the search space into chemically similar clusters. By grouping reagents based on the properties retrieved via the Hybrid RAG pipeline, the LLM constructs a subspace tree to guide the BO.

Table 8 illustrates the clustering outcomes for key variable categories—Ligands, Bases, and Solvents—across different independent runs. For instance, in the Ligand category, phosphine ligands are consistently grouped by steric and electronic characteristics (e.g., grouping bulky, electron-rich ligands like XPhos and SPhos), distinct from simple triphenylphosphine derivatives. These partitions effectively reduce the combinatorial complexity by allowing the UCB algorithm to prioritize subspaces with high-potential chemical properties.

#### J.1.3    ROBUSTNESS AGAINST LLM STOCHASTICITY

Since the Knowledge-driven module relies on Large Language Models (LLMs) for chemical space partitioning, the inherent stochasticity of LLM generation could potentially lead to variations in the search tree structure. To evaluate the robustness of our framework against these variations, we conducted four independent runs of the partitioning process, generating distinct subspace structures denoted as Subspace-1 through Subspace-4. We compared these against the primary reported partition Subspace across four datasets. The results, detailed in Table 9, provide compelling evidence for the stability of ChemBOMAS.

**Consistency in Optimization Performance.** The primary metric, "Best Found" performance, demonstrates remarkable stability. As shown in Table 9, the vast majority of variations yielded

Table 8: Summary of clustering results across different subspaces.

| Agent | Class | Subspace | Subspace-1 | Subspace-2 | Subspace-3 | Subspace-4 |
|---|---|---|---|---|---|---|
| Ligand | Class 1 | Xantphos
dtbpf
dppf | Xantphos
dtbpf
XPhos
dppf
SPhos
CataCXium A | Xantphos
dtbpf
XPhos
dppf | Xantphos
dtbpf
XPhos
dppf
SPhos | Xantphos
dtbpf
dppf |
| | Class 2 | XPhos
P(tBu)3
CataCXium A
P(Cy)3
AmPhos
SPhos | P(Ph)3
P(tBu)3
P(Cy)3
AmPhos
P(o-Tol)3 | P(Ph)3
P(Cy)3
P(o-Tol)3
SPhos
CataCXium A | P(Ph)3
P(Cy)3
AmPhos
P(o-Tol)3
CataCXium A | P(Ph)3
P(o-Tol)3
SPhos
CataCXium A
XPhos |
| | Class 3 | P(Ph)3
P(o-Tol)3
nothing | nothing | P(tBu)3
nothing
AmPhos | P(tBu)3
nothing | P(tBu)3
P(Cy)3
AmPhos
nothing |
| Base | Class 1 | KOH
NaOH
LiOtBu | KOH
NaOH
K3PO4
LiOtBu
NaHCO3
CsF | KOH
NaOH
K3PO4
LiOtBu
NaHCO3
CsF | KOH
NaOH | KOH
NaOH
K3PO4
LiOtBu
NaHCO3
CsF |
| | Class 2 | K3PO4
CsF
Et3N | Et3N | Et3N | Et3N
nothing
LiOtBu | Et3N
nothing |
| | Class 3 | NaHCO3
nothing | nothing | nothing | K3PO4
NaHCO3
CsF | / |
| Solvent | Class 1 | MeOH
MeOH/H2O_V2 9:1 | MeOH
MeOH/H2O_V2 9:1 | MeOH
MeOH/H2O_V2 9:1 | MeOH
MeOH/H2O_V2 9:1 | MeOH
MeOH/H2O_V2 9:1 |
| | Class 2 | DMF
MeCN | DMF
MeCN | DMF
MeCN | DMF
MeCN | DMF
MeCN |
| | Class 3 | THF
THF_V7 | THF
THF_V2 | THF
THF_V3 | THF
THF_V4 | THF
THF_V5 |

Table 9: Robust Performance of Subspace Partitioning Across Different Datasets. 95% Iter and Best Iter are rounded to the nearest integer.

| Method | Best Found | | | Initial | | | 95% Iter | | | Best Iter | | |
|---|---|---|---|---|---|---|---|---|---|---|---|---|
| | Mean ± Std | 95% CI | p-val | Mean ± Std | 95% CI | p-val | Mean ± Std | 95% CI | p-val | Mean ± Std | 95% CI | p-val |
| **Suzuki** | | | | | | | | | | | | |
| Subspace | 82.04 ± 4.49 | [78.83, 85.26] | - | 66.94 ± 8.02 | [61.21, 72.68] | - | 8 ± 11 | [0, 16] | - | 18 ± 14 | [8, 28] | - |
| Subspace-1 | 82.3 ± 5.14 | [78.63, 85.98] | > 0.05 | 60.9 ± 17.11 | [48.66, 73.14] | > 0.05 | 12 ± 13 | [2, 21] | > 0.05 | 21 ± 12 | [12, 30] | > 0.05 |
| Subspace-2 | 84.74 ± 7.87 | [79.11, 90.37] | > 0.05 | 64.46 ± 8.83 | [58.14, 70.77] | > 0.05 | 15 ± 13 | [5, 24] | > 0.05 | 26 ± 12 | [18, 34] | > 0.05 |
| Subspace-3 | 94.92 ± 1.98 | [93.51, 96.34] | 0.0000 | 60.38 ± 15.03 | [49.63, 71.13] | > 0.05 | 13 ± 11 | [5, 21] | > 0.05 | 15 ± 11 | [6, 23] | > 0.05 |
| Subspace-4 | 85.83 ± 7.61 | [80.39, 91.27] | > 0.05 | 58.09 ± 16.78 | [46.08, 70.10] | > 0.05 | 13 ± 14 | [3, 23] | > 0.05 | 20 ± 14 | [10, 30] | > 0.05 |
| **Arylation** | | | | | | | | | | | | |
| Subspace | 81.28 ± 2.12 | [79.76, 82.80] | - | 59.38 ± 17.79 | [46.66, 72.11] | - | 11 ± 13 | [2, 21] | - | 20 ± 14 | [10, 30] | - |
| Subspace-1 | 81.4 ± 2.14 | [79.87, 82.93] | > 0.05 | 56.87 ± 20.19 | [42.13, 71.02] | > 0.05 | 10 ± 9 | [3, 17] | > 0.05 | 28 ± 9 | [22, 35] | > 0.05 |
| Subspace-2 | 81.4 ± 2.14 | [79.87, 82.93] | > 0.05 | 56.87 ± 20.19 | [42.13, 71.02] | > 0.05 | 10 ± 9 | [3, 17] | > 0.05 | 28 ± 9 | [22, 35] | > 0.05 |
| Subspace-3 | 81.25 ± 2.03 | [79.80, 82.70] | > 0.05 | 60.58 ± 13.88 | [50.65, 70.51] | > 0.05 | 14 ± 9 | [8, 20] | > 0.05 | 28 ± 9 | [21, 34] | > 0.05 |
| Subspace-4 | 82.14 ± 2.14 | [80.61, 83.67] | > 0.05 | 54.12 ± 19.98 | [39.83, 68.42] | > 0.05 | 15 ± 12 | [6, 23] | > 0.05 | 28 ± 8 | [22, 34] | > 0.05 |
| **Buchwald_sub1** | | | | | | | | | | | | |
| Subspace | 80.25 ± 2.22 | [78.66, 81.83] | - | 44.08 ± 26.71 | [24.98, 63.19] | - | 4 ± 2 | [2, 6] | - | 11 ± 7 | [5, 16] | - |
| Subspace-1 | 79.01 ± 1.31 | [78.08, 79.95] | > 0.05 | 50.39 ± 17.4 | [37.94, 62.83] | > 0.05 | 9 ± 8 | [4, 15] | 0.0383 | 27 ± 13 | [18, 37] | 0.0025 |
| Subspace-2 | 79.52 ± 0.39 | [79.24, 79.80] | > 0.05 | 41.35 ± 22.53 | [25.24, 57.47] | > 0.05 | 9 ± 9 | [3, 15] | > 0.05 | 33 ± 13 | [23, 42] | 0.0033 |
| Subspace-3 | 79.01 ± 1.31 | [78.08, 79.95] | > 0.05 | 50.39 ± 17.4 | [37.94, 62.83] | > 0.05 | 9 ± 8 | [4, 15] | 0.0383 | 27 ± 13 | [18, 37] | 0.0025 |
| Subspace-4 | 79.01 ± 1.31 | [78.08, 79.95] | > 0.05 | 50.39 ± 17.4 | [37.94, 62.83] | > 0.05 | 9 ± 8 | [4, 15] | 0.0383 | 27 ± 13 | [18, 37] | 0.0025 |
| **Buchwald_sub2** | | | | | | | | | | | | |
| Subspace | 53.23 ± 0.38 | [52.95, 53.50] | - | 18.19 ± 22.47 | [2.12, 34.26] | - | 18 ± 9 | [12, 25] | - | 26 ± 7 | [22, 31] | - |
| Subspace-1 | 53.72 ± 0.89 | [53.08, 54.36] | > 0.05 | 16.56 ± 17.42 | [4.10, 29.02] | > 0.05 | 22 ± 7 | [17, 27] | > 0.05 | 26 ± 9 | [20, 33] | > 0.05 |
| Subspace-2 | 52.67 ± 1.89 | [51.32, 54.03] | > 0.05 | 22.52 ± 20.2 | [8.08, 36.97] | > 0.05 | 14 ± 10 | [6, 21] | > 0.05 | 18 ± 12 | [10, 26] | 0.0402 |
| Subspace-3 | 51.62 ± 1.47 | [50.57, 52.67] | 0.0079 | 19.12 ± 19.96 | [4.85, 33.40] | > 0.05 | 7 ± 6 | [3, 11] | 0.0120 | 28 ± 10 | [21, 35] | > 0.05 |
| Subspace-4 | 51.62 ± 1.47 | [50.57, 52.67] | 0.0079 | 19.12 ± 19.96 | [4.85, 33.40] | > 0.05 | 7 ± 6 | [3, 11] | 0.0120 | 28 ± 10 | [21, 35] | > 0.05 |

results that are statistically indistinguishable from the baseline ($p$-value $> 0.05$). For example, in the Arylation dataset, all four variants achieved mean yields between $81.25\%$ and $82.14\%$, with no significant deviation from the baseline ($81.28\%$). This indicates that while the specific topological structure of the search tree may vary due to LLM ranking and clustering differences, the framework consistently identifies high-potential regions that contain the global or near-global optima.

**Variance as Exploration Opportunity.** In the rare instances where statistically significant differences were observed, the deviations often favored improved performance. Notably, in the Suzuki dataset, Subspace-3 achieved a significantly higher mean yield of $94.92\%$ compared to the baseline's $82.04\%$ ($p < 0.001$). This suggests that the stochastic nature of the LLM-guided partitioning can occasionally serve as a beneficial exploration mechanism, uncovering superior subspace configurations without catastrophic failure modes. Even in the worst-case scenario in Buchwald_sub2 of Subspace-4, the performance drop was marginal ($< 1.7\%$), further confirming the method's resilience.

**Efficiency Stability.** While the convergence metrics exhibit naturally higher variance due to the differing depths and branching factors of the generated trees, the optimization process remains efficient. The "Initial" values across all variants are comparable, ensuring that the BO process starts from a robust baseline regardless of the specific partition.

## J.2 Validation of Pseudo-data Efficacy across Data Volumn

To rigorously evaluate the data efficiency of the ChemBOMAS framework and determine the minimal supervision required for robust optimization, we conducted a sensitivity analysis regarding the volume of labeled data utilized during the Supervised Fine-Tuning (SFT) phase. We varied the size of the labeled dataset $\mathcal{D}_{\text{labeled}}$ from a scarce regime of $0.00\%$ to a data-rich regime of $32.00\%$ across four distinct chemical reaction datasets. The impact of data volume was assessed through two lenses: the predictive accuracy of the LLM regressor measured by MSE, MAE and $R^2$ and the downstream efficacy of the Bayesian Optimization measured by the best yield found and convergence speed. The comprehensive results are presented in Table 10.

Our analysis reveals a distinct non-linear relationship between data availability and optimization performance. Initially, we observe a critical performance threshold below which the framework fails to gain traction. In regimes where the labeled data constitutes less than $0.50\%$ of the total pool, the regression metrics indicate a failure to learn meaningful representations, evidenced by negative $R^2$ values across most datasets. Specifically, at the $0.02\%$ and $0.10\%$ levels, the LLM-generated pseudo-data exhibits high noise, leading to UCB initializations that are often comparable to, or marginally better than, random baselines. For instance, in the Suzuki dataset, the $0.25\%$ setting results in a negative $R^2$ of -0.53 and a best-found yield of $81.17\%$, significantly underperforming compared to settings with adequate supervision. This suggests that insufficient few-shot examples prevent the LLM from aligning its pre-trained chemical knowledge with the specific response landscape of the target reaction, thereby degrading the guidance provided to the BO module.

Conversely, the results demonstrate a performance saturation, beyond which increasing data volume yields diminishing returns for the optimization objective. As the data volume increases from $2.00\%$ to $32.00\%$, the regression accuracy improves monotonically, with MSE decreasing and $R^2$ approaching 0.96 in the Buchwald cases. However, this increase in predictive precision does not translate linearly into improved BO outcomes. The "Best Found" yields effectively plateau once the data volume surpasses the $1.00\%$ to $2.00\%$ range. For example, in the Buchwald_sub1 dataset, quadrupling the data from $1.00\%$ to $4.00\%$ improves the $R^2$ from 0.09 to 0.61, yet the best yield found improves only marginally from $79.97\%$ to $80.45\%$. This phenomenon indicates that while higher data volumes refine the surrogate model's global fidelity, the coarse-grained topology provided by the LLM at moderate data levels is sufficiently accurate to identify high-potential subspaces for the UCB algorithm.

Based on these observations, we selected $1.00\%$ as the optimal data volume for the ChemBOMAS framework. This setting represents a strategic equilibrium, situated immediately past the inflection point of the lower bound where the model begins to demonstrate positive $R^2$ values and reliable ranking capabilities. At $1.00\%$, the framework achieves near-optimal optimization results—matching the peak performance of data-rich settings in datasets like Suzuki ($96.15\%$) and Buchwald_sub2 ($56.81\%$)—while requiring a minimal experimental budget. This decision aligns with the core objective of Bayesian Optimization in chemistry: to maximize reaction yield with the fewest possible wet-lab experiments. By leveraging just $1.00\%$ of labeled data, ChemBOMAS effectively activates

Table 10: Summary of Pseudo Data Analysis and BO Results. The table compares different SFT settings across four datasets. 95% Iter and Best Iter are rounded to the nearest integer.

| Variant | Data % | Num | Regression Metrics | | | BO Result (Mean ± Std) | | | |
|---|---|---|---|---|---|---|---|---|---|
| | | | MSE | MAE | $R^2$ | Best Found | Initial | 95% Iter | Best Iter |
| **Suzuki** | | | | | | | | | |
| Random Pseudo | / | / | 1734.34 | 33.97 | -1.20 | 80.98 ± 3.94 | 73.12 ± 0.00 | 14 ± 9 | 24 ± 12 |
| No SFT | 0 | 0 | 2403.41 | 40.18 | -2.05 | 88.18 ± 8.41 | 44.00 ± 0.00 | 17 ± 16 | 31 ± 8 |
| SFT 0.02 | 0.02% | 1 | 1205.49 | 26.04 | -0.53 | 87.41 ± 8.45 | 21.55 ± 0.00 | 20 ± 15 | 27 ± 11 |
| SFT 0.1 | 0.10% | 6 | 1332.07 | 27.51 | -0.69 | 85.38 ± 6.41 | 37.34 ± 0.00 | 14 ± 11 | 25 ± 9 |
| SFT 0.25 | 0.25% | 14 | 1205.19 | 27.80 | -0.53 | 81.17 ± 3.84 | 76.01 ± 0.00 | 5 ± 12 | 14 ± 13 |
| SFT 0.5 | 0.50% | 29 | 774.70 | 21.13 | 0.02 | 92.06 ± 0.00 | 92.06 ± 0.00 | 1 ± 0 | 1 ± 0 |
| SFT 1.0 | 1.00% | 50 | 633.68 | 19.47 | 0.20 | 96.15 ± 0.00 | 92.24 ± 0.00 | 1 ± 0 | 3 ± 0 |
| SFT 2.0 | 2.00% | 115 | 479.09 | 15.92 | 0.39 | 93.41 ± 1.89 | 92.24 ± 0.00 | 1 ± 0 | 13 ± 19 |
| SFT 4.0 | 4.00% | 230 | 360.02 | 13.44 | 0.54 | 92.24 ± 0.00 | 92.24 ± 0.00 | 1 ± 0 | 1 ± 0 |
| SFT 8.0 | 8.00% | 461 | 252.85 | 10.77 | 0.68 | 91.89 ± 0.52 | 74.96 ± 0.00 | 2 ± 0 | 24 ± 10 |
| SFT 16.0 | 16.00% | 922 | 163.48 | 8.23 | 0.79 | 92.24 ± 0.00 | 88.80 ± 0.00 | 1 ± 0 | 2 ± 0 |
| SFT 32.0 | 32.00% | 1844 | 89.84 | 5.46 | 0.89 | 96.15 ± 0.00 | 88.80 ± 0.00 | 2 ± 0 | 2 ± 0 |
| **Arylation** | | | | | | | | | |
| Random Pseudo | / | / | 1849.86 | 35.18 | -1.48 | 79.56 ± 0.42 | 35.70 ± 0.00 | 2 ± 0 | 21 ± 9 |
| No SFT | 0 | 0 | 1853.70 | 33.24 | -1.49 | 82.20 ± 1.38 | 65.86 ± 0.00 | 11 ± 6 | 22 ± 14 |
| SFT 0.02 | 0.02% | 1 | 885.06 | 25.33 | -0.19 | 80.58 ± 2.09 | 0.00 ± 0.00 | 10 ± 6 | 17 ± 9 |
| SFT 0.1 | 0.10% | 4 | 1330.97 | 31.57 | -0.79 | 81.50 ± 2.35 | 32.41 ± 0.00 | 16 ± 11 | 29 ± 9 |
| SFT 0.25 | 0.25% | 10 | 800.09 | 24.22 | -0.07 | 80.94 ± 1.65 | 28.33 ± 0.00 | 3 ± 1 | 21 ± 12 |
| SFT 0.5 | 0.50% | 20 | 1016.37 | 26.53 | -0.36 | 81.70 ± 1.41 | 76.34 ± 0.00 | 2 ± 0 | 9 ± 9 |
| SFT 1.0 | 1.00% | 34 | 650.00 | 19.55 | 0.13 | 82.83 ± 0.64 | 82.63 ± 0.00 | 1 ± 0 | 4 ± 10 |
| SFT 2.0 | 2.00% | 79 | 462.52 | 15.75 | 0.38 | 82.98 ± 0.53 | 82.57 ± 0.00 | 1 ± 0 | 13 ± 16 |
| SFT 4.0 | 4.00% | 158 | 286.56 | 11.97 | 0.62 | 83.60 ± 0.00 | 76.95 ± 0.00 | 2 ± 0 | 14 ± 9 |
| SFT 8.0 | 8.00% | 316 | 170.07 | 8.49 | 0.77 | 83.60 ± 0.00 | 82.57 ± 0.00 | 1 ± 0 | 13 ± 3 |
| SFT 16.0 | 16.00% | 633 | 110.42 | 6.39 | 0.85 | 83.29 ± 0.70 | 77.47 ± 0.00 | 4 ± 5 | 26 ± 13 |
| SFT 32.0 | 32.00% | 1266 | 38.14 | 3.65 | 0.95 | 83.60 ± 0.00 | 82.57 ± 0.00 | 1 ± 0 | 11 ± 3 |
| **Buchwald_sub1** | | | | | | | | | |
| Random Pseudo | / | / | 1608.17 | 33.08 | -1.14 | 79.57 ± 0.20 | 32.17 ± 0.00 | 5 ± 2 | 26 ± 12 |
| No SFT | 0 | 0 | 3486.42 | 52.28 | -3.64 | 79.83 ± 0.38 | 77.63 ± 0.00 | 1 ± 0 | 27 ± 11 |
| SFT 0.02 | 0.02% | 1 | 3159.89 | 49.18 | -3.20 | 79.93 ± 0.53 | 37.12 ± 0.00 | 2 ± 0 | 16 ± 6 |
| SFT 0.1 | 0.10% | 4 | 3019.90 | 47.77 | -3.02 | 80.11 ± 0.55 | 66.43 ± 0.00 | 2 ± 0 | 13 ± 13 |
| SFT 0.25 | 0.25% | 10 | 701.56 | 22.90 | 0.07 | 79.60 ± 0.28 | 75.55 ± 0.00 | 4 ± 2 | 22 ± 12 |
| SFT 0.5 | 0.50% | 20 | 750.86 | 22.72 | 0.00 | 79.91 ± 0.53 | 28.40 ± 0.00 | 2 ± 0 | 27 ± 8 |
| SFT 1.0 | 1.00% | 34 | 680.99 | 21.65 | 0.09 | 79.97 ± 0.12 | 75.55 ± 0.00 | 4 ± 1 | 23 ± 13 |
| SFT 2.0 | 2.00% | 79 | 448.16 | 16.90 | 0.40 | 79.73 ± 0.08 | 65.59 ± 0.00 | 2 ± 0 | 22 ± 13 |
| SFT 4.0 | 4.00% | 158 | 291.79 | 13.16 | 0.61 | 80.45 ± 0.59 | 79.53 ± 0.00 | 3 ± 0 | 10 ± 7 |
| SFT 8.0 | 8.00% | 316 | 165.57 | 8.99 | 0.78 | 80.68 ± 0.48 | 79.08 ± 0.00 | 1 ± 0 | 13 ± 9 |
| SFT 16.0 | 16.00% | 633 | 90.34 | 6.40 | 0.88 | 80.11 ± 0.55 | 79.08 ± 0.00 | 3 ± 0 | 8 ± 5 |
| SFT 32.0 | 32.00% | 1266 | 33.33 | 3.59 | 0.96 | 80.44 ± 0.60 | 79.53 ± 0.00 | 3 ± 0 | 20 ± 8 |
| **Buchwald_sub2** | | | | | | | | | |
| Random Pseudo | / | / | 2111.08 | 37.88 | -4.21 | 51.74 ± 0.79 | 12.94 ± 0.00 | 13 ± 3 | 18 ± 8 |
| No SFT | 0 | 0 | 809.26 | 20.16 | -1.00 | 53.69 ± 2.19 | 46.94 ± 0.00 | 16 ± 12 | 23 ± 15 |
| SFT 0.02 | 0.02% | 1 | 716.50 | 19.00 | -0.77 | 52.52 ± 2.35 | 0.00 ± 0.00 | 6 ± 3 | 14 ± 12 |
| SFT 0.1 | 0.10% | 4 | 670.44 | 18.79 | -0.66 | 55.57 ± 2.24 | 14.90 ± 0.00 | 33 ± 13 | 36 ± 9 |
| SFT 0.25 | 0.25% | 10 | 714.63 | 23.07 | -0.76 | 52.77 ± 1.96 | 35.26 ± 0.00 | 5 ± 2 | 14 ± 12 |
| SFT 0.5 | 0.50% | 20 | 408.47 | 16.02 | -0.01 | 53.34 ± 0.77 | 43.09 ± 0.00 | 11 ± 11 | 28 ± 8 |
| SFT 1.0 | 1.00% | 34 | 247.70 | 12.15 | 0.39 | 56.81 ± 0.00 | 53.33 ± 0.00 | 2 ± 0 | 2 ± 0 |
| SFT 2.0 | 2.00% | 79 | 195.03 | 9.76 | 0.51 | 53.94 ± 1.11 | 53.10 ± 0.00 | 5 ± 12 | 18 ± 14 |
| SFT 4.0 | 4.00% | 158 | 178.88 | 8.70 | 0.56 | 53.83 ± 2.10 | 52.01 ± 0.00 | 10 ± 15 | 18 ± 14 |
| SFT 8.0 | 8.00% | 316 | 75.81 | 5.26 | 0.81 | 55.21 ± 1.70 | 50.21 ± 0.00 | 8 ± 7 | 17 ± 11 |
| SFT 16.0 | 16.00% | 633 | 59.02 | 4.06 | 0.85 | 54.78 ± 1.47 | 53.33 ± 0.00 | 8 ± 13 | 15 ± 15 |
| SFT 32.0 | 32.00% | 1266 | 33.32 | 2.61 | 0.92 | 54.72 ± 1.16 | 53.33 ± 0.00 | 2 ± 2 | 10 ± 11 |

the latent knowledge of the LLM to guide the search, avoiding the prohibitive costs associated with collecting larger datasets required for traditional supervised learning saturation.

## J.3 Robustness Analysis of Acquisition Functions

To determine the optimal configuration for the Bayesian Optimization component within Chem-BOMAS, we conducted a comparative analysis of four standard acquisition functions: Expected Improvement (EI), Minimum Variance Estimation (MVE), Probability of Improvement (PI), and Upper Confidence Bound (UCB). To isolate the specific impact of the acquisition strategy from the proposed LLM-enhanced modules, these experiments were performed using a traditional BO framework across the four benchmark datasets.

Table 11: Comparison of different acquisition functions across four chemical reaction datasets. The datasets are arranged in a $2 \times 2$ grid for compact comparison.

| Acq. | Suzuki | | | | | | Arylation | | | | | |
|---|---|---|---|---|---|---|---|---|---|---|---|---|
| | Best Found | | 95% Iter | | Best Iter | | Best Found | | 95% Iter | | Best Iter | |
| | Mean ± Std | $p$-val | Mean ± Std | $p$-val | Mean ± Std | $p$-val | Mean ± Std | $p$-val | Mean ± Std | $p$-val | Mean ± Std | $p$-val |
| EI | 91.45 ± 7.58 | - | 12 ± 10 | - | 16 ± 7 | - | 82.83 ± 1.77 | - | 12 ± 9 | - | 27 ± 8 | - |
| MVE | 91.35 ± 7.73 | > 0.05 | 15 ± 13 | > 0.05 | 24 ± 15 | > 0.05 | 82.40 ± 2.33 | > 0.05 | 14 ± 12 | > 0.05 | 30 ± 9 | > 0.05 |
| PI | 92.99 ± 6.67 | > 0.05 | 16 ± 9 | > 0.05 | 21 ± 11 | > 0.05 | 82.99 ± 2.02 | > 0.05 | 16 ± 14 | > 0.05 | 26 ± 11 | > 0.05 |
| UCB | 94.61 ± 4.88 | > 0.05 | 16 ± 11 | > 0.05 | 16 ± 11 | > 0.05 | 83.55 ± 2.06 | > 0.05 | 21 ± 11 | > 0.05 | 28 ± 5 | > 0.05 |

| Acq. | Buchwald_sub1 | | | | | | Buchwald_sub2 | | | | | |
|---|---|---|---|---|---|---|---|---|---|---|---|---|
| | Best Found | | 95% Iter | | Best Iter | | Best Found | | 95% Iter | | Best Iter | |
| | Mean ± Std | $p$-val | Mean ± Std | $p$-val | Mean ± Std | $p$-val | Mean ± Std | $p$-val | Mean ± Std | $p$-val | Mean ± Std | $p$-val |
| EI | 79.74 ± 0.42 | - | 5 ± 2 | - | 12 ± 10 | - | 56.61 ± 0.61 | - | 20 ± 9 | - | 22 ± 9 | - |
| MVE | 79.39 ± 1.35 | > 0.05 | 4 ± 3 | > 0.05 | 12 ± 9 | > 0.05 | 56.81 ± 0.00 | > 0.05 | 16 ± 6 | > 0.05 | 25 ± 4 | > 0.05 |
| PI | 79.78 ± 0.41 | > 0.05 | 5 ± 3 | > 0.05 | 12 ± 13 | > 0.05 | 55.77 ± 1.66 | > 0.05 | 26 ± 11 | > 0.05 | 29 ± 9 | > 0.05 |
| UCB | 79.83 ± 0.39 | > 0.05 | 6 ± 4 | > 0.05 | 17 ± 16 | > 0.05 | 54.07 ± 0.58 | > 0.05 | 13 ± 8 | > 0.05 | 22 ± 11 | > 0.05 |

The experimental results are summarized in Table **??**. We observe distinct performance characteristics across the different strategies:

- **Performance Consistency:** While UCB achieves marginally higher mean "Best Found" values in the *Suzuki* (94.61±4.88) and *Arylation* (83.55±2.06) datasets, it exhibits instability in more complex landscapes. Notably, in the *Buchwald_sub2* dataset, UCB yields the lowest performance ($54.07 \pm 0.58$), whereas EI maintains robust performance ($56.61 \pm 0.61$), comparable to the top-performing MVE method.

- **Statistical Significance:** Crucially, the statistical analysis reveals that the performance differences between EI and the other methods are generally not statistically significant ($p > 0.05$) across most metrics and datasets. This suggests that while UCB may offer aggressive exploration benefits in specific contexts, it does not consistently outperform EI.

- **Convergence Efficiency:** In terms of convergence speed ("95% Iter"), EI demonstrates high efficiency. For instance, in the *Arylation* dataset, EI requires an average of 12 iterations to reach 95% of the optimum, compared to 21 iterations for UCB. This efficiency is critical for chemical optimization tasks where experimental evaluations are costly.

Given that EI provides a parameter-free mechanism that effectively balances exploration and exploitation while maintaining consistent performance across diverse chemical spaces, we adopt **Expected Improvement** as the default acquisition function for the proposed ChemBOMAS framework.

## K Generalization to Broader Scientific Domains

To assess the cross-domain universality of the ChemBOMAS framework, we extended our evaluation to a materials science benchmark. This expansion serves to validate a core hypothesis: that the fundamental principle of combining knowledge-driven decomposition with data-driven fine-tuning is transmissible beyond chemistry to generic complex black-box optimization problems. Accordingly, we selected the LNP3 dataset, which presents unique challenges in scientific discovery.

### K.1 DATASET OVERVIEW AND PROBLEM CONTEXT

The LNP3 dataset originates from the field of nanomedicine, specifically addressing the optimization of lipid nanoparticle (LNP) formulations for the effective delivery of cannabidiol **?**. The original experimental campaign encompassed 768 unique formulations defined by a 5-dimensional parameter space, including the type and quantity of solid lipids, liquid lipids, and surfactants.

While the formulation of LNPs is inherently a multi-objective problem—aiming to simultaneously maximize drug loading and encapsulation efficiency while minimizing particle size—this study isolates the specific challenge of maximizing Encapsulation Efficiency (EE). This creates a complex single-objective optimization task constrained by a discrete, categorical search space that encapsulates the non-trivial trade-offs found in real-world material design.

### K.2 TASK DEFINITION AND EXPERIMENTAL SETUP

We formulate the LNP3 challenge as a static, offline black-box optimization task.

- **Objective:** Maximize the raw, non-normalized Encapsulation Efficiency.
- **Constraint:** The search is restricted to the predefined discrete experimental grid.
- **Data Source:** The dataset is accessible via the Olympus benchmark suite[1].

The parameter space $\mathcal{X}$ is constructed from five categorical variables with explicitly defined levels:

1. **Drug Input:** Dosage levels of $\{6, 12, 24, 48\}$ mg.
2. **Solid Lipid Type:** Categorical selection from $\{$Stearic Acid, Compritol 888, Glyceryl Monostearate$\}$.
3. **Solid Lipid Quantity:** Amount levels of $\{72, 96, 108, 120\}$ mg.
4. **Liquid Lipid Input:** Amount levels of $\{0, 12, 24, 48\}$ mg.
5. **Surfactant Concentration:** Weight-to-weight ratios of $\{0.0, 0.0025, 0.005, 0.01\}$.

#### K.2.1 STATISTICAL DISTRIBUTION

The optimization target, Encapsulation Efficiency, exhibits significant variation across the design space, as summarized in Table 12. To further visualize the landscape difficulty, Figure 6 presents the global distribution of the objective values.

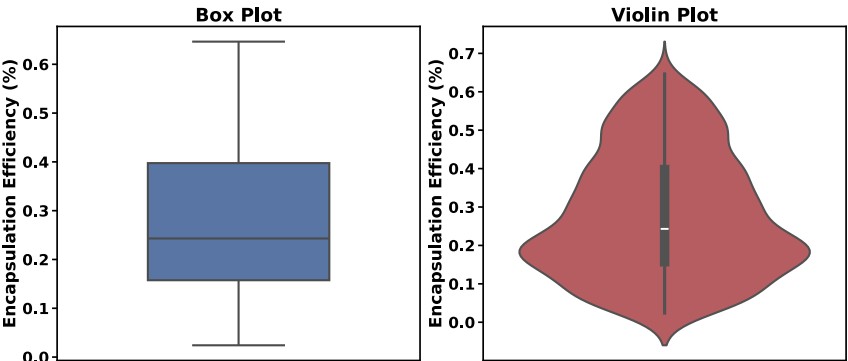

Figure 6: **Distribution of the optimization objective across the LNP3 dataset.** The figure employs a hybrid visualization using box plots and violin plots to characterize the target variable. The vertical axis represents the raw Encapsulation Efficiency.

The performance comparison between ChemBOMAS and baseline methods on this dataset is summarized in Table 13.

---

[1] https://github.com/aspuru-guzik-group/olympus

Table 12: Statistical Summary of the LNP3 Benchmark. The target variable is the raw Encapsulation Efficiency, exhibiting a broad dynamic range.

| Statistical Metric | Value |
|---|---|
| Total Data Points ($N$) | 768 |
| Maximum | 0.6464 |
| Minimum | 0.0241 |
| Mean | 0.28 |
| Median | 0.24 |
| Standard Deviation | 0.16 |
| 25th Percentile | 0.16 |
| 75th Percentile | 0.40 |

Table 13: Performance Comparison on a Non-Chemical Scientific Benchmark.

| Dataset | Method | Best Found | Initial Value | 95% Max Iter↓ | Iteration of Best↓ |
|---|---|---|---|---|---|
| | ChemBOMAS | **0.62** | 0.23 | 12 | 28 |
| | Gollum | **0.62** | 0.21 | 13 | 33 |
| LNP3 | BO | **0.62** | 0.25 | 12 | 38 |
| | LA-MCTS | 0.47 | **0.44** | **4** | **15** |
| | BO-ICL | 0.60 | 0.15 | 24 | 38 |

In the LNP3 material formulation benchmark, ChemBOMAS demonstrated highly competitive performance. It successfully identified an optimal value of 0.62, matching the final performance achieved by GoLLuM and traditional Bayesian optimization (BO). More importantly, ChemBOMAS demonstrated higher sample efficiency, locating this optimal solution in just 28 iterations, compared to 33 for GoLLuM and 38 for BO. This result indicates that the framework's structured exploration mechanism can effectively accelerate convergence even in non-chemical optimization scenarios. Notably, while LA-MCTS delivered strong initial performance, it prematurely converged to a suboptimal solution, highlighting the risks of overly aggressive early exploration.

Overall, testing results in the field of materials science demonstrate that ChemBOMAS's fundamental architecture—namely, the synergistic integration of knowledge-based search space partitioning with data-driven model optimization—holds potential as a universal strategy. It has proven that beyond core chemical domains, this framework possesses equally robust applicability and competitiveness in accelerating black-box optimization across diverse scientific discovery tasks.

