# OpenReview forum: "ChemBOMAS: Accelerated Bayesian Optimization for Scientific Discovery in Chemistry with LLM-Enhanced Multi-Agent System"
_ICLR.cc/2026/Conference — Submitted to ICLR 2026_

### Official Review · Reviewer_GLbf · 2025-10-30

**Soundness:** 2
**Presentation:** 1
**Contribution:** 2
**Rating:** 2
**Confidence:** 4

**Summary:**

The paper introduces ChemBOMAS, a novel framework integrating large language models (LLMs) into Bayesian optimization (BO) for chemical reaction optimization tasks. The approach tries to address the challenges of data scarcity and inefficiency in vast search spaces. ChemBOMAS employs a knowledge-driven module to decompose the search space and a data-driven module for generating pseudo-data to enhance optimization performance. The framework is evaluated through extensive experiments indicating improvements in yield and speed compared to several baselines.

**Strengths:**

1. Their framework shows integrating LLMs improves BO for chemical optimization
2. The framework was validated through wet-lab experiments

**Weaknesses:**

1. Presentation and Clarity:
  *  The metrics (like best found, object value max Iter, and Initial) used in the paper are not well explained, neither in the main paper, nor in the Appendix. So it's hard for the reviewers to identify their significance when comparing BO performance.
  * RAG is introduced in the knowledge-driven strategy. How is it implemented, and how does it affect the results?
  * The Notations are confusing. In Line 165, $x_t$ represents the $t$-th token, while in line 169, $x$ is the reaction configuration that contains $R, P,c$.

2. Rigour of Experiments and Conclusion:
  * As the proposed method has introduced data augmentation and different data initialization conditions for different datasets, it's unclear whether the proposed method is fairly compared to the baselines. Especially in Figure 2, the improvement is more obvious on Suzuki while not too significant for other datasets. If we use the same percentage of datasets as the initial points, as Suzuki has more data points, then the use of it for pretrain and SFT will be more stable.
  * In Table 4, the experimental results are reported without variance; the knowledge module and data module do not always have a positive impact on the results, where the best iteration for Arylation is obtained when neither of these modules is imported.
  * All the results are reported based on five random seeds; this is not reliable for heuristic approaches.
3. The model's performance relies heavily on fine-tuning data volume, suggesting that substantial effort is needed to optimize the model for specific datasets.
4. The integration of knowledge-driven and data-driven modules seem complex and convoluted

**Questions:**

1. How does ChemBOMAS handle scalability issues, especially in scenarios involving extremely large and complex chemical spaces?
2. What's the cost for the knowledge-driven process that combines RAG and general LLMs? Did you query for space partition at every BO iteration? If so, it should be quite expensive.
3. What mechanisms are in place to adapt ChemBOMAS effectively under varying initial conditions concerning data volume?

---

> ### Author Response · Authors · 2025-12-03
> **Reply to Reviewer GLbf - Part 1/8**
>
> Thank you for your constructive feedback. In light of your comments, we have carefully revised the manuscript, adding new experiments, results, and analyses where appropriate. We have provided a point-by-point response to each of your concerns below, and we hope that these revisions and replies satisfactorily address your reservations.
>
> **Q1: The metrics (like best found, object value max Iter, and Initial) used in the paper are not well explained, neither in the main paper, nor in the Appendix. So it's hard for the reviewers to identify their significance when comparing BO performance.**
>
> **A1:** We appreciate this comment for improving the readability of our paper. Below, we provide precise definitions of the metrics used to evaluate BO, and explain why each of the metrics is important for interpreting optimization performance.
> 1. Initial: The best objective value observed in the first "search", reflecting the quality of the initial design or warm start.
> 2. Best Found: The final best objective value achieved by a method over all iterations, showing the optimization performance under the same evaluation budget.
> 3. 95% Iter: The smallest iteration index for which the best-so-far value reaches at least 95% of the final best value, measuring the convergence speed in terms of the number of iterations needed to get “close enough” to the final optimum.
> 4. Best Iter: The earliest iteration at which the final best value is first achieved, together with 95% Max Iteration, gives a more detailed view of the convergence trajectory.
>
> We have added a corresponding description of these metrics in Appendix H.
>
> **Q2: The Notations are confusing. In Line 165, $x_t$ represents the $t$-th token, while in line 169, $x$ is the reaction configuration that contains $R, P,c$.**
>
> **A2:** Thank you for pointing out that the symbol $x$ is overloaded in two different contexts. In the revised manuscript, we denote a token sequence by $s=(w_1, ..., w_t)$, in which $w_t$ represents the $t$-th token, and reserve $x$ exclusively for reaction configurations.
>
> Moreover, we systematically check and unify all notation throughout the paper and the appendix to avoid any similar overloading or ambiguity. We appreciate you again for improving the readability of the manuscript, and we hope these changes resolve the confusion.
>
> **Q3: As the proposed method has introduced data augmentation and different data initialization conditions for different datasets, it's unclear whether the proposed method is fairly compared to the baselines.**
>
> **A3:** We are grateful to you for pointing out the ambiguity of fairness in the baseline comparison. Here, we clarify:
>
> **All methods indeed start from a strictly equivalent initial state**, having access to the same prior dataset (the 1% labeled data used for LLM fine-tuning) for a fair comparison. The key point is that "Iteration 0" in Figure 2 does not represent this shared starting point, but rather the first search step after each algorithm has processed the identical initial data. We displayed the shared prior data to demonstrate fairness **in the revised Figure 2**.

---

> ### Author Response · Authors · 2025-12-03
> **Reply to Reviewer GLbf - Part 2/8**
>
> **Q4: RAG is introduced in the knowledge-driven strategy. How is it implemented, and how does it affect the results?**
>
> **A4:** We appreciate the opportunity to clarify the implementation of our "Hybrid RAG" architecture. It operates as a three-tier, prioritized pipeline to retrieve comprehensive chemical knowledge.
>
> 1. **Tier 1: Literature Extraction (High Priority):** The process begins by querying a curated local repository of academic literature using chemical reaction-type keywords to retrieve top-k passages of unstructured text that encapsulate expert knowledge on mechanisms and properties.
> 2. **Tier 2: Database Search (Intermediate Priority):** If literature extraction yields insufficient context, the RAG pipeline will supplement the chemical information by querying structured databases, such as RDKit and PubChem, using exact identifiers (SMILES or IUPAC names). These domain databases store diverse attributes, such as molecular fingerprints and physicochemical descriptors.
> 3. **Tier 3: Web Search (Low Priority):** When both the literature and structured databases lack relevant information, a web search API will be used as a constrained fallback to obtain short, factual descriptions.
>
> **How does it affect the results?**
>
> To evaluate the impact of RAG's partition variations introduced by LLMs' generative character, we conducted two additional analyses.
> 1. We ran the RAG module five times and found that while partition boundaries show slight variations, the underlying chemical logic remains highly consistent (detailed results in Appendix J.1.2).
> 2. In order to avoid the influence of pseudo-data, which may cover up the negative impact of the partition variation, we evaluated the Bayesian Optimization performance without the data module. Table R1 confirmed that the BO performance is robust to these partition variations, with different metrics remaining largely unaffected across 5 × 10 runs and being similar or better than the expert partition. These results demonstrate that our knowledge-driven strategy delivers reliable and consistent outcomes.
>
> **Table R1: RAG Sensitivity Analysis.**
>
> | Dataset | Partition Scheme | Initial (Mean ± Std.) | Best Found (Mean ± Std.) | 95% Iter (Mean ± Std.) | Best Iter (Mean ± Std.) |
> | :--- | :--- | :---: | :---: | :---: | :---: |
> | **Suzuki** | Subspaces | 66.94 ± 8.02 | 82.04 ± 4.49 | 8 ± 11 | 18 ± 14 |
> | | Subspaces-1 | 60.90 ± 17.11 | 82.30 ± 5.14 | 12 ± 13 | 21 ± 12 |
> | | Subspaces-2 | 64.46 ± 8.83 | 84.74 ± 7.87 | 15 ± 13 | 26 ± 12 |
> | | Subspaces-3 | 60.38 ± 15.03 | 94.92 ± 1.98 | 13 ± 11 | 14 ± 11 |
> | | Subspaces-4 | 58.09 ± 16.78 | 85.83 ± 7.61 | 13 ± 14 | 20 ± 14 |
> | | Expert | 61.82 ± 14.78 | 87.85 ± 6.44 | 17 ± 13 | 28 ± 11 |
> | **Arylation** | Subspaces | 59.38 ± 17.79 | 81.28 ± 2.12 | 11 ± 13 | 20 ± 14 |
> | | Subspaces-1 | 56.87 ± 20.19 | 81.40 ± 2.14 | 10 ± 9 | 28 ± 9 |
> | | Subspaces-2 | 56.87 ± 20.19 | 81.40 ± 2.14 | 10 ± 9 | 28 ± 9 |
> | | Subspaces-3 | 60.58 ± 13.88 | 81.25 ± 2.03 | 14 ± 9 | 28 ± 9 |
> | | Subspaces-4 | 54.12 ± 19.98 | 82.14 ± 2.14 | 15 ± 12 | 28 ± 8 |
> | | Expert | 50.07 ± 19.15 | 82.05 ± 2.32 | 16 ± 12 | 26 ± 13 |
> | **Buchwald_sub1** | Subspaces | 44.08 ± 26.71 | 80.25 ± 2.22 | 4 ± 2 | 10 ± 7 |
> | | Subspaces-1 | 50.39 ± 17.40 | 79.01 ± 1.31 | 9 ± 8 | 27 ± 13 |
> | | Subspaces-2 | 50.39 ± 17.40 | 79.01 ± 1.31 | 9 ± 8 | 27 ± 13 |
> | | Subspaces-3 | 50.39 ± 17.40 | 79.01 ± 1.31 | 9 ± 8 | 27 ± 13 |
> | | Subspaces-4 | 41.35 ± 22.53 | 79.52 ± 0.39 | 9 ± 9 | 32 ± 13 |
> | | Expert | 35.63 ± 28.22 | 79.37 ± 1.00 | 7 ± 3 | 23 ± 11 |
> | **Buchwald_sub2** | Subspaces | 18.19 ± 22.47 | 53.23 ± 0.38 | 18 ± 9 | 26 ± 7 |
> | | Subspaces-1 | 16.56 ± 17.42 | 53.72 ± 0.89 | 22 ± 7 | 26 ± 9 |
> | | Subspaces-2 | 22.52 ± 20.20 | 52.67 ± 1.89 | 13 ± 10 | 18 ± 12 |
> | | Subspaces-3 | 19.12 ± 19.96 | 51.62 ± 1.47 | 7 ± 6 | 28 ± 10 |
> | | Subspaces-4 | 19.12 ± 19.96 | 51.62 ± 1.47 | 7 ± 6 | 28 ± 10 |
> | | Expert | 12.23 ± 16.70 | 53.01 ± 0.87 | 14 ± 7 | 30 ± 11 |

---

> ### Author Response · Authors · 2025-12-03
> **Reply to Reviewer GLbf - Part 3/8**
>
> **Q5: Especially in Figure 2, the improvement is more obvious on Suzuki while not too significant for other datasets. If we use the same percentage of datasets as the initial points, as Suzuki has more data points, then the use of it for pretrain and SFT will be more stable.**
>
> **A5:** We agree that the improvement appears to be most pronounced on the Suzuki dataset in Figure 2. However, this is not due to its larger size and more stable fine-tuning. Instead, it is a direct consequence of two factors: ChemBOMAS's powerful initialization and the specific characteristics of the global optimum in each dataset.
>
> As detailed in the newly added Table R2, while ChemBOMAS rapidly converges to the global optimum across all benchmarks, the initial maximum value found at the first "search" step varies relative to this optimum. Specifically, in the Suzuki dataset, the gap between the initial maximum and the global optimum is the largest among all benchmarks. It is this substantial initial gap that makes the subsequent optimization steps in Figure 2 visually more dramatic.
>
> Furthermore, the apparent flatness of the curves after a few iterations is a sign of efficient convergence, not a lack of search. ChemBOMAS quickly identifies the extremely rare high-value regions (as shown by the scarcity of high-value points in Table R2), after which further significant improvement is not possible.
>
> **Table R2: Benchmark Dataset Details and ChemBOMAS Metrics over Ten Runs.**
>
> | Benchmarks | Global Maximum | Number and Proportion of ≥ 95% Maximum | Initial. (Mean $\pm$ Std.) | Best Found (Mean $\pm$ Std.) | 95% Iter. Round (Mean $\pm$ Std.) | Best Iter. Round (Mean $\pm$ Std.) |
> | :--- | :---: | :---: | :---: | :---: | :---: | :---: |
> | **Suzuki** | 96.15 | 2 / 0.04% | 92.24 $\pm$ 0.00 | 96.15 $\pm$ 0.00 | 1 $\pm$ 0 | 3 $\pm$ 0 |
> | **Arylation** | 84.65 | 3 / 0.08% | 82.63 $\pm$ 0.00 | 82.83 $\pm$ 0.64 | 1 $\pm$ 0 | 4 $\pm$ 10 |
> | **Buchwald_sub1** | 80.91 | 31 / 4.93% | 75.55 $\pm$ 0.00 | 79.97 $\pm$ 0.50 | 4 $\pm$ 5 | 23 $\pm$ 13 |
> | **Buchwald_sub2** | 56.81 | 5 / 0.65% | 53.33 $\pm$ 0.00 | 56.81 $\pm$ 0.00 | 2 $\pm$ 0 | 2 $\pm$ 0 |
>
> Q6: The knowledge module and data module do not always have a positive impact on the results, where the best iteration for Arylation is obtained when neither of these modules is imported.
> A6: We agree that the ablation result may appear counter-intuitive at first, but it is actually consistent with our expectations in the small, noisy datasets.
>
> **Why single-module variants can hurt in low-information settings?**
>
> Across these relatively small and noisy benchmarks (shown as Table 5 in Appendix F.2), introducing only a single module may increase the risk of an overconfident yet erroneous prior distribution:
>
> 1. Without the data-driven module: early clustering and pruning in the knowledge module are based on very limited and noisy statistics, which can lead to over-aggressive pruning of subspaces that actually contain high-value conditions.
>
> 2. Without the knowledge-driven module: the Gaussian Process is initialized purely from noisy pseudo-data; if these pseudo-labels are miscalibrated, they can misguide BO in the early rounds, a classical “noisy warm-start degradation” effect.
>
> Hence, the result that “no module” can outperform a single module on some metrics is not surprising: when information is extremely scarce, a weak prior can be worse than no prior.
>
> **Why is ChemBOMAS (both modules) still the desired configuration?**
>
> The full ChemBOMAS integrated data-driven and knowledge-driven modules can effectively adapt to scenarios with limited data and high noise. First, the data-driven module provides a broad global warm start from pseudo-labels, even if they are somewhat noisy. Second, the knowledge-driven module imposes partitioning of space, preventing noisy pseudo-data from misleading BO globally. When both are used together, noisy pseudo-data cannot overturn knowledge-guided pruning everywhere, and imperfect knowledge-based pruning cannot override dense pseudo-label evidence, yielding better overall behavior in low-data, high-noise conditions.

---

> ### Author Response · Authors · 2025-12-03
> **Reply to Reviewer GLbf - Part 4/8**
>
> **Q7: In Table 4, the experimental results are reported without variance;**
>
> **A7:** We appreciate the observation and added Table R3. Table R3 reports the various statistics (mean, standard deviation, confidence intervals, and p-values) for the optimization metrics across ten independent ablation experiments per benchmark.
>
> The significant test with most p-values below 0.05 in Table R3  further supports our conclusion that ablating either the knowledge-driven or the data-driven module leads to a significant degradation in optimization efficiency.
>
> **Table R3: Ablation Study with Statistical Analysis.**
>
> | Dataset | Metrics | Initial. (Mean ± Std.) | Initial. (95% CI) | Initial. ($p$-val) | Best Found (Mean ± Std.) | Best Found (95% CI) | Best Found ($p$-val) | 95% Iter. Round (Mean ± Std.) | 95% Iter. Round (95% CI) | 95% Iter. Round ($p$-val) | Best Iter. Round (Mean ± Std.) | Best Iter. Round (95% CI) | Best Iter. Round ($p$-val) |
> | :--- | :--- | :---: | :---: | :---: | :---: | :---: | :---: | :---: | :---: | :---: | :---: | :---: | :---: |
> | **Suzuki** | **Full ChemBOMAS** | 92.24 ± 0.00 | [92.24, 92.24] | - | 96.15 ± 0.00 | [96.15, 96.15] | - | 1 ± 0 | [1, 1] | - | 3 ± 0 | [3, 3] | - |
> | | w/o data | 65.09 ± 9.88 | [58.02, 72.16] | $P < 0.05$ | 83.26 ± 5.40 | [79.40, 87.12] | $P < 0.05$ | 10 ± 12 | [1, 18] | $P < 0.05$ | 20 ± 13 | [11, 30] | $P < 0.05$ |
> | | w/o knowledge | 58.91 ± 12.14 | [50.23, 67.60] | $P < 0.05$ | 96.15 ± 0.00 | [96.15, 96.15] | $P > 0.05$ | 8 ± 5 | [5, 12] | $P < 0.05$ | 9 ± 5 | [5, 12] | $P < 0.05$ |
> | | w/o both | 58.91 ± 12.14 | [50.23, 67.60] | $P < 0.05$ | 91.44 ± 7.58 | [86.02, 96.87] | $P > 0.05$ | 12 ± 10 | [5, 19] | $P < 0.05$ | 16 ± 7 | [10, 21] | $P < 0.05$ |
> | **Arylation** | **Full ChemBOMAS** | 82.63 ± 0.00 | [82.63, 82.63] | - | 82.83 ± 0.64 | [82.38, 83.29] | - | 1 ± 0 | [1, 1] | - | 4 ± 10 | [1, 11] | - |
> | | w/o data | 59.38 ± 17.79 | [46.66, 72.11] | $P < 0.05$ | 81.28 ± 2.12 | [79.76, 82.80] | $P > 0.05$ | 12 ± 13 | [2, 21] | $P < 0.05$ | 20 ± 14 | [10, 30] | $P < 0.05$ |
> | | w/o knowledge | 49.59 ± 15.10 | [38.79, 60.40] | $P < 0.05$ | 79.76 ± 0.11 | [79.68, 79.84] | $P < 0.05$ | 8 ± 3 | [5, 10] | $P < 0.05$ | 24 ± 11 | [16, 32] | $P < 0.05$ |
> | | w/o both | 49.59 ± 15.10 | [38.79, 60.40] | $P < 0.05$ | 82.83 ± 1.77 | [81.57, 84.10] | $P > 0.05$ | 12 ± 9 | [5, 18] | $P < 0.05$ | 27 ± 8 | [21, 33] | $P < 0.05$ |
> | **Buchwald_sub1\*** | **Full ChemBOMAS** | 79.52 ± 0.00 | [79.53, 79.53] | - | 80.45 ± 0.59 | [80.03, 80.87] | - | 1 ± 1 | [1, 1] | - | 10 ± 7 | [5, 15] | - |
> | | w/o data | 50.35 ± 28.93 | [29.66, 71.05] | $P < 0.05$ | 80.30 ± 2.18 | [78.74, 81.86] | $P > 0.05$ | 4 ± 2 | [2, 6] | $P < 0.05$ | 14 ± 8 | [8, 20] | $P < 0.05$ |
> | | w/o knowledge | 49.67 ± 18.48 | [36.45, 62.89] | $P < 0.05$ | 80.66 ± 0.53 | [80.28, 81.04] | $P > 0.05$ | 8 ± 5 | [5, 12] | $P < 0.05$ | 20 ± 10 | [13, 27] | $P > 0.05$ |
> | | w/o both | 53.57 ± 25.22 | [35.54, 71.61] | $P < 0.05$ | 79.74 ± 0.42 | [79.44, 80.04] | $P < 0.05$ | 5 ± 2 | [3, 6] | $P < 0.05$ | 12 ± 10 | [5, 19] | $P < 0.05$ |
> | **Buchwald_sub2** | **Full ChemBOMAS** | 53.33 ± 0.00 | [53.34, 53.34] | - | 56.81 ± 0.00 | [56.81, 56.81] | - | 2 ± 0 | [2, 2] | - | 2 ± 0 | [2, 2] | - |
> | | w/o data | 16.58 ± 22.98 | [0.14, 33.02] | $P < 0.05$ | 53.22 ± 0.38 | [52.95, 53.49] | $P < 0.05$ | 16 ± 8 | [10, 21] | $P < 0.05$ | 20 ± 5 | [16, 23] | $P < 0.05$ |
> | | w/o knowledge | 31.62 ± 13.55 | [21.92, 41.31] | $P < 0.05$ | 56.81 ± 0.00 | [56.81, 56.81] | $P > 0.05$ | 8 ± 3 | [6, 10] | $P < 0.05$ | 8 ± 3 | [6, 10] | $P < 0.05$ |
> | | w/o both | 31.62 ± 13.55 | [21.92, 41.31] | $P < 0.05$ | 56.61 ± 0.61 | [56.12, 57.05] | $P > 0.05$ | 20 ± 9 | [13, 27] | $P < 0.05$ | 22 ± 9 | [16, 28] | $P < 0.05$ |
>
> **To significantly show the degradation in the ablation study, Buchwald_sub1 was initiated with 4% labels, whereas other benchmarks were still initiated with 1% labels.*

---

> ### Author Response · Authors · 2025-12-03
> **Reply to Reviewer GLbf - Part 5/8**
>
> **Q8: The model's performance relies heavily on fine-tuning data volume, suggesting that substantial effort is needed to optimize the model for specific datasets.**
>
> **A8:** We thank the reviewer for this comment. We want to emphasize that our default setting already uses only 1% of the labeled data for fine-tuning, which corresponds to a very small absolute number of experiments on each benchmark.
>
> Importantly, we added sensitivity experiments that systematically vary the fine-tuning data volume (see Table R4). The additional experiment shows that ChemBOMAS maintains strong optimization performance even when the fine-tuning data is reduced to 0.25%–0.5% (merely ~20 labeled points per benchmark), a level that is readily feasible for initial wetlab exploration. Moreover, improvements in optimization performance exhibit diminishing marginal returns beyond the 1% labeled data volume.
>
> **Table R4: Impact of  Labeled Data Volume and Pseudo-data Quality.**
>
> | Dataset | SFT Data Ratio | SFT Data Num. | MAE | $R^2$ | Initial. (Mean ± Std.) | Best Found (Mean ± Std.) | 95% Iter. Round (Mean ± Std.) | Best Iter. Round (Mean ± Std.) |
> | :--- | :---: | :---: | :---: | :---: | :---: | :---: | :---: | :---: |
> | **Suzuki** | 0 | 0 | 40.18 | -2.05 | 44.00 ± 0.00 | 88.18 ± 8.41 | 17 ± 16 | 31 ± 8 |
> | | 0.02% | 1 | 26.04 | -0.53 | 21.55 ± 0.00 | 87.41 ± 8.45 | 20 ± 15 | 27 ± 11 |
> | | 0.10% | 6 | 27.51 | -0.69 | 37.34 ± 0.00 | 85.38 ± 6.41 | 15 ± 11 | 25 ± 9 |
> | | 0.25% | 14 | 27.8 | -0.53 | 76.01 ± 0.00 | 81.17 ± 3.84 | 5 ± 12 | 14 ± 13 |
> | | 0.50% | 29 | 21.13 | 0.02 | 92.06 ± 0.00 | 92.06 ± 0.00 | 1 ± 0 | 1 ± 0 |
> | | 1.00% | 50 | 19.47 | 0.2 | 92.24 ± 0.00 | 96.15 ± 0.00 | 1 ± 0 | 3 ± 0 |
> | | 2.00% | 115 | 15.92 | 0.39 | 92.24 ± 0.00 | 93.41 ± 1.89 | 1 ± 0 | 13 ± 19 |
> | | 4.00% | 230 | 13.44 | 0.54 | 92.24 ± 0.00 | 92.24 ± 0.00 | 1 ± 0 | 1 ± 0 |
> | | 8.00% | 461 | 10.77 | 0.68 | 74.96 ± 0.00 | 91.89 ± 0.52 | 2 ± 0 | 24 ± 10 |
> | | 16.00% | 922 | 8.23 | 0.79 | 88.80 ± 0.00 | 92.24 ± 0.00 | 1 ± 0 | 2 ± 0 |
> | | 32.00% | 1844 | 5.46 | 0.89 | 88.80 ± 0.00 | 96.15 ± 0.00 | 2 ± 0 | 2 ± 0 |
> | **Arylation** | 0 | 0 | 33.24 | -1.49 | 65.86 ± 0.00 | 82.20 ± 1.38 | 11 ± 6 | 22 ± 14 |
> | | 0.02% | 1 | 25.33 | -0.19 | 0.00 ± 0.00 | 80.58 ± 2.09 | 10 ± 6 | 16 ± 9 |
> | | 0.10% | 4 | 31.57 | -0.79 | 32.41 ± 0.00 | 81.50 ± 2.35 | 16 ± 11 | 29 ± 9 |
> | | 0.25% | 10 | 24.22 | -0.07 | 28.33 ± 0.00 | 80.94 ± 1.65 | 3 ± 1 | 21 ± 12 |
> | | 0.50% | 20 | 26.53 | -0.36 | 76.34 ± 0.00 | 81.70 ± 1.41 | 2 ± 1 | 9 ± 9 |
> | | 1.00% | 34 | 19.55 | 0.13 | 82.63 ± 0.00 | 82.83 ± 0.64 | 1 ± 0 | 4 ± 10 |
> | | 2.00% | 79 | 15.75 | 0.38 | 82.57 ± 0.00 | 82.98 ± 0.53 | 1 ± 0 | 13 ± 16 |
> | | 4.00% | 158 | 11.97 | 0.62 | 76.95 ± 0.00 | 83.60 ± 0.00 | 2 ± 0 | 14 ± 9 |
> | | 8.00% | 316 | 8.49 | 0.77 | 82.57 ± 0.00 | 83.60 ± 0.00 | 1 ± 0 | 13 ± 3 |
> | | 16.00% | 633 | 6.39 | 0.85 | 77.47 ± 0.00 | 83.29 ± 0.70 | 4 ± 5 | 26 ± 13 |
> | | 32.00% | 1266 | 3.65 | 0.95 | 82.57 ± 0.00 | 83.60 ± 0.00 | 1 ± 0 | 11 ± 2 |
> | **Buchwald_sub1** | 0 | 0 | 52.28 | -3.64 | 77.63 ± 0.00 | 79.83 ± 0.38 | 1 ± 0 | 27 ± 11 |
> | | 0.02% | 1 | 49.18 | -3.2 | 37.12 ± 0.00 | 79.93 ± 0.53 | 2 ± 0 | 16 ± 6 |
> | | 0.10% | 4 | 47.77 | -3.02 | 66.43 ± 0.00 | 80.11 ± 0.55 | 2 ± 0 | 13 ± 13 |
> | | 0.25% | 10 | 22.9 | 0.07 | 75.55 ± 0.00 | 79.60 ± 0.28 | 4 ± 2 | 22 ± 12 |
> | | 0.50% | 20 | 22.72 | 0 | 28.40 ± 0.00 | 79.91 ± 0.53 | 2 ± 0 | 27 ± 8 |
> | | 1.00% | 34 | 21.65 | 0.09 | 75.55 ± 0.00 | 79.97 ± 0.12 | 4 ± 1 | 23 ± 13 |
> | | 2.00% | 79 | 16.9 | 0.4 | 65.59 ± 0.00 | 79.73 ± 0.08 | 2 ± 0 | 22 ± 13 |
> | | 4.00% | 158 | 13.16 | 0.61 | 79.53 ± 0.00 | 80.45 ± 0.59 | 3 ± 0 | 10 ± 7 |
> | | 8.00% | 316 | 8.99 | 0.78 | 79.08 ± 0.00 | 80.68 ± 0.48 | 1 ± 0 | 13 ± 9 |
> | | 16.00% | 633 | 6.4 | 0.88 | 79.08 ± 0.00 | 80.11 ± 0.55 | 3 ± 0 | 8 ± 5 |
> | | 32.00% | 1266 | 3.59 | 0.96 | 79.53 ± 0.00 | 80.44 ± 0.60 | 3 ± 0 | 20 ± 8 |
> | **Buchwald_sub2** | 0 | 0 | 20.16 | -1 | 46.94 ± 0.00 | 53.69 ± 2.19 | 16 ± 12 | 23 ± 15 |
> | | 0.02% | 1 | 19 | -0.77 | 0.00 ± 0.00 | 52.52 ± 2.35 | 6 ± 3 | 14 ± 12 |
> | | 0.10% | 4 | 18.79 | -0.66 | 14.90 ± 0.00 | 55.57 ± 2.24 | 33 ± 13 | 36 ± 9 |
> | | 0.25% | 10 | 23.07 | -0.76 | 35.26 ± 0.00 | 52.77 ± 1.96 | 5 ± 2 | 14 ± 12 |
> | | 0.50% | 20 | 16.02 | -0.01 | 43.09 ± 0.00 | 53.34 ± 0.77 | 11 ± 11 | 28 ± 8 |
> | | 1.00% | 34 | 12.15 | 0.39 | 53.33 ± 0.00 | 56.81 ± 0.00 | 2 ± 0 | 2 ± 0 |
> | | 2.00% | 79 | 9.76 | 0.51 | 53.10 ± 0.00 | 53.94 ± 1.11 | 5 ± 12 | 18 ± 14 |
> | | 4.00% | 158 | 8.7 | 0.56 | 52.01 ± 0.00 | 53.83 ± 2.10 | 10 ± 15 | 18 ± 14 |
> | | 8.00% | 316 | 5.26 | 0.81 | 50.21 ± 0.00 | 55.21 ± 1.70 | 8 ± 7 | 17 ± 11 |
> | | 16.00% | 633 | 4.06 | 0.85 | 53.33 ± 0.00 | 54.78 ± 1.47 | 8 ± 13 | 14 ± 15 |
> | | 32.00% | 1266 | 2.61 | 0.92 | 53.33 ± 0.00 | 54.72 ± 1.16 | 2 ± 2 | 10 ± 11 |

---

> ### Author Response · Authors · 2025-12-03
> **Reply to Reviewer GLbf - Part 6/8**
>
> **Q9: All the results are reported based on five random seeds; this is not reliable for heuristic approaches.**
>
> **A9:** In response, we have rerun all key experiments with **ten** independent random seeds and updated the results to include the **mean, standard deviation, 95% confidence intervals, and significance tests (paired t-tests)** for baseline comparisons, ablation studies, RAG sensitivity analyses, and pseudo-data experiments. The expanded statistical analysis further strengthens our conclusions:
> 1. ChemBOMAS consistently and significantly outperforms all baseline methods across all four benchmark datasets. (**see Table R5**)
> 2. The ablation study confirms that removing either the knowledge-driven or data-driven module leads to a statistically significant performance degradation. (**see Table R3**)
> 3. The RAG sensitivity analysis demonstrates that our knowledge-driven module is robust, with performance remaining stable across different partitioning runs. (**see Table R1**)
> 4. Even when compared to scenarios using a high proportion of labeled data (8% to 32%), ChemBOMAS achieves comparable optimization performance with only 1% or even 0.5% labeled data. (**see Table R4**)
>
> **Table R1: Baseline Comparisons.**
>
> | Dataset | Method | Initial. (Mean ± Std.) | Initial. (95% CI) | Initial. ($p$-val) | Best Found (Mean ± Std.) | Best Found (95% CI) | Best Found ($p$-val) | Best Iter. Round (Mean ± Std.) | Best Iter. Round (95% CI) | Best Iter. Round ($p$-val) | 95% Iter. Round (Mean ± Std.) | 95% Iter. Round (95% CI) | 95% Iter. Round ($p$-val) |
> | :--- | :--- | :---: | :---: | :---: | :---: | :---: | :---: | :---: | :---: | :---: | :---: | :---: | :---: |
> | **Suzuki** | BO | 58.91 ± 12.14 | [50.23, 67.60] | $P < 0.05$ | 91.45 ± 7.58 | [86.02, 96.87] | $P > 0.05$ | 16 ± 7 | [10, 21] | $P < 0.05$ | 12 ± 10 | [5, 19] | $P < 0.05$ |
> | | BO-ICL | 76.02 ± 0.07 | [75.94, 76.10] | $P < 0.05$ | 80.37 ± 5.93 | [74.14, 86.59] | $P < 0.05$ | 21 ± 13 | [7, 34] | $P < 0.05$ | 6 ± 12 | [1, 18] | $P > 0.05$ |
> | | LA-MCTS | 77.70 ± 1.67 | [76.51, 78.90] | $P < 0.05$ | 78.43 ± 1.15 | [77.60, 79.25] | $P < 0.05$ | 3 ± 4 | [0, 6] | $P > 0.05$ | 1 ± 0 | [1, 1] | $P > 0.05$ |
> | | GOLLuM | 79.10 ± 26.53 | [26.27, 26.79] | $P < 0.05$ | 78.07 ± 6.67 | [73.30, 82.85] | $P < 0.05$ | 31 ± 7 | [26, 36] | $P < 0.05$ | 25 ± 9 | [18, 31] | $P < 0.05$ |
> | | **ChemBOMAS** | 92.24 ± 0.00 | [92.24, 92.24] | - | 96.15 ± 0.00 | [96.15, 96.15] | - | 3 ± 0 | [3, 3] | - | 1 ± 0| [1, 1] | - |
> | **Arylation** | BO | 49.59 ± 15.10 | [38.79, 60.40] | $P < 0.05$ | 82.83 ± 1.77 | [81.57, 84.10] | $P > 0.05$ | 27 ± 8 | [21, 33] | $P < 0.05$ | 12 ± 9 | [0, 18] | $P < 0.05$ |
> | | BO-ICL | 76.43 ± 0.77 | [5.62, 77.23] | $P < 0.05$ | 78.63 ± 1.21 | [77.36, 79.91] | $P < 0.05$ | 27 ± 13 | [13, 41] | $P < 0.05$ | 1 ± 0 | [1, 1] | $P > 0.05$ |
> | | LA-MCTS | 67.85 ± 7.53 | [2.46, 73.24] | $P < 0.05$ | 74.12 ± 4.03 | [71.23, 77.01] | $P < 0.05$ | 13 ± 11 | [5, 21] | $P > 0.05$ | 6 ± 8 | [1, 11] | $P > 0.05$ |
> | | GOLLuM | 23.77 ± 1.17 | [22.93, 24.60] | $P < 0.05$ | 76.99 ± 8.39 | [70.99, 83.00] | $P > 0.05$ | 29 ± 11 | [21, 36] | $P < 0.05$ | 21 ± 11 | [13, 29] | $P < 0.05$ |
> | | **ChemBOMAS** | 82.63 ± 0.00 | [82.63, 82.63] | - | 82.83 ± 0.64 | [82.38, 83.29] | - | 4 ± 10 | [1, 11] | - | 1 ± 0 | [1, 1] | - |
> | **Buchwald_sub1** | BO | 53.57 ± 25.22 | [35.54, 71.61] | $P < 0.05$ | 79.74 ± 0.42 | [79.44, 80.04] | $P > 0.05$ | 12 ± 10 | [5, 19] | $P < 0.05$ | 5 ± 2 | [3, 6] | $P > 0.05$ |
> | | BO-ICL | 69.78 ± 0.64 | [69.11, 70.45] | $P < 0.05$ | 78.26 ± 2.53 | [75.61, 80.91] | $P > 0.05$ | 19 ± 13 | [5, 33] | $P > 0.05$ | 7 ± 6 | [1, 13] | $P > 0.05$ |
> | | LA-MCTS | 71.10 ± 13.22 | [1.64, 80.55] | $P > 0.05$ | 75.52 ± 3.95 | [72.69, 78.34] | $P < 0.05$ | 3 ± 5 | [1, 7] | $P < 0.05$ | 2 ± 4 | [1, 6] | $P > 0.05$ |
> | | GOLLuM | 36.32 ± 4.02 | [3.45, 39.19] | $P < 0.05$ | 79.77 ± 0.72 | [79.25, 80.28] | $P > 0.05$ | 25 ± 12 | [17, 34] | $P > 0.05$ | 13 ± 7 | [8, 18] | $P < 0.05$ |
> | | **ChemBOMAS** | 75.55 ± 0.00 | [5.55, 75.55] | - | 79.97 ± 0.50 | [79.62, 80.33] | - | 23 ± 13 | [4, 33] | - | 4 ± 5 | [0, 8] | - |
> | **Buchwald_sub2** | BO | 31.62 ± 13.55 | [21.92, 41.31] | $P < 0.05$ | 56.61 ± 0.61 | [56.18, 57.05] | $P > 0.05$ | 22 ± 9 | [16, 28] | $P < 0.05$ | 20 ± 9 | [3, 27] | $P < 0.05$ |
> | | BO-ICL | 46.12 ± 2.85 | [43.12, 49.11] | $P < 0.05$ | 53.14 ± 1.93 | [51.12, 55.16] | $P < 0.05$ | 24 ± 10 | [20, 40] | $P < 0.05$ | 23 ± 12 | [1, 35] | $P < 0.05$ |
> | | LA-MCTS | 48.25 ± 4.00 | [45.39, 51.11] | $P < 0.05$ | 51.63 ± 3.40 | [49.19, 54.06] | $P < 0.05$ | 6 ± 8 | [0, 12] | $P > 0.05$ | 4 ± 7 | [0, 9] | $P > 0.05$ |
> | | GOLLuM | 7.46 ± 3.08 | [5.25, 9.66] | $P < 0.05$ | 54.99 ± 1.99 | [53.57, 56.41] | $P < 0.05$ | 28 ± 12 | [19, 37] | $P < 0.05$ | 24 ± 11 | [16, 31] | $P < 0.05$ |
> | | **ChemBOMAS** | 53.33 ± 0.00 | [53.34, 53.34] | - | 56.81 ± 0.00 | [56.81, 56.81] | - | 2 ± 0 | [2, 2] | - | 2 ± 0 | [2, 2] | - |

---

> ### Author Response · Authors · 2025-12-03
> **Reply to Reviewer GLbf - Part 7/8**
>
> **Q10: The integration of knowledge-driven and data-driven modules seem complex and convoluted.**
>
> **A10:** Thanks for your comments. To make the implemented ChemBOMAS pipeline easier to understand, we simplify the description around Figure 1 to explicitly frame ChemBOMAS as four steps.
>
> 1. Use RAG + LLM once to rank variable dimensions by importance and cluster candidate variables into a hierarchical search tree.
>
> 2. Use the fine-tuned LLM regressor once to assign pseudo-labels to all unsampled points; this scores the partitions by Upper Confidence Bound (UCB) on the search tree and defines an informative prior for Gaussian Process (GP).
>
> 3. Within the high-scoring subspaces, run a conventional GP-based BO loop, initialized with real data and pseudo-data.
>
> 4. After each real observation, update node statistics along the path in the tree to rescore the subspace; the next iteration repeats from Step 3.
>
> Furthermore, we want to emphasize that knowledge-driven and data-driven modules are not cosmetic but necessary. As shown in the response to Q7, removing either module results in a significant decrease in optimization efficiency.
>
> **Q11: How does ChemBOMAS handle scalability issues, especially in scenarios involving extremely large and complex chemical spaces?**
>
> **A11:** We thank the reviewer for this critical question regarding scalability. ChemBOMAS is specifically designed to tackle large and complex chemical spaces through a multi-stage algorithmic structure that avoids exhaustive search, and its efficacy has been validated in a real-world wet-lab experiment.
>
> 1. **Algorithmic Mechanisms for Scalability.**
>
> ChemBOMAS enhances scalability through a structured, hierarchical approach:
>
> - Knowledge-Driven Space Partitioning: The RAG-augmented LLM first partitions the vast original space into a small set of chemically meaningful subspaces. This critical step reduces the problem from exploring all raw combinations to evaluating a manageable number of coherent regions.
> - Efficient Subspace Prioritization: An LLM regressor, fine-tuned on minimal data (1%), generates pseudo-labels for candidates. By aggregating these labels at the subspace level, our method uses the UCB criterion to rapidly identify and focus on the most promising subspaces, bypassing the need for exhaustive evaluation.
> - Focused Bayesian Optimization: BO is subsequently conducted only within the selected high-value subspaces, which are initialized with pseudo-labels and efficiently updated with real experimental observations.
>
> 2. **Empirical Evidence from a Real Wet-Lab Experiment.**
>
> The scalability of ChemBOMAS is demonstrated in a wet-lab study involving a ~6-dimensional reaction system (e.g., catalyst, ligand, base, solvent, temperature, water content), constituting a design space of approximately 10^5 conditions. Guided by ChemBOMAS, chemists identified a high-performing condition (96% yield) for a previously unreported reaction after only 43 experiments over five rounds, starting from zero prior data. This result underscores the method's power to efficiently navigate extremely large spaces that are infeasible for exhaustive approaches.
>
> **Q12: Did you query for space partition at every BO iteration? If so, it should be quite expensive. What's the cost for the knowledge-driven process that combines RAG and general LLMs?**
>
> **A12:** No, the RAG module for knowledge-driven space partitioning is executed **only once** during the initialization phase of ChemBOMAS. It is not queried at every BO iteration. Specifically, this one-time computational cost is approximately $0.0173 in API calls and about 36.9 seconds in processing time. This fixed, low cost makes the approach practical, as it can be completed offline without imposing any burden on the online optimization cycle.

---

> ### Author Response · Authors · 2025-12-03
> **Reply to Reviewer GLbf - Part 8/8**
>
> **Q13: What mechanisms are in place to adapt ChemBOMAS effectively under varying initial conditions concerning data volume?**
>
> **A13:** Thank you for this question. ChemBOMAS is explicitly designed to operate across different label regimes, from very scarce data to moderately rich settings.
> 1. **For low-data scenarios (<1%)**, the LLM regressor is first pretrained on large unlabeled chemical corpora, and all prior data is then used for SFT and GP initialization.  Pseudo-labels generated by the LLM regressor are progressively down-weighted and pruned as real observations accumulate. Moreover, the knowledge-driven module is independent of the exact label fraction and provides structural space partitioning, preventing noisy pseudo-data from globally misleading BO. Importantly, the additional data volume sensitivity experiments (Table R4) prove the validity of the ChemBOMAS down to 0.25%–0.5% (merely ~20 labeled points per benchmark).
>
> ****
>
> 2. **For high-data scenarios (>1%)**, Table R4 shows that improvements in optimization performance exhibit diminishing marginal returns beyond the 1% labeled data volume. To prevent overfitting and unnecessary computational costs, it is recommended to limit SFT to a small subset (1-2%), while leveraging the surplus of real data to enrich the GP model, and increasing the proportion of pruned pseudo labels. Under this configuration, ChemBOMAS is gradually evolving into a knowledge-driven BO accelerator.
>
> Thank you again for your time and review. **We believe these revisions, together with the additional experiments, results, and analyses, substantially strengthen the soundness, clarity, and significance of the work.**

---

### Official Review · Reviewer_NBfQ · 2025-10-31

**Soundness:** 3
**Presentation:** 3
**Contribution:** 3
**Rating:** 6
**Confidence:** 3

**Summary:**

The paper proposes ChemBOMAS, an LLM-enhanced multi-agent framework that accelerates Bayesian optimization for experimental chemistry by combining a knowledge-driven RAG module that partitions the design space with a data-driven module that uses a fine-tuned 8B LLM regressor to produce pseudo-labels for more informative initialization. A UCB-guided subspace selection policy coordinates the two modules in a closed loop, after which BO is run within promising regions. The authors evaluate on standard reaction-optimization benchmarks and a materials-style task, and they report faster early-stage convergence and higher best-of-budget outcomes than strong baselines. Importantly, they provide supplementary materials with code and data that enable replication, and they include an experimental wet-lab validation on a previously unpublished reaction system.

**Strengths:**

1. Reproducibility and transparency. The authors provide supplementary materials with code and data, including implementation details sufficient to rerun core experiments and inspect prompts and baselines.
2. Practical validation. The approach is tested in a real wet-lab setting with strong early-round gains, demonstrating operational relevance beyond simulation.
3. Clear algorithmic scaffold. The closed-loop interplay between RAG-based subspace construction, UCB-guided selection, and BO within selected regions is well specified and easy to follow.
4. Broad and relevant evaluation. Benchmarks cover standard reaction datasets and a materials-style task, with ablations that probe the value of pseudo-data and partitioning.
5. Sensible problem framing. The method directly addresses cold-start and high-dimensionality pain points that routinely limit BO in chemistry.

**Weaknesses:**

1. Statistical reporting. Claims such as up to 5-fold acceleration and several percent gains over baselines are not consistently accompanied by confidence intervals, variance across multiple seeds, or significance tests. This is especially important for stochastic BO and LLM-generated pseudo-labels.
2. Safety and feasibility constraints. The current loop does not appear to enforce domain safety or feasibility filters to prevent proposing hazardous or impractical conditions. Incorporating rule-based checks, surrogate predictive models of feasibility, or expert-validated constraints would improve readiness for autonomous labs.
3. Pseudo-label calibration and bias. The LLM regressor can introduce bias if mis-calibrated. There is limited analysis of reliability, label-noise sensitivity, or robustness when pseudo-data conflict with early measurements.
4. RAG partitioning auditability. The retrieval corpus, filtering criteria, and sensitivity of subspace splits to retrieval noise are under-documented. Without this, it is hard to assess whether partitions encode prior bias that drives observed gains.
5. Baseline alignment and coverage. While strong baselines are included, details on kernel choices, acquisition settings, restart budgets, batch sizes, and runtime are scattered. Broader comparisons to recent hybrid methods and a clearer accounting of computational cost would strengthen fairness and scope.
6. Generality claims. The materials-style task is promising but under-described. More context on its construction and accessibility would help calibrate the claim of cross-domain applicability.

**Questions:**

1. Can you provide means, standard deviations, confidence intervals, and significance tests for all benchmarks and wet-lab studies?
2. How sensitive is the performance of the Bayesian optimization loop to noise or mis-calibration of the LLM-generated pseudo-labels?
3. Is the LLM regressor kept fixed after initial fine-tuning or updated online, and what safeguards prevent confirmation bias in that process?
4. What retrieval corpus and filtering criteria are used for the RAG partitioning, and how often do retrieval errors lead to sub-optimal subspace splits?
5. Are safety or feasibility constraints (e.g., chemical hazard filters) integrated into the proposal generation loop, and if not, how might this affect deployment in autonomous labs?
6. For each reaction dataset and pre-trained component, has overlap between pre-training data and evaluation sets been checked to rule out data leakage?
7. Could you clarify the materials-style “LNP3” task: define its dataset, availability, and whether a minimal open-version will be released for benchmarking?

---

> ### Author Response · Authors · 2025-12-03
> **Reply to Reviewer NBfQ - Part 1/9**
>
> We are grateful for the detailed comments. Here, we present our responses to your questions, point by point.
>
> **Q1: Statistical reporting. Claims such as up to 5-fold acceleration and several percent gains over baselines are not consistently accompanied by confidence intervals, variance across multiple seeds, or significance tests. This is especially important for stochastic BO and LLM-generated pseudo-labels.**
>
> **A1:** We thank the reviewer for raising this important concern regarding statistical reliability. In response, we have rerun all key experiments with ten independent random seeds and updated the results to include the mean, standard deviation, 95% confidence intervals, and significance tests (paired t-tests) for baseline comparisons, ablation studies, RAG sensitivity analyses, and pseudo-data experiments. The expanded statistical analysis further strengthens our conclusions:
> 1. ChemBOMAS consistently and significantly outperforms all baseline methods across all four benchmark datasets (**see Table R1**).
> 2. The ablation study confirms that removing either the knowledge-driven or data-driven module leads to a statistically significant performance degradation (**see Table R2**).
> 3. The RAG sensitivity analysis demonstrates that our knowledge-driven module is robust, with performance remaining stable across different partitioning runs (**see Table R3**).
> 4. Even when compared to scenarios using a high proportion of labeled data (8% to 32%), ChemBOMAS achieves comparable optimization performance with only 1% or even 0.5% labeled data (**see Table R4**).

---

> ### Author Response · Authors · 2025-12-03
> **Reply to Reviewer NBfQ - Part 2/9**
>
> **Table R1: Baseline Comparisons.**
>
> | Dataset | Method | Initial. (Mean ± Std.) | Initial. (95% CI) | Initial. ($p$-val) | Best Found (Mean ± Std.) | Best Found (95% CI) | Best Found ($p$-val) | Best Iter. Round (Mean ± Std.) | Best Iter. Round (95% CI) | Best Iter. Round ($p$-val) | 95% Iter. Round (Mean ± Std.) | 95% Iter. Round (95% CI) | 95% Iter. Round ($p$-val) |
> | :--- | :--- | :---: | :---: | :---: | :---: | :---: | :---: | :---: | :---: | :---: | :---: | :---: | :---: |
> | **Suzuki** | BO | 58.91 ± 12.14 | [50.23, 67.60] | $P < 0.05$ | 91.45 ± 7.58 | [86.02, 96.87] | $P > 0.05$ | 16 ± 7 | [10, 21] | $P < 0.05$ | 12 ± 10 | [5, 19] | $P < 0.05$ |
> | | BO-ICL | 76.02 ± 0.07 | [75.94, 76.10] | $P < 0.05$ | 80.37 ± 5.93 | [74.14, 86.59] | $P < 0.05$ | 21 ± 13 | [7, 34] | $P < 0.05$ | 6 ± 12 | [1, 18] | $P > 0.05$ |
> | | LA-MCTS | 77.70 ± 1.67 | [76.51, 78.90] | $P < 0.05$ | 78.43 ± 1.15 | [77.60, 79.25] | $P < 0.05$ | 3 ± 4 | [0, 6] | $P > 0.05$ | 1 ± 0 | [1, 1] | $P > 0.05$ |
> | | GOLLuM | 79.10 ± 26.53 | [26.27, 26.79] | $P < 0.05$ | 78.07 ± 6.67 | [73.30, 82.85] | $P < 0.05$ | 31 ± 7 | [26, 36] | $P < 0.05$ | 25 ± 9 | [18, 31] | $P < 0.05$ |
> | | **ChemBOMAS** | 92.24 ± 0.00 | [92.24, 92.24] | - | 96.15 ± 0.00 | [96.15, 96.15] | - | 3 ± 0 | [3, 3] | - | 1 ± 0| [1, 1] | - |
> | **Arylation** | BO | 49.59 ± 15.10 | [38.79, 60.40] | $P < 0.05$ | 82.83 ± 1.77 | [81.57, 84.10] | $P > 0.05$ | 27 ± 8 | [21, 33] | $P < 0.05$ | 12 ± 9 | [0, 18] | $P < 0.05$ |
> | | BO-ICL | 76.43 ± 0.77 | [5.62, 77.23] | $P < 0.05$ | 78.63 ± 1.21 | [77.36, 79.91] | $P < 0.05$ | 27 ± 13 | [13, 41] | $P < 0.05$ | 1 ± 0 | [1, 1] | $P > 0.05$ |
> | | LA-MCTS | 67.85 ± 7.53 | [2.46, 73.24] | $P < 0.05$ | 74.12 ± 4.03 | [71.23, 77.01] | $P < 0.05$ | 13 ± 11 | [5, 21] | $P > 0.05$ | 6 ± 8 | [1, 11] | $P > 0.05$ |
> | | GOLLuM | 23.77 ± 1.17 | [22.93, 24.60] | $P < 0.05$ | 76.99 ± 8.39 | [70.99, 83.00] | $P > 0.05$ | 29 ± 11 | [21, 36] | $P < 0.05$ | 21 ± 11 | [13, 29] | $P < 0.05$ |
> | | **ChemBOMAS** | 82.63 ± 0.00 | [82.63, 82.63] | - | 82.83 ± 0.64 | [82.38, 83.29] | - | 4 ± 10 | [1, 11] | - | 1 ± 0 | [1, 1] | - |
> | **Buchwald_sub1** | BO | 53.57 ± 25.22 | [35.54, 71.61] | $P < 0.05$ | 79.74 ± 0.42 | [79.44, 80.04] | $P > 0.05$ | 12 ± 10 | [5, 19] | $P < 0.05$ | 5 ± 2 | [3, 6] | $P > 0.05$ |
> | | BO-ICL | 69.78 ± 0.64 | [69.11, 70.45] | $P < 0.05$ | 78.26 ± 2.53 | [75.61, 80.91] | $P > 0.05$ | 19 ± 13 | [5, 33] | $P > 0.05$ | 7 ± 6 | [1, 13] | $P > 0.05$ |
> | | LA-MCTS | 71.10 ± 13.22 | [1.64, 80.55] | $P > 0.05$ | 75.52 ± 3.95 | [72.69, 78.34] | $P < 0.05$ | 3 ± 5 | [1, 7] | $P < 0.05$ | 2 ± 4 | [1, 6] | $P > 0.05$ |
> | | GOLLuM | 36.32 ± 4.02 | [3.45, 39.19] | $P < 0.05$ | 79.77 ± 0.72 | [79.25, 80.28] | $P > 0.05$ | 25 ± 12 | [17, 34] | $P > 0.05$ | 13 ± 7 | [8, 18] | $P < 0.05$ |
> | | **ChemBOMAS** | 75.55 ± 0.00 | [5.55, 75.55] | - | 79.97 ± 0.50 | [79.62, 80.33] | - | 23 ± 13 | [4, 33] | - | 4 ± 5 | [0, 8] | - |
> | **Buchwald_sub2** | BO | 31.62 ± 13.55 | [21.92, 41.31] | $P < 0.05$ | 56.61 ± 0.61 | [56.18, 57.05] | $P > 0.05$ | 22 ± 9 | [16, 28] | $P < 0.05$ | 20 ± 9 | [3, 27] | $P < 0.05$ |
> | | BO-ICL | 46.12 ± 2.85 | [43.12, 49.11] | $P < 0.05$ | 53.14 ± 1.93 | [51.12, 55.16] | $P < 0.05$ | 24 ± 10 | [20, 40] | $P < 0.05$ | 23 ± 12 | [1, 35] | $P < 0.05$ |
> | | LA-MCTS | 48.25 ± 4.00 | [45.39, 51.11] | $P < 0.05$ | 51.63 ± 3.40 | [49.19, 54.06] | $P < 0.05$ | 6 ± 8 | [0, 12] | $P > 0.05$ | 4 ± 7 | [0, 9] | $P > 0.05$ |
> | | GOLLuM | 7.46 ± 3.08 | [5.25, 9.66] | $P < 0.05$ | 54.99 ± 1.99 | [53.57, 56.41] | $P < 0.05$ | 28 ± 12 | [19, 37] | $P < 0.05$ | 24 ± 11 | [16, 31] | $P < 0.05$ |
> | | **ChemBOMAS** | 53.33 ± 0.00 | [53.34, 53.34] | - | 56.81 ± 0.00 | [56.81, 56.81] | - | 2 ± 0 | [2, 2] | - | 2 ± 0 | [2, 2] | - |

---

> ### Author Response · Authors · 2025-12-03
> **Reply to Reviewer NBfQ - Part 3/9**
>
> **Table R2: Ablation Study.**
>
> | Dataset | Metrics | Initial. (Mean ± Std.) | Initial. (95% CI) | Initial. ($p$-val) | Best Found (Mean ± Std.) | Best Found (95% CI) | Best Found ($p$-val) | 95% Iter. Round (Mean ± Std.) | 95% Iter. Round (95% CI) | 95% Iter. Round ($p$-val) | Best Iter. Round (Mean ± Std.) | Best Iter. Round (95% CI) | Best Iter. Round ($p$-val) |
> | :--- | :--- | :---: | :---: | :---: | :---: | :---: | :---: | :---: | :---: | :---: | :---: | :---: | :---: |
> | **Suzuki** | **Full ChemBOMAS** | 92.24 ± 0.00 | [92.24, 92.24] | - | 96.15 ± 0.00 | [96.15, 96.15] | - | 1 ± 0 | [1, 1] | - | 3 ± 0 | [3, 3] | - |
> | | w/o data | 65.09 ± 9.88 | [58.02, 72.16] | $P < 0.05$ | 83.26 ± 5.40 | [79.40, 87.12] | $P < 0.05$ | 10 ± 12 | [1, 18] | $P < 0.05$ | 20 ± 13 | [11, 30] | $P < 0.05$ |
> | | w/o knowledge | 58.91 ± 12.14 | [50.23, 67.60] | $P < 0.05$ | 96.15 ± 0.00 | [96.15, 96.15] | $P > 0.05$ | 8 ± 5 | [5, 12] | $P < 0.05$ | 9 ± 5 | [5, 12] | $P < 0.05$ |
> | | w/o both | 58.91 ± 12.14 | [50.23, 67.60] | $P < 0.05$ | 91.44 ± 7.58 | [86.02, 96.87] | $P > 0.05$ | 12 ± 10 | [5, 19] | $P < 0.05$ | 16 ± 7 | [10, 21] | $P < 0.05$ |
> | **Arylation** | **Full ChemBOMAS** | 82.63 ± 0.00 | [82.63, 82.63] | - | 82.83 ± 0.64 | [82.38, 83.29] | - | 1 ± 0 | [1, 1] | - | 4 ± 10 | [1, 11] | - |
> | | w/o data | 59.38 ± 17.79 | [46.66, 72.11] | $P < 0.05$ | 81.28 ± 2.12 | [79.76, 82.80] | $P > 0.05$ | 12 ± 13 | [2, 21] | $P < 0.05$ | 20 ± 14 | [10, 30] | $P < 0.05$ |
> | | w/o knowledge | 49.59 ± 15.10 | [38.79, 60.40] | $P < 0.05$ | 79.76 ± 0.11 | [79.68, 79.84] | $P < 0.05$ | 8 ± 3 | [5, 10] | $P < 0.05$ | 24 ± 11 | [16, 32] | $P < 0.05$ |
> | | w/o both | 49.59 ± 15.10 | [38.79, 60.40] | $P < 0.05$ | 82.83 ± 1.77 | [81.57, 84.10] | $P > 0.05$ | 12 ± 9 | [5, 18] | $P < 0.05$ | 27 ± 8 | [21, 33] | $P < 0.05$ |
> | **Buchwald_sub1\*** | **Full ChemBOMAS** | 79.52 ± 0.00 | [79.53, 79.53] | - | 80.45 ± 0.59 | [80.03, 80.87] | - | 1 ± 1 | [1, 1] | - | 10 ± 7 | [5, 15] | - |
> | | w/o data | 50.35 ± 28.93 | [29.66, 71.05] | $P < 0.05$ | 80.30 ± 2.18 | [78.74, 81.86] | $P > 0.05$ | 4 ± 2 | [2, 6] | $P < 0.05$ | 14 ± 8 | [8, 20] | $P < 0.05$ |
> | | w/o knowledge | 49.67 ± 18.48 | [36.45, 62.89] | $P < 0.05$ | 80.66 ± 0.53 | [80.28, 81.04] | $P > 0.05$ | 8 ± 5 | [5, 12] | $P < 0.05$ | 20 ± 10 | [13, 27] | $P > 0.05$ |
> | | w/o both | 53.57 ± 25.22 | [35.54, 71.61] | $P < 0.05$ | 79.74 ± 0.42 | [79.44, 80.04] | $P < 0.05$ | 5 ± 2 | [3, 6] | $P < 0.05$ | 12 ± 10 | [5, 19] | $P < 0.05$ |
> | **Buchwald_sub2** | **Full ChemBOMAS** | 53.33 ± 0.00 | [53.34, 53.34] | - | 56.81 ± 0.00 | [56.81, 56.81] | - | 2 ± 0 | [2, 2] | - | 2 ± 0 | [2, 2] | - |
> | | w/o data | 16.58 ± 22.98 | [0.14, 33.02] | $P < 0.05$ | 53.22 ± 0.38 | [52.95, 53.49] | $P < 0.05$ | 16 ± 8 | [10, 21] | $P < 0.05$ | 20 ± 5 | [16, 23] | $P < 0.05$ |
> | | w/o knowledge | 31.62 ± 13.55 | [21.92, 41.31] | $P < 0.05$ | 56.81 ± 0.00 | [56.81, 56.81] | $P > 0.05$ | 8 ± 3 | [6, 10] | $P < 0.05$ | 8 ± 3 | [6, 10] | $P < 0.05$ |
> | | w/o both | 31.62 ± 13.55 | [21.92, 41.31] | $P < 0.05$ | 56.61 ± 0.61 | [56.12, 57.05] | $P > 0.05$ | 20 ± 9 | [13, 27] | $P < 0.05$ | 22 ± 9 | [16, 28] | $P < 0.05$ |
>
> **To significantly show the degradation in the ablation study, Buchwald_sub1 was initiated with 4% labels, whereas other benchmarks were still initiated with 1% labels.*

---

> ### Author Response · Authors · 2025-12-03
> **Reply to Reviewer NBfQ - Part 4/9**
>
> **Table R3: RAG Sensitivity Analysis on BO Performance.**
>
> | Dataset | Partitioning | Initial. (Mean ± Std.) | Initial. (95% CI) | Initial. ($p$-val) | Best Found (Mean ± Std.) | Best Found (95% CI) | Best Found ($p$-val) | 95% Iter. Round (Mean ± Std) | 95% Iter. Round (95% CI) | 95% Iter. Round ($p$-val) | Best Iter. Round (Mean ± Std) | Best Iter. Round (95% CI) | Best Iter. Round ($p$-val) |
> | :--- | :--- | :---: | :---: | :---: | :---: | :---: | :---: | :---: | :---: | :---: | :---: | :---: | :---: |
> | **Suzuki** | Subspaces | 66.94 ± 8.02 | [61.21, 72.68] | - | 82.04 ± 4.49 | [78.83, 85.26] | - | 8 ± 11 | [0, 16] | - | 18 ± 14 | [8, 28] | - |
> | | Subspaces-1 | 60.90 ± 17.11 | [48.66, 73.14] | $p>0.05$ | 82.30 ± 5.14 | [78.63, 85.98] | $p>0.05$ | 12 ± 13 | [2, 21] | $p>0.05$ | 21 ± 12 | [12, 30] | $p>0.05$ |
> | | Subspaces-2 | 64.46 ± 8.83 | [58.14, 70.77] | $p>0.05$ | 84.74 ± 7.87 | [79.11, 90.37] | $p>0.05$ | 15 ± 13 | [5, 24] | $p>0.05$ | 26 ± 12 | [18, 34] | $p>0.05$ |
> | | Subspaces-3 | 60.38 ± 15.03 | [49.63, 71.13] | $p>0.05$ | 94.92 ± 1.98 | [93.51, 96.34] | $p<0.05$ | 13 ± 11 | [5, 21] | $p>0.05$ | 14 ± 11 | [6, 23] | $p>0.05$ |
> | | Subspaces-4 | 58.09 ± 16.78 | [46.08, 70.10] | $p>0.05$ | 85.83 ± 7.61 | [80.39, 91.27] | $p>0.05$ | 13 ± 14 | [3, 23] | $p>0.05$ | 20 ± 14 | [10, 30] | $p>0.05$ |
> | | Expert | 61.82 ± 14.78 | [51.24, 72.39] | $p>0.05$ | 87.85 ± 6.44 | [83.24, 92.45] | $p>0.05$ | 17 ± 13 | [7, 27] | $p>0.05$ | 28 ± 11 | [19, 36] | $p>0.05$ |
> | **Arylation** | Subspaces | 59.38 ± 17.79 | [46.66, 72.11] | $p>0.05$ | 81.28 ± 2.12 | [79.76, 82.80] | $p>0.05$ | 11 ± 13 | [2, 21] | $p>0.05$ | 20 ± 14 | [10, 30] | $p>0.05$ |
> | | Subspaces-1 | 56.87 ± 20.19 | [42.13, 71.02] | $p>0.05$ | 81.40 ± 2.14 | [79.87, 82.93] | $p>0.05$ | 10 ± 9 | [3, 17] | $p>0.05$ | 28 ± 9 | [22, 35] | $p>0.05$ |
> | | Subspaces-2 | 56.87 ± 20.19 | [42.13, 71.02] | $p>0.05$ | 81.40 ± 2.14 | [79.87, 82.93] | $p>0.05$ | 10 ± 9 | [3, 17] | $p>0.05$ | 28 ± 9 | [22, 35] | $p>0.05$ |
> | | Subspaces-3 | 60.58 ± 13.88 | [50.65, 70.51] | $p>0.05$ | 81.25 ± 2.03 | [79.80, 82.70] | $p>0.05$ | 14 ± 9 | [8, 20] | $p>0.05$ | 28 ± 9 | [21, 34] | $p>0.05$ |
> | | Subspaces-4 | 54.12 ± 19.98 | [39.83, 68.42] | $p>0.05$ | 82.14 ± 2.14 | [80.61, 83.67] | $p>0.05$ | 15 ± 12 | [6, 23] | $p>0.05$ | 28 ± 8 | [22, 34] | $p>0.05$ |
> | | Expert | 50.07 ± 19.15 | [36.37, 63.77] | $p>0.05$ | 82.05 ± 2.32 | [80.39, 83.71] | $p>0.05$ | 16 ± 12 | [7, 25] | $p>0.05$ | 26 ± 13 | [16, 35] | $p>0.05$ |
> | **Buchwald_sub1** | Subspaces | 44.08 ± 26.71 | [24.98, 63.19] | $p>0.05$ | 80.25 ± 2.22 | [78.66, 81.83] | $p>0.05$ | 4 ± 2 | [2, 6] | $p>0.05$ | 10 ± 7 | [5, 16] | $p>0.05$ |
> | | Subspaces-1 | 50.39 ± 17.40 | [37.94, 62.83] | $p>0.05$ | 79.01 ± 1.31 | [78.08, 79.95] | $p>0.05$ | 9 ± 8 | [4, 15] | $p<0.05$ | 27 ± 13 | [18, 37] | $p<0.05$ |
> | | Subspaces-2 | 50.39 ± 17.40 | [37.94, 62.83] | $p>0.05$ | 79.01 ± 1.31 | [78.08, 79.95] | $p>0.05$ | 9 ± 8 | [4, 15] | $p<0.05$ | 27 ± 13 | [18, 37] | $p<0.05$ |
> | | Subspaces-3 | 50.39 ± 17.40 | [37.94, 62.83] | $p>0.05$ | 79.01 ± 1.31 | [78.08, 79.95] | $p>0.05$ | 9 ± 8 | [4, 15] | $p<0.05$ | 27 ± 13 | [18, 37] | $p<0.05$ |
> | | Subspaces-4 | 41.35 ± 22.53 | [25.24, 57.47] | $p>0.05$ | 79.52 ± 0.39 | [79.24, 79.80] | $p>0.05$ | 9 ± 9 | [3, 15] | $p>0.05$ | 32 ± 13 | [23, 42] | $p<0.05$ |
> | | Expert | 35.63 ± 28.22 | [15.44, 55.81] | $p>0.05$ | 79.37 ± 1.00 | [78.65, 80.09] | $p>0.05$ | 7 ± 3 | [5, 9] | $p<0.05$ | 23 ± 11 | [15, 31] | $p<0.05$ |
> | **Buchwald_sub2** | Subspaces | 18.19 ± 22.47 | [2.12, 34.26] | $p>0.05$ | 53.23 ± 0.38 | [52.95, 53.50] | $p>0.05$ | 18 ± 9 | [12, 25] | $p>0.05$ | 26 ± 7 | [22, 31] | $p>0.05$ |
> | | Subspaces-1 | 16.56 ± 17.42 | [4.10, 29.02] | $p>0.05$ | 53.72 ± 0.89 | [53.08, 54.36] | $p>0.05$ | 22 ± 7 | [17, 27] | $p>0.05$ | 26 ± 9 | [20, 33] | $p>0.05$ |
> | | Subspaces-2 | 22.52 ± 20.20 | [8.08, 36.97] | $p>0.05$ | 52.67 ± 1.89 | [51.32, 54.03] | $p>0.05$ | 13 ± 10 | [6, 21] | $p>0.05$ | 18 ± 12 | [10, 26] | $p<0.05$ |
> | | Subspaces-3 | 19.12 ± 19.96 | [4.85, 33.40] | $p<0.05$ | 51.62 ± 1.47 | [50.57, 52.67] | $p>0.05$ | 7 ± 6 | [3, 11] | $p<0.05$ | 28 ± 10 | [21, 35] | $p>0.05$ |
> | | Subspaces-4 | 19.12 ± 19.96 | [4.85, 33.40] | $p<0.05$ | 51.62 ± 1.47 | [50.57, 52.67] | $p>0.05$ | 7 ± 6 | [3, 11] | $p<0.05$ | 28 ± 10 | [21, 35] | $p>0.05$ |
> | | Expert | 12.23 ± 16.70 | [0.29, 24.18] | $p>0.05$ | 53.01 ± 0.87 | [52.39, 53.63] | $p>0.05$ | 14 ± 7 | [9, 19] | $p>0.05$ | 30 ± 11 | [22, 39] | $p>0.05$ |

---

> ### Author Response · Authors · 2025-12-03
> **Reply to Reviewer NBfQ - Part 5/9**
>
> **Table R4: Impact of  Labeled Data Volume and Pseudo-data Quality.**
>
> | Dataset | SFT Data Ratio | SFT Data Num. | MAE | $R^2$ | Initial. (Mean ± Std.) | Best Found (Mean ± Std.) | 95% Iter. Round (Mean ± Std.) | Best Iter. Round (Mean ± Std.) |
> | :--- | :---: | :---: | :---: | :---: | :---: | :---: | :---: | :---: |
> | **Suzuki** | 0 | 0 | 40.18 | -2.05 | 44.00 ± 0.00 | 88.18 ± 8.41 | 17 ± 16 | 31 ± 8 |
> | | 0.02% | 1 | 26.04 | -0.53 | 21.55 ± 0.00 | 87.41 ± 8.45 | 20 ± 15 | 27 ± 11 |
> | | 0.10% | 6 | 27.51 | -0.69 | 37.34 ± 0.00 | 85.38 ± 6.41 | 15 ± 11 | 25 ± 9 |
> | | 0.25% | 14 | 27.8 | -0.53 | 76.01 ± 0.00 | 81.17 ± 3.84 | 5 ± 12 | 14 ± 13 |
> | | 0.50% | 29 | 21.13 | 0.02 | 92.06 ± 0.00 | 92.06 ± 0.00 | 1 ± 0 | 1 ± 0 |
> | | 1.00% | 50 | 19.47 | 0.2 | 92.24 ± 0.00 | 96.15 ± 0.00 | 1 ± 0 | 3 ± 0 |
> | | 2.00% | 115 | 15.92 | 0.39 | 92.24 ± 0.00 | 93.41 ± 1.89 | 1 ± 0 | 13 ± 19 |
> | | 4.00% | 230 | 13.44 | 0.54 | 92.24 ± 0.00 | 92.24 ± 0.00 | 1 ± 0 | 1 ± 0 |
> | | 8.00% | 461 | 10.77 | 0.68 | 74.96 ± 0.00 | 91.89 ± 0.52 | 2 ± 0 | 24 ± 10 |
> | | 16.00% | 922 | 8.23 | 0.79 | 88.80 ± 0.00 | 92.24 ± 0.00 | 1 ± 0 | 2 ± 0 |
> | | 32.00% | 1844 | 5.46 | 0.89 | 88.80 ± 0.00 | 96.15 ± 0.00 | 2 ± 0 | 2 ± 0 |
> | **Arylation** | 0 | 0 | 33.24 | -1.49 | 65.86 ± 0.00 | 82.20 ± 1.38 | 11 ± 6 | 22 ± 14 |
> | | 0.02% | 1 | 25.33 | -0.19 | 0.00 ± 0.00 | 80.58 ± 2.09 | 10 ± 6 | 16 ± 9 |
> | | 0.10% | 4 | 31.57 | -0.79 | 32.41 ± 0.00 | 81.50 ± 2.35 | 16 ± 11 | 29 ± 9 |
> | | 0.25% | 10 | 24.22 | -0.07 | 28.33 ± 0.00 | 80.94 ± 1.65 | 3 ± 1 | 21 ± 12 |
> | | 0.50% | 20 | 26.53 | -0.36 | 76.34 ± 0.00 | 81.70 ± 1.41 | 2 ± 1 | 9 ± 9 |
> | | 1.00% | 34 | 19.55 | 0.13 | 82.63 ± 0.00 | 82.83 ± 0.64 | 1 ± 0 | 4 ± 10 |
> | | 2.00% | 79 | 15.75 | 0.38 | 82.57 ± 0.00 | 82.98 ± 0.53 | 1 ± 0 | 13 ± 16 |
> | | 4.00% | 158 | 11.97 | 0.62 | 76.95 ± 0.00 | 83.60 ± 0.00 | 2 ± 0 | 14 ± 9 |
> | | 8.00% | 316 | 8.49 | 0.77 | 82.57 ± 0.00 | 83.60 ± 0.00 | 1 ± 0 | 13 ± 3 |
> | | 16.00% | 633 | 6.39 | 0.85 | 77.47 ± 0.00 | 83.29 ± 0.70 | 4 ± 5 | 26 ± 13 |
> | | 32.00% | 1266 | 3.65 | 0.95 | 82.57 ± 0.00 | 83.60 ± 0.00 | 1 ± 0 | 11 ± 2 |
> | **Buchwald_sub1** | 0 | 0 | 52.28 | -3.64 | 77.63 ± 0.00 | 79.83 ± 0.38 | 1 ± 0 | 27 ± 11 |
> | | 0.02% | 1 | 49.18 | -3.2 | 37.12 ± 0.00 | 79.93 ± 0.53 | 2 ± 0 | 16 ± 6 |
> | | 0.10% | 4 | 47.77 | -3.02 | 66.43 ± 0.00 | 80.11 ± 0.55 | 2 ± 0 | 13 ± 13 |
> | | 0.25% | 10 | 22.9 | 0.07 | 75.55 ± 0.00 | 79.60 ± 0.28 | 4 ± 2 | 22 ± 12 |
> | | 0.50% | 20 | 22.72 | 0 | 28.40 ± 0.00 | 79.91 ± 0.53 | 2 ± 0 | 27 ± 8 |
> | | 1.00% | 34 | 21.65 | 0.09 | 75.55 ± 0.00 | 79.97 ± 0.12 | 4 ± 1 | 23 ± 13 |
> | | 2.00% | 79 | 16.9 | 0.4 | 65.59 ± 0.00 | 79.73 ± 0.08 | 2 ± 0 | 22 ± 13 |
> | | 4.00% | 158 | 13.16 | 0.61 | 79.53 ± 0.00 | 80.45 ± 0.59 | 3 ± 0 | 10 ± 7 |
> | | 8.00% | 316 | 8.99 | 0.78 | 79.08 ± 0.00 | 80.68 ± 0.48 | 1 ± 0 | 13 ± 9 |
> | | 16.00% | 633 | 6.4 | 0.88 | 79.08 ± 0.00 | 80.11 ± 0.55 | 3 ± 0 | 8 ± 5 |
> | | 32.00% | 1266 | 3.59 | 0.96 | 79.53 ± 0.00 | 80.44 ± 0.60 | 3 ± 0 | 20 ± 8 |
> | **Buchwald_sub2** | 0 | 0 | 20.16 | -1 | 46.94 ± 0.00 | 53.69 ± 2.19 | 16 ± 12 | 23 ± 15 |
> | | 0.02% | 1 | 19 | -0.77 | 0.00 ± 0.00 | 52.52 ± 2.35 | 6 ± 3 | 14 ± 12 |
> | | 0.10% | 4 | 18.79 | -0.66 | 14.90 ± 0.00 | 55.57 ± 2.24 | 33 ± 13 | 36 ± 9 |
> | | 0.25% | 10 | 23.07 | -0.76 | 35.26 ± 0.00 | 52.77 ± 1.96 | 5 ± 2 | 14 ± 12 |
> | | 0.50% | 20 | 16.02 | -0.01 | 43.09 ± 0.00 | 53.34 ± 0.77 | 11 ± 11 | 28 ± 8 |
> | | 1.00% | 34 | 12.15 | 0.39 | 53.33 ± 0.00 | 56.81 ± 0.00 | 2 ± 0 | 2 ± 0 |
> | | 2.00% | 79 | 9.76 | 0.51 | 53.10 ± 0.00 | 53.94 ± 1.11 | 5 ± 12 | 18 ± 14 |
> | | 4.00% | 158 | 8.7 | 0.56 | 52.01 ± 0.00 | 53.83 ± 2.10 | 10 ± 15 | 18 ± 14 |
> | | 8.00% | 316 | 5.26 | 0.81 | 50.21 ± 0.00 | 55.21 ± 1.70 | 8 ± 7 | 17 ± 11 |
> | | 16.00% | 633 | 4.06 | 0.85 | 53.33 ± 0.00 | 54.78 ± 1.47 | 8 ± 13 | 14 ± 15 |
> | | 32.00% | 1266 | 2.61 | 0.92 | 53.33 ± 0.00 | 54.72 ± 1.16 | 2 ± 2 | 10 ± 11 |

---

> ### Author Response · Authors · 2025-12-03
> **Reply to Reviewer NBfQ - Part 6/9**
>
> **Q2: Safety and feasibility constraints. The current loop does not appear to enforce domain safety or feasibility filters to prevent proposing hazardous or impractical conditions. Incorporating rule-based checks, surrogate predictive models of feasibility, or expert-validated constraints would improve readiness for autonomous labs.**
>
> **A2:** We fully agree that safety and feasibility are critical for real-world autonomous labs. In the present study, we do not incorporate an explicit hazard/feasibility model inside the ChemBOMAS. Instead, safety is handled in two more conservative ways:
>
> 1. For all dry-lab benchmarks, the search spaces are derived from published real high-throughput reaction datasets, where reagents, additives, and condition ranges are already experimentally validated and routinely used, so the space is implicitly restricted to feasible laboratory conditions.
>
> 2. For the wet-lab experiment, the design space (≈6-dimensional variables, ~10⁵ combinations) was pre-defined by chemists to exclude obviously hazardous or incompatible settings (e.g., unsafe temperatures, known incompatible solvents, forbidden additives). All suggested conditions were reviewed by human experts under standard safety protocols.
>
> **Q3: Pseudo-label calibration and bias. The LLM regressor can introduce bias if mis-calibrated. There is limited analysis of reliability, label-noise sensitivity, or robustness when pseudo-data conflict with early measurements.**
>
> **A3:** We thank the reviewer for raising this critical point regarding pseudo-label reliability. We have systematically evaluated pseudo-data quality and its impact on the BO process through experiments across all benchmarks (**see Table R4**). Our findings demonstrate the robustness of our approach:
>
> 1. BO Robustness to Noise and Data Scarcity: Experiments show that only extremely low-quality pseudo-labels (no SFT) significantly negatively affect optimization performance. Even moderately accurate pseudo-data, readily obtained with minimal (0.25%-0.5%) fine-tuning, can drive optimization performance close to the optimum.
>
> 2. Conflict Resolution Mechanism: Importantly, our algorithm is designed to prioritize real measurements. Whenever real data becomes available, the corresponding pseudo-labels are automatically discarded. Furthermore, pseudo-labels are progressively down-weighted and pruned as real observations accumulate. These ensure that conflicting or excessive pseudo-labels do not mislead the optimization process.
>
> **Q4: RAG partitioning auditability. The retrieval corpus, filtering criteria, and sensitivity of subspace splits to retrieval noise are under-documented. Without this, it is hard to assess whether partitions encode prior bias that drives observed gains.**
>
> **A4:** We agree that the RAG partitioning must be auditable. Below, we clarify:
>
> **(i) How to implement RAG through corpus retrieval and filtering criteria?**
>
> Our "Hybrid RAG" architecture operates as a three-tier, prioritized pipeline to retrieve comprehensive chemical knowledge.
> 1.  **Tier 1: Literature Extraction (High Priority):** The process begins by querying a curated local repository of academic literature using chemical reaction-type keywords to retrieve top-k passages of unstructured text that encapsulate expert knowledge on mechanisms and properties.
> 2.  **Tier 2: Database Search (Intermediate Priority):** If literature extraction yields insufficient context, the RAG pipeline will supplement the chemical information by querying structured databases, such as RDKit and PubChem, using exact identifiers (SMILES or IUPAC names). These domain databases store diverse attributes, such as molecular fingerprints and physicochemical descriptors.
> 3.  **Tier 3: Web Search (Low Priority):** When both the literature and structured databases lack relevant information, a web search API will be used as a constrained fallback to obtain short, factual descriptions.
>
> **(ii) What is the sensitivity of subspace splits to retrieval noise?**
>
> To evaluate the impact of RAG's partition variations introduced by LLMs' generative character, we conducted two additional analyses.
> 1.  We ran the RAG module five times and found that while partition boundaries show slight variations, the underlying chemical logic remains highly consistent (detailed results in Appendix J.1.2).
> 2.  In order to avoid the influence of pseudo-data, which may cover up the negative impact of the partition variation, we evaluated the optimization performance of ChemBOAMS without the data module. The results (see Table R3) confirmed that the BO performance is robust to these partition variations, with different metrics remaining largely unaffected across 5 × 10 runs and being similar or better than the expert partition. These results demonstrate that our knowledge-driven strategy delivers reliable and consistent outcomes.

---

> ### Author Response · Authors · 2025-12-03
> **Reply to Reviewer NBfQ - Part 7/9**
>
> **Q5: Baseline alignment and coverage. While strong baselines are included, details on kernel choices, acquisition settings, restart budgets, batch sizes, and runtime are scattered. Broader comparisons to recent hybrid methods and a clearer accounting of computational cost would strengthen fairness and scope.**
>
> **A5:** Thanks for your suggestion. We have added Table R5 to report the detailed settings and computational cost for each baseline. Table R3 highlights two key facts:
> 1. All baselines are run under a unified experimental setting to ensure strict fairness.
> 2. ChemBOMAS achieves a superior trade-off between optimization performance and runtime. ChemBOMAS requires a similar time as Standard BO and is significantly faster than API-based LLM methods like BO-ICL. Although lightweight methods, such as LA-MCTS and GoLLum, are slightly quicker, ChemBOMAS consistently and substantially outperforms both across all four benchmark datasets.
>
> **Table R5: Detailed Experimental Settings and Computational Cost.**
>
> | Item | BO | BO-ICL | GoLLum | LA-MCTS | ChemBOMAS |
> | :--- | :---: | :---: | :---: | :---: | :---: |
> | **Initialization** | 1% of dataset | 1% of dataset | 1% of dataset | 1% of dataset | 1% of dataset |
> | **BO Batch Size** | 0.1% of dataset | 0.1% of dataset | 0.1% of dataset | 0.1% of dataset | 0.1% of dataset |
> | **BO Acq. Function** | Expected Improvement | Expected Improvement | Expected Improvement | Expected Improvement | Expected Improvement |
> | **Iterations** | 40 | 40 | 40 | 40 | 40 |
> | **Repeat Campaigns** | 10 | 10 | 10 | 10 | 10 |
> | **Kernel Function** | Matérn| Matérn| Matérn | Matérn| Matérn|
> | **Comp. Cost (s): Suzuki** | 144 | 19 | 22 | 2575 | 212 |
> | **Comp. Cost (s): Arylation** | 88 | 20 | 21 | 1957 | 100 |
> | **Comp. Cost (s): Buchwald-sub1** | 21 | 22 | 13 | 1734 | 28 |
> | **Comp. Cost (s): Buchwald-sub2** | 17 | 19 | 13 | 1767 | 22 |
>
> **Q6: Generality claims. The materials-style task is promising but under-described. More context on its construction and accessibility would help calibrate the claim of cross-domain applicability.**
>
> **A6:** Thanks for your suggestions. In the response, we have added a comprehensive description of the material-style task.
>
> The lipid nanoparticle (LNP3) dataset originates from nanomedicine and focuses on optimizing LNP formulations for effective cannabidiol delivery. It contains 768 unique experiments defined over a 5-dimensional parameter space, including the type and amount of solid lipids, liquid lipids, and surfactants. While LNP formulations exhibit multiple properties, in this study, we isolate the specific objective of maximizing encapsulation efficiency. Moreover, the dataset is derived from the established Olympus benchmark suite in [https://github.com/aspuru-guzik-group/olympus][1-2], ensuring accessibility and reproducibility.
> We also supplemented a detailed context on the LNP3 datasets in Appendix K.
>
> **Q7: Can you provide means, standard deviations, confidence intervals, and significance tests for all benchmarks and wet-lab studies?**
>
> **A7:** We thank the reviewer for the request for comprehensive statistical reporting. In response, we have rerun all key benchmark experiments with ten independent random seeds. The updated results now include the mean, standard deviation, 95% confidence intervals, and paired t-tests for all baseline comparisons, ablation studies, RAG sensitivity analyses, and pseudo-data experiments, as detailed in **Table R1-R4**.
>
> Regarding the wet-lab study, we must clarify a methodological distinction: it represents a single, real-world optimization campaign conducted under practical constraints, not a replicated benchmark. Performing multiple independent wet-lab campaigns would be prohibitively expensive and time-consuming, and was not feasible within the revision timeline. Therefore, we present this result as a proof-of-concept demonstration of the method's real-world applicability and scalability, acknowledging its nature as a case study rather than a statistically replicated experiment.
>
> **Q8: How sensitive is the performance of the Bayesian optimization loop to noise or mis-calibration of the LLM-generated pseudo-labels?**
>
> **A8:** This is an important issue, and we are keen to investigate it. Hence, we have systematically evaluated pseudo-data quality and its impact on the BO process across ten independent experiments per benchmark (**see Table R4**).  Experiments show that only extremely low-quality pseudo-labels (no SFT) significantly negatively affect optimization performance. Even moderately accurate pseudo-data, readily obtained with minimal (0.25%-0.5%) fine-tuning, can drive optimization performance close to the optimum. Furthermore, using higher-quality pseudo-labels does not yield significant gains, further underscoring its ability to extract maximum utility from minimal inputs.

---

> ### Author Response · Authors · 2025-12-03
> **Reply to Reviewer NBfQ - Part 8/9**
>
> **Q9: Is the LLM regressor kept fixed after initial fine-tuning or updated online,**
>
> **A9:** We thank the reviewer for this question regarding the LLM regressor's updating strategy. The implementation differs between our dry-lab benchmarks and the wet-lab experiment:
>
> 1. **Dry-lab Benchmarks: Keep Fixed.** In all four benchmark datasets (Suzuki, Arylation, Buchwaldsub, LNP3), the LLM regressor is fine-tuned once on the initial 1% of labeled data and then kept fixed throughout the entire Bayesian Optimization process. We found that updating the regressor with additional data acquired during BO yielded negligible improvements in final optimization performance (as supported by Table R1). Therefore, to maintain computational efficiency, we employed a fixed regressor in these dry-lab experiments, which also completely eliminates any risk of online self-reinforcement from pseudo-labels.
>
> 2. **Wet-lab Experiment: Online Updates.** For the previously unreported wet-lab reaction, we employed an online adaptation scheme since no pre-existing labeled dataset was available. Firstly, the initial experimental round was designed purely by the knowledge-driven module without any prior wet-lab data. Then, after each round of experiments, the LLM regressor was updated on the newly acquired ground-truth experimental observations. Crucially, pseudo-labels were never used as training data for the regressor. The 2–3 day feedback cycle in wet-lab experiments provided sufficient time for model retraining without delaying experimental progress, making this online update strategy both feasible and beneficial.
>
> **Q10 and what safeguards prevent confirmation bias in that process?**
>
> **A10:** We have explicitly designed three mechanisms to mitigate the risk of "confirmation bias":
> 1. **Dual-Stage Pseudo-Data Pruning (Appendix C):** We employ a dual refinement strategy to prevent pseudo-data from dominating as real observations accumulate. First, we perform local similarity pruning: when a new real data point is obtained, we remove pseudo-points that are too similar in the LLM embedding space, preventing biased early predictions from being over-represented in regions with real feedback. Second, we implement weighted global removal: as Bayesian Optimization progresses, pseudo-points with very high predicted values are discarded with higher probability. This prevents the optimizer from being trapped in a small set of self-reinforced "high-value" regions. As more real data is collected, the number of discarded pseudo-points increases, thereby reducing noise in the surrogate model.
>
> 2. **The knowledge-driven module (RAG+LLM) operates independently from the regressor.** It provides a fixed, initial structural partitioning of the search space. This partition acts as a constraint, preventing the optimizer from being globally misled by noisy or over-optimistic pseudo-data, as search is confined and compared within meaningful chemical subspaces.
>
> 3. **Explicit Exploration via Acquisition Function:** Both the subspace selection (using UCB on the tree) and the standard BO acquisition function incorporate the hyperparameters to control the exploration bias. Even if pseudo-labels are over-optimistic in certain regions, the exploration component forces the algorithm to test under-explored subspaces and variables periodically.
>
> Together, these safeguards ensure that while the regressor learns from new experimental data, the overall optimization process remains robust against reinforcing its own early errors.
>
> **Q11: ... and how often do retrieval errors lead to sub-optimal subspace splits?**
>
> **A11:** The results (see Table R3) confirmed that the BO performance is robust to these partition variations, with 80% of metric statistics remaining unaffected (p-value > 0.05) in eighty significant tests. These results demonstrate that our knowledge-driven strategy delivers reliable and consistent outcomes.
>
> **Q12: Are safety or feasibility constraints (e.g., chemical hazard filters) integrated into the proposal generation loop, and if not, how might this affect deployment in autonomous labs?**
>
> **A12:** Thanks for your question. The safety constraint mechanism is not directly integrated into the ChemBOMAS algorithm framework. As an important direction for future work, we plan to explicitly integrate safety into the ChemBOMAS pipeline for autonomous labs. Concretely, we will first extend the RAG corpus with safety datasheets (e.g., MSDS) and operating guidelines, enabling LLM-based ranking and partitioning variables to incorporate safety annotations. Then, the subspaces that satisfy these safety constraints will be selected to construct the search tree, so unsafe regions are never passed to BO. In this way, safety and feasibility filters can be layered on top of ChemBOMAS without changing its core structure, facilitating a realistic path toward deployment in autonomous labs.

---

> ### Author Response · Authors · 2025-12-03
> **Reply to Reviewer NBfQ - Part 9/9**
>
> **Q13: For each reaction dataset and pre-trained component, has overlap between pre-training data and evaluation sets been checked to rule out data leakage?**
>
> **A13:** We thank you for your concern regarding data leakage and have checked the datasets to rule it out. Importantly, to lend the credibility of data security, we added a comparative experiment between:
>
> 1. Pre-train only: the LLM regressor is used without supervised fine-tuning across four benchmarks, and BO is run on top of this frozen regression model.
>
> 2. Pre-train + SFT: the same pre-trained model is then fine-tuned on a minimal amount of labeled data from each benchmark and used within ChemBOMAS.
>
> The results (Table R6) show that:
>
> 1. The pre-train-only regressor has poor predictive performance (negative $R^2$) and leads to substantially worse BO outcomes.
>
> 2. The pre-train + SFT regressor achieves significantly higher regression accuracy and significantly better BO performance.
>
> If there were substantial data leakage, we would expect comparable regression and optimization performance between the two settings; however, we do not observe the similarities, which strongly demonstrates that there is no leakage between the pre-training dataset and all benchmarks.
>
> **Table R6: Data Leakage Check.**
>
> | Dataset | Setting | SFT Data Ratio | SFT Data Num. | MAE | $R^2$ | Initial. (Mean ± Std.) | Best Found (Mean ± Std.) | 95% Iter. Round (Mean ± Std.) | Best Iter. Round (Mean ± Std.) |
> | :--- | :--- | :---: | :---: | :---: | :---: | :---: | :---: | :---: | :---: |
> | **Suzuki** | Pre-train only | 0 | 0 | 40.18 | -2.05 | 44.00 ± 0.00 | 88.18 ± 8.41 | 17 ± 16 | 31 ± 8 |
> | | Pre-train + SFT | 1.00% | 50 | 19.47 | 0.2 | 92.24 ± 0.00 | 96.15 ± 0.00 | 1 ± 0 | 3 ± 0 |
> | **Arylation** | Pre-train only | 0 | 0 | 33.24 | -1.49 | 65.86 ± 0.00 | 82.20 ± 1.38 | 11 ± 6 | 22 ± 14 |
> | | Pre-train + SFT | 1.00% | 34 | 19.55 | 0.13 | 82.63 ± 0.00 | 82.83 ± 0.64 | 1 ± 0 | 4 ± 10 |
> | **Buchwald_sub1** | Pre-train only | 0 | 0 | 52.28 | -3.64 | 77.63 ± 0.00 | 79.83 ± 0.38 | 1 ± 0 | 27 ± 11 |
> | | Pre-train + SFT | 4.00% | 158 | 13.16 | 0.61 | 79.53 ± 0.00 | 80.45 ± 0.59 | 3 ± 0 | 10 ± 7 |
> | **Buchwald_sub2** | Pre-train only | 0 | 0 | 20.16 | -1 | 46.94 ± 0.00 | 53.69 ± 2.19 | 16 ± 12 | 23 ± 15 |
> | | Pre-train + SFT | 1.00% | 34 | 12.15 | 0.39 | 53.33 ± 0.00 | 56.81 ± 0.00 | 2 ± 0 | 2 ± 0 |
>
> **Q14: Could you clarify the materials-style “LNP3” task: define its dataset, availability, and whether a minimal open-version will be released for benchmarking?**
>
> **A14:** Thanks for your suggestions. In the response, we have added a comprehensive description and Table RX of the material-style task.
>
> The lipid nanoparticle (LNP3) dataset originates from nanomedicine and focuses on optimizing LNP formulations for effective cannabidiol delivery. It contains 768 unique experiments defined over a 5-dimensional parameter space, including the type and amount of solid lipids, liquid lipids, and surfactants. While LNP formulations exhibit multiple properties, in this study, we isolate the specific objective of maximizing encapsulation efficiency. Moreover, the dataset is derived from the established Olympus benchmark suite in [https://github.com/aspuru-guzik-group/olympus][1-2], ensuring accessibility and reproducibility.
> We also added detailed context on the LNP3 datasets in Appendix K.
>
> **Reference**
>
> [1] Häse F, Aldeghi M, Hickman R J, et al. Olympus: a benchmarking framework for noisy optimization and experiment planning[J]. Machine Learning: Science and Technology, 2021, 2(3): 035021.
>
> [2] Hickman R, Parakh P, Cheng A, et al. Olympus, enhanced: benchmarking mixed-parameter and multi-objective optimization in chemistry and materials science[J]. 2023.
>
> **We greatly appreciate your comprehensive feedback. We hope that our responses have satisfactorily addressed all your questions.**

---

### Official Review · Reviewer_brq8 · 2025-11-01

**Soundness:** 3
**Presentation:** 4
**Contribution:** 3
**Rating:** 8
**Confidence:** 3

**Summary:**

This paper presents a well-designed and innovative pipeline, combining LLM and Bayesian optimization, for chemical optimization.

**Strengths:**

- The paper is very well written and structure. The flow chart cleanly demonstrates the design of the algorithm while the results are also clearly communicated.
- It appears that the resulting language model si very capable of chemical optimization.
- The optimization trajectories showcase the potential utility of this model in real-world chemical optimization tasks.
- The problem addressed here is exciting and topical.

**Weaknesses:**

- The design space of BO, in terms of acquisition functions and policies, is explored insufficiently. I also understand that this is not the focus of this paper.
- In the beginning paragraph, the aim of the paper seems to be addressing chemical discovery in general, but only synthesis is considered. I would tune down the claimed scope.
- It seems that the performance gap between the proposed method and LLMs trained on labeled data is not too huge.

**Questions:**

- Have you considered generalizing this method to other modalities of chemical discovery, such as prediction tasks such as physical property predictions?

---

> ### Author Response · Authors · 2025-12-03
> **Reply to Reviewer brq8 - Part 1/2**
>
> We greatly appreciate your recognition and suggestions regarding our work. These suggestions can enhance the completeness and presentation quality of our work.
>
> **Q1: The design space of BO, in terms of acquisition functions and policies, is explored insufficiently. I also understand that this is not the focus of this paper.**
>
> **A1:** Thank you for this comment. Although the main focus in this work is not on the acquisition-function design of BO, in response to your suggestion we have conducted an additional comparison of four standard acquisition functions and policies—Expected Improvement (EI, used in the work), Probability of Improvement (PI), Max-Value Entropy (MVE), and Upper Confidence Bound (UCB)— across 10 independent runs per benchmark (See Table R1). The results show that the optimization performance metrics under EI, PI, MVE, and UCB are very similar and exhibit no statistically significant differences. Hence, using EI in this work is representative rather than cherry-picked.
>
> **Table R1 Acquisition Function Comparison.**
> | Dataset | Acquisition Function | Best Found (Mean ± Std.) | Best Found (p-val) | 95% Iter. Round (Mean ± Std) | 95% Iter. Round (p-val) | Best Iter. Round (Mean ± Std) | Best Iter. Round (p-val) |
> | :--- | :---: | :---: | :---: | :---: | :---: | :---: | :---: |
> | Suzuki | EI (this work) | 91.45 ± 7.58 | - | 12 ± 10 | - | 16 ± 7 | - |
> | | MVE | 91.35 ± 7.73 | p > 0.05 | 15 ± 13 | p > 0.05 | 24 ± 15 | p > 0.05 |
> | | PI | 92.99 ± 6.67 | p > 0.05 | 16 ± 9 | p > 0.05 | 21 ± 11 | p > 0.05 |
> | | UCB | 94.61 ± 4.88 | p > 0.05 | 16 ± 11 | p > 0.05 | 16 ± 11 | p > 0.05 |
> | Arylation | EI (this work) | 82.83 ± 1.77 | - | 12 ± 9 | - | 27 ± 8 | - |
> | | MVE | 82.40 ± 2.33 | p > 0.05 | 14 ± 12 | p > 0.05 | 30 ± 9 | p > 0.05 |
> | | PI | 82.99 ± 2.02 | p > 0.05 | 16 ± 14 | p > 0.05 | 26 ± 11 | p > 0.05 |
> | | UCB | 83.55 ± 2.06 | p > 0.05 | 21 ± 11 | p > 0.05 | 28 ± 5 | p > 0.05 |
> | Buchwald_sub1 | EI (this work) | 79.74 ± 0.42 | - | 5 ± 2 | - | 12 ± 10 | - |
> | | MVE | 79.39 ± 1.35 | p > 0.05 | 4 ± 3 | p > 0.05 | 12 ± 9 | p > 0.05 |
> | | PI | 79.78 ± 0.41 | p > 0.05 | 5 ± 3 | p > 0.05 | 12 ± 13 | p > 0.05 |
> | | UCB | 79.83 ± 0.39 | p > 0.05 | 6 ± 4 | p > 0.05 | 17 ± 16 | p > 0.05 |
> | Buchwald_sub2 | EI (this work) | 56.61 ± 0.61 | - | 20 ± 9 | - | 22 ± 9 | - |
> | | MVE | 56.81 ± 0.00 | p > 0.05 | 16 ± 6 | p > 0.05 | 25 ± 4 | p > 0.05 |
> | | PI | 55.77 ± 1.66 | p > 0.05 | 26 ± 11 | p > 0.05 | 29 ± 9 | p > 0.05 |
> | | UCB | 54.07 ± 0.58 | p > 0.05 | 13 ± 8 | p < 0.05 | 22 ± 11 | p > 0.05 |
>
>
> **Q2: In the beginning paragraph, the aim of the paper seems to be addressing chemical discovery in general, but only synthesis is considered. I would tune down the claimed scope.**
>
> **A2:** We thank the reviewer for this valuable observation. Following this suggestion, we have revised the Introduction in the manuscript to more precisely reflect our contribution. The opening paragraph has been updated to specifically frame the work as “accelerating Bayesian optimization for reaction condition optimization in synthetic chemistry.”
>
> Furthermore, to demonstrate the potential for broader applicability, we have conducted an initial extension beyond synthesis. As detailed in Appendix K, we evaluated ChemBOMAS on the LNP3 materials science dataset, where the objective is a physical property related to lipid-nanoparticle formulation quality rather than a reaction yield. The results show that ChemBOMAS matches or outperforms strong baselines and converges more efficiently, suggesting the framework's potential utility in non-reaction optimization domains as well.

---

> ### Author Response · Authors · 2025-12-03
> **Reply to Reviewer brq8 - Part 2/2**
>
> **Q3: It seems that the performance gap between the proposed method and LLMs trained on labeled data is not too huge.**
>
> **A3:** Thank you for the observation. We would like to clarify that even apparently modest improvements in MSE/MAE and ${R^2}$ correspond to substantial gains in downstream BO performance, particularly when the baseline ${R^2}$ is negative or near-zero. To support our conclusion, we have added Table R2, which systematically studies the effect of varying regression quality on BO across the four benchmarks. Specifically, we fine-tune the LLM regressors with different amounts of labeled data, thereby inducing a range of ${R^2}$ values. Table R2 shows that increasing the regressor’s ${R^2}$ to around 0.2 across all evaluation datasets is sufficient for ChemBOMAS to achieve robust and superior optimization performance.
>
> **Table R2: Impact of Regressor Quality on Optimization.**
>
> | Dataset | SFT Data Ratio | MAE | $R^2$ | Initial. (Mean ± Std.) | Best Found (Mean ± Std.) | 95% Iter. Round (Mean ± Std.) | Best Iter. Round (Mean ± Std.) |
> | :--- | :---: | :---: | :---: | :---: | :---: | :---: | :---: |
> | **Suzuki** | 0.10% | 27.51 | -0.69 | 37.34 ± 0.00 | 85.38 ± 6.41 | 15 ± 11 | 25 ± 9 |
> | | 0.02% | 26.04 | -0.53 | 21.55 ± 0.00 | 87.41 ± 8.45 | 20 ± 15 | 27 ± 11 |
> | | 0.50% | 21.13 | 0.02 | 92.06 ± 0.00 | 92.06 ± 0.00 | 1 ± 0 | 1 ± 0 |
> | | 1.00% | 19.47 | 0.2 | 92.24 ± 0.00 | 96.15 ± 0.00 | 1 ± 0 | 3 ± 0 |
> | **Arylation** | / | 35.18 | -1.48 | 35.70 ± 0.00 | 79.56 ± 0.42 | 2 ± 0 | 20 ± 9 |
> | | 0.10% | 31.57 | -0.79 | 32.41 ± 0.00 | 81.50 ± 2.35 | 16 ± 11 | 29 ± 9 |
> | | 0.25% | 24.22 | -0.07 | 28.33 ± 0.00 | 80.94 ± 1.65 | 3 ± 1 | 21 ± 12 |
> | | 1.00% | 19.55 | 0.13 | 82.63 ± 0.00 | 82.83 ± 0.64 | 1 ± 0 | 4 ± 10 |
> | **Buchwald_sub1** | / | 33.08 | -1.14 | 32.17 ± 0.00 | 79.57 ± 0.20 | 5 ± 2 | 26 ± 12 |
> | | 0.50% | 22.72 | 0 | 28.40 ± 0.00 | 79.91 ± 0.53 | 2 ± 0 | 27 ± 8 |
> | | 0.25% | 22.9 | 0.07 | 75.55 ± 0.00 | 79.60 ± 0.28 | 4 ± 2 | 22 ± 12 |
> | | 1.00% | 21.65 | 0.09 | 75.55 ± 0.00 | 79.97 ± 0.12 | 4 ± 1 | 23 ± 13 |
> | **Buchwald_sub2** | 0.02% | 19 | -0.77 | 0.00 ± 0.00 | 52.52 ± 2.35 | 6 ± 3 | 14 ± 12 |
> | | 0.25% | 23.07 | -0.76 | 35.26 ± 0.00 | 52.77 ± 1.96 | 5 ± 2 | 14 ± 12 |
> | | 0.50% | 16.02 | -0.01 | 43.09 ± 0.00 | 53.34 ± 0.77 | 11 ± 11 | 28 ± 8 |
> | | 1.00% | 12.15 | 0.39 | 53.33 ± 0.00 | 56.81 ± 0.00 | 2 ± 0 | 2 ± 0 |
>
> **Q4: Have you considered generalizing this method to other modalities of chemical discovery, such as prediction tasks such as physical property predictions?**
>
> **A4:** We thank the reviewer for this thoughtful question. The potential for generalization to other domains of chemical discovery is indeed an important aspect of our work, and we have taken preliminary steps to explore this direction.
>
> ChemBOMAS is inherently a modality-agnostic framework. It operates on a general design space $X$ and a scalar objective function $h(x)$ that can be experimentally or computationally evaluated. While our primary application in this work is reaction yield optimization, the algorithmic framework is equally applicable to other scalar properties of interest in chemistry and materials science, such as solubility, stability, toxicity, or electronic properties. This generalization requires only that (1) the data-driven module's regressor is trained to predict the target property, and (2) the knowledge-driven module (or a expert-supplied alternative) can meaningfully partition the relevant variable space.
>
> As an initial validation of this generalizability, we have applied ChemBOMAS to the LNP3 materials science dataset (detailed in Appendix K). In this task, the objective is a physical property related to lipid-nanoparticle formulation quality, rather than a synthetic yield. The results show that ChemBOMAS matches or surpasses strong baselines in finding the best objective value and achieves convergence in fewer iterations (see Table 13). This demonstrates the framework's promising potential for extension to non-reaction optimization tasks in chemical discovery.
>
> **We hope that our responses have satisfactorily addressed all your questions. Thank you again for your recommendation.**

---

### Official Review · Reviewer_BP3g · 2025-11-01

**Soundness:** 2
**Presentation:** 2
**Contribution:** 2
**Rating:** 2
**Confidence:** 4

**Summary:**

This paper presents ChemBOMAS, a multi-agent system using Large Language Models (LLMs) to accelerate Bayesian optimization (BO) for scientific discovery in chemistry. It addresses the common BO challenges of sparse experimental data and vast search spaces. The system combines two strategies: a data-driven approach, using an LLM to generate pseudo-data for robust initialization , and a knowledge-driven approach, using an LLM to intelligently partition the search space, allowing BO to focus on high-potential areas.

**Strengths:**

It proposed a novel multi-agent system-based Bayesian optimization method.

**Weaknesses:**

**Lack of LLM regression baseline**

The paper proposed their LLM regression model, but the experimental evaluation is insufficient due to a lack of comprehensive baselines[8~9]. It is difficult to assess the proposed method's true contribution without comparing it against a wider range of existing or even simpler alternative methods.

**About Optimization results**

The results in Figure 2 are concerning and require clarification.
It is unclear why the objective values differ at iteration 0 for the various methods being compared. For a fair comparison, all methods should presumably start from an equivalent initial state.
Following this, the optimization curves for the proposed method appear almost entirely flat after the initial iteration. This suggests that minimal, if any, optimization or "search" is occurring. This raises a critical question: Is the method's performance simply an artifact of finding a good initial data point, rather than a demonstration of an effective optimization process?

**Lack of optimization baselines**

A notable omission is the comparison against LBO (Latent Bayesian Optimization)[1~7]. Given the nature of the problem, LBO methods seem like a highly relevant and potentially strong baseline. The authors should justify this exclusion or provide results for this comparison.

**Reference**

[1] Tripp, Austin, Erik Daxberger, and José Miguel Hernández-Lobato. "Sample-efficient optimization in the latent space of deep generative models via weighted retraining." Advances in Neural Information Processing Systems 33 (2020): 11259-11272.

[2] Maus, Natalie, et al. "Local latent space bayesian optimization over structured inputs." Advances in neural information processing systems 35 (2022): 34505-34518.

[3] Lee, Seunghun, et al. "Advancing bayesian optimization via learning correlated latent space." Advances in Neural Information Processing Systems 36 (2023): 48906-48917.

[4] Chu, Jaewon, et al. "Inversion-based latent bayesian optimization." Advances in Neural Information Processing Systems 37 (2024): 68258-68286.

[5] Moss, Henry B., Sebastian W. Ober, and Tom Diethe. "Return of the latent space COWBOYS: Re-thinking the use of VAEs for Bayesian optimisation of structured spaces." arXiv preprint arXiv:2507.03910 (2025).

[6] Grosnit, Antoine, et al. "High-dimensional Bayesian optimisation with variational autoencoders and deep metric learning." arXiv preprint arXiv:2106.03609 (2021).

[7] Lee, Seunghun, et al. "Latent bayesian optimization via autoregressive normalizing flows." arXiv preprint arXiv:2504.14889 (2025).

[8] Song, Xingyou, et al. "Omnipred: Language models as universal regressors." arXiv preprint arXiv:2402.14547 (2024).

[9] Kristiadi, Agustinus, et al. "A sober look at LLMs for material discovery: Are they actually good for Bayesian optimization over molecules?." arXiv preprint arXiv:2402.05015 (2024).

**Questions:**

See weakness section above

---

> ### Author Response · Authors · 2025-12-03
> **Reply to Reviewer BP3g - Part 1/3**
>
> Thank you for the review and feedback provided. We have carefully revised our manuscript regarding your concerns, incorporating additional experiments, results, and analyses. Please find our point-by-point reply to your questions as follows. We hope that these revisions and replies address your reservations.
>
> **Q1: The paper proposed their LLM regression model, but the experimental evaluation is insufficient due to a lack of comprehensive baselines[8~9]. It is difficult to assess the proposed method's true contribution without comparing it against a wider range of existing or even simpler alternative methods.**
>
> **A1:** We thank the reviewer for the valuable suggestion [8-9]. To enable a more comprehensive evaluation of our LLM regression module, we have added comparisons with several additional baselines (see Table R1), including simple machine learning models (Decision Tree, Random Forest, XGBoost, MLP), general-purpose language models (Bert), and domain-specific language models (Chem-T5, MolFormer). The additional results further confirm the superior effectiveness and versatility of our regression module in ChemBOMAS.
>
> **Table R1: Comparison with Additional Baselines.**
>
> | Model | Suzuki - MSE | MAE | $R^2$ | Arylation - MSE | MAE | $R^2$ | Buchwald - MSE |MAE | $R^2$ |
> | :--- | :---: | :---: | :---: | :---: | :---: | :---: | :---: | :---: | :---: |
> | Decision Tree | 749.86 | 20.8 | 0.05 | 927.01 | 223.38 | -0.24 | 1003.35 | 25.19 | -0.35 |
> | Random Forest | 693.08 | 22.25 | 0.12 | 735.65 | 22.78 | 0.01 | 761.55 | 23.5 | -0.02 |
> | XGBoost | 643.03 | 21.89 | 0.18 | 653.86 | 21.25 | **0.13** | 667.22 | 21.63 | 0.1 |
> | MLP | 737.88 | 23.93 | 0.04 | 596.3 | 22.62 | 0.03 | 612.57 | 22.51 | 0.06 |
> | Bert | 808.12 | 24.04 | -0.03 | 746.78 | 23.18 | 0 | 747.05 | 23.19 | 0 |
> | Chem-T5 | 1551.12 | 29.79 | -0.96 | 1189.38 | 26.14 | -0.6 | 1184.85 | 26.09 | -0.59 |
> | MolFormer | 788.57 | 24.04 | 0 | 746.85 | 23.17 | 0 | 744.97 | 23.19 | 0 |
> | ChemBOMAS | **633.68** |**19.47** | **0.2** | **650** | **19.55** | **0.13** | **593.76** | **18.52** | **0.2** |
>
> **Q2: The results in Figure 2 are concerning and require clarification. It is unclear why the objective values differ at iteration 0 for the various methods being compared. For a fair comparison, all methods should presumably start from an equivalent initial state.**
>
> **A2:** We thank the reviewer for this observation, which allows us to clarify two points about the initialization procedure.
> 1. All methods indeed start from a strictly equivalent initial state, having access to the same prior dataset (the 1% labeled data used for LLM fine-tuning) for a fair comparison. The key point is that "Iteration 0" in original Figure 2 does not represent this shared starting point, but rather the first search step after each algorithm has processed the identical initial data. We displayed the shared prior data to demonstrate fairness in the revised Figure 2.
> 2. The difference in objective values at "Iteration 0" arises because the methods process the shared prior differently. While most baselines fit their surrogate models directly to the 1% data, our method uses it to generate high-quality pseudo-labels and perform knowledge-driven subspace partitioning. Therefore, by "Iteration 0", our approach has already refined its surrogate model and shrunk the search space, leading to a more informed and advantageous first search step. This initial performance gap effectively demonstrates the benefit of our method's integration of knowledge and data modules.

---

> ### Author Response · Authors · 2025-12-03
> **Reply to Reviewer BP3g - Part 2/3**
>
> **Q3: Following Figure 2, the optimization curves for the proposed method appear almost entirely flat after the initial iteration. This suggests that minimal, if any, optimization or "search" is occurring. This raises a critical question: Is the method's performance simply an artifact of finding a good initial data point, rather than a demonstration of an effective optimization process?**
>
> **A3:** To verify that our method's performance stems from an effective optimization process rather than merely a favorable initialization, we have included Table R2 below. This table reports, for each benchmark dataset and across ten independent runs: (i) the maximum value at iteration 0 (initial), (ii) the best value discovered during the entire Bayesian Optimization run (best found), and (iii) the iteration at which this best value was first identified (best iter).
> The results in Table R2 demonstrate three key facts:
>
> 1. The best-found value consistently surpasses the initial maximum and matches the global maximum of every dataset.
> 2. The iteration at which the best value is first achieved is always strictly greater than 0. This confirms that the optimum was discovered in subsequent optimization steps, not by the initial design alone.
> 3. The high-yield points are extremely rare in the search space. ChemBOMAS manages to reach the global or a quasi-optimum within just a few iterations, after which the curve flattens simply because the proposed algorithm has already converged. This rapid convergence, compared to other baselines, underscores its superior sample efficiency.
>
> **Table R2: Benchmark Dataset Details and ChemBOMAS Metrics over Ten Runs.**
>
> | Benchmarks | Global Maximum | Number and Proportion of ≥ 95% Maximum | Initial. (Mean $\pm$ Std.) | Best Found (Mean $\pm$ Std.) | 95% Iter. Round (Mean $\pm$ Std.) | Best Iter. Round (Mean $\pm$ Std.) |
> | :--- | :---: | :---: | :---: | :---: | :---: | :---: |
> | **Suzuki** | 96.15 | 2 / 0.04% | 92.24 $\pm$ 0.00 | 96.15 $\pm$ 0.00 | 1 $\pm$ 0 | 3 $\pm$ 0 |
> | **Arylation** | 84.65 | 3 / 0.08% | 82.63 $\pm$ 0.00 | 82.83 $\pm$ 0.64 | 1 $\pm$ 0 | 4 $\pm$ 10 |
> | **Buchwald_sub1** | 80.91 | 31 / 4.93% | 75.55 $\pm$ 0.00 | 79.97 $\pm$ 0.50 | 4 $\pm$ 5 | 23 $\pm$ 13 |
> | **Buchwald_sub2** | 56.81 | 5 / 0.65% | 53.33 $\pm$ 0.00 | 56.81 $\pm$ 0.00 | 2 $\pm$ 0 | 2 $\pm$ 0 |

---

> ### Author Response · Authors · 2025-12-03
> **Reply to Reviewer BP3g - Part 3/3**
>
> **Q4: A notable omission is the comparison against LBO (Latent Bayesian Optimization)[1~7]. Given the nature of the problem, LBO methods seem like a highly relevant and potentially strong baseline. The authors should justify this exclusion or provide results for this comparison.**
>
> **A4:** We thank the reviewer for suggesting the comparison with Latent Bayesian Optimization (LBO) methods. We have actively attempted to include relevant LBO baselines and provide the following clarifications:
>
> 1.  **LLM-based LBO Baseline:** We have included GOLLuM [10] as a representative LBO method that performs optimization in a latent space constructed via an LLM. As shown in Table R3, ChemBOMAS statistically matches or outperforms GOLLuM across all benchmarks.
> 2.  **Challenges with VAE-based LBO:** We found VAE-based LBO methods unsuitable for our problem for two key reasons:
>     *   Domain Mismatch: Off-the-shelf chemical VAEs are pre-trained on molecular structures, not for representing mixed-type reaction conditions (categorical and continuous variables). Adapting them would require substantial domain-specific pre-training data, which was not feasible for our low-label tasks within the rebuttal period.
>     *   Representation Collapse: When testing the VAE from the COWBOY framework [7], we found that distinct reagents were often mapped to identical latent vectors due to vocabulary limitations. This collapse of chemically meaningful distinctions renders the latent space unreliable for chemical reaction optimization tasks.
> 3.  Technical Infeasibility: We also attempted to run other VAE-based LBO frameworks [1-6],  but these relied on deprecated software libraries and older CUDA/PyTorch stacks that are incompatible with our current environment (CUDA 12.8). Porting these frameworks was not practicable within the revision timeline.
>
> **Table R3: Comparison Between ChemBOMAS and GOLLuM Across Benchmarks (Values formatted as: ChemBOMAS / GOLLuM).**
>
> | Dataset | Initial. (Mean ± Std.) | Paired t-test | Best Found (Mean ± Std.) | Paired t-test | 95% Iter. Round (Mean ± Std.) | Paired t-test | Best Iter. Round (Mean ± Std.) | Paired t-test |
> | :--- | :-----: | :---: | :---: | :---: | :-------: | :---: | :-------: | :---: |
> | **Suzuki** | **92.24 ± 0** / 26.53 ± 0.36 | p < 0.05 | **96.15 ± 0** / 78.07 ± 6.67 | p < 0.05 | **1 ± 0** / 25 ± 9 | p < 0.05 | **3 ± 0** / 31 ± 7 | p < 0.05 |
> | **Arylation** | **82.63 ± 0** / 23.77 ± 1.17 | p < 0.05 | 82.83 ± 0.64 / 76.99 ± 8.39 | p = 0.056 | **1 ± 0** / 21 ± 11 | p < 0.05 | **4 ± 10** / 29 ± 11 | p < 0.05 |
> | **Buchwald_sub1** | **75.55 ± 0** / 36.32 ± 4.02 | p < 0.05 | 79.97 ± 0.50 / 79.77 ± 0.72 | p = 0.398 | **4 ± 5** / 13 ± 7 | p < 0.05 | 23 ± 13 / 25 ± 12 | p = 0.738 |
> | **Buchwald_sub2** | **53.33 ± 0** / 7.46 ± 3.08 | p < 0.05 | **56.81 ± 0** / 54.99 ± 1.99 | p < 0.05 | **2 ± 0** / 24 ± 11 | p < 0.05 | **2 ± 0** / 28 ± 12 | p < 0.05 |
>
> **Reference**
>
> [1] Tripp, Austin, Erik Daxberger, and José Miguel Hernández-Lobato. "Sample-efficient optimization in the latent space of deep generative models via weighted retraining." Advances in Neural Information Processing Systems 33 (2020): 11259-11272.
>
> [2] Maus, Natalie, et al. "Local latent space bayesian optimization over structured inputs." Advances in neural information processing systems 35 (2022): 34505-34518.
>
> [3] Lee, Seunghun, et al. "Advancing bayesian optimization via learning correlated latent space." Advances in Neural Information Processing Systems 36 (2023): 48906-48917.
>
> [4] Chu, Jaewon, et al. "Inversion-based latent bayesian optimization." Advances in Neural Information Processing Systems 37 (2024): 68258-68286.
>
> [5] Moss, Henry B., Sebastian W. Ober, and Tom Diethe. "Return of the latent space COWBOYS: Re-thinking the use of VAEs for Bayesian optimisation of structured spaces." arXiv preprint arXiv:2507.03910 (2025).
>
> [6] Grosnit, Antoine, et al. "High-dimensional Bayesian optimisation with variational autoencoders and deep metric learning." arXiv preprint arXiv:2106.03609 (2021).
>
> [7] Lee, Seunghun, et al. "Latent bayesian optimization via autoregressive normalizing flows." arXiv preprint arXiv:2504.14889 (2025).
>
> [8] Song, Xingyou, et al. "Omnipred: Language models as universal regressors." arXiv preprint arXiv:2402.14547 (2024).
>
> [9] Kristiadi, Agustinus, et al. "A sober look at LLMs for material discovery: Are they actually good for Bayesian optimization over molecules?." arXiv preprint arXiv:2402.05015 (2024).
>
> [10] Ranković B, Schwaller P. "GOLLuM: Gaussian Process Optimized LLMs--Reframing LLM Finetuning through Bayesian Optimization." arXiv preprint arXiv:2504.06265 (2025).
>
> **We hope these additional experiments and clarifications would substantially improve the assessment of soundness and presentation of this work.**

---

### Author Response · Authors · 2025-12-03
**Cover Letter to Area Chairs and All Reviewers**

We thank the Area Chairs and Reviewers for their time and constructive feedback. To assist you in efficiently evaluating this submission, we provide below a concise overview of the paper’s **key contributions**, the **reviewers’ main feedback**, and the **most significant revisions** made in response to their suggestions.

**Key Contributions**
1. **Challenge & Motivation**: Bayesian Optimization (BO) in chemistry struggles with exploring vast combinatorial spaces under extreme data scarcity. We systematically investigate how Large Language Model (LLM)-based approaches could address these two bottlenecks.
2. **Framework**: We propose ChemBOMAS, a novel framework that synergistically combines an LLM-based knowledge module (for chemically meaningful space partitioning) and a data module (for pseudo-data generation and warm-starting), which enhances BO performance when exploring vast combinatorial spaces with extreme data scarcity.
3. **Performance**: Across four chemical reaction optimization benchmarks with only 1% labeled data, ChemBOMAS consistently and significantly outperforms strong baselines with 10 independent seeds.
4. **Real-World Validation**: We demonstrate practical utility and scalability through a wet-lab campaign in a ~10⁵-combination variable space.

**Reviewer's Feedback**

Reviewers recognized the **novelty, significance**, and **practical value** of the work, noting:
1. *“...a novel multi-agent system-based Bayesian optimization method.”* [Reviewer BP3g]
2. *"This paper presents a well-designed and innovative pipeline, ...., the problem addressed here is exciting and topical."* [Reviewer brq8]
3. *“The framework was validated through wet-lab experiments.”* [Reviewer GLbf]
4. *“Directly addresses cold-start and high-dimensionality pain points… demonstrating operational relevance beyond simulation.”* [Reviewer NBfQ]

The manuscript was also praised for its clarity, structure, reproducibility, and thorough supplementary materials by the reviewers.

**Major Revisions in Response**

We sincerely appreciate the reviewers’ constructive suggestions. We have carefully revised our manuscript and provided a detailed point-by-point reply to all comments in our author response. Below, we concisely **summarize the most substantial revisions, including newly added experiments, results, and analyses** that directly strengthen the paper’s robustness and clarity:

1. **Expanded Baseline Comparisons**
Added comprehensive baselines, including traditional ML models (Decision Tree, Random Forest, XGBoost, MLP), general‑purpose LMs (BERT), domain‑specific LMs (Chem‑T5, MolFormer) for pseudodata generation performance, and a latent‑space BO method (GOLLum) for BO performance. ChemBOMAS consistently matches or outperforms all of them (**Tables 1 & 7**).
2. **Validated Robustness of the Knowledge‑Driven Module**
Detailed the retrieval‑augmented generation (RAG) implementation and conducted sensitivity experiments (5 reruns per benchmark), confirming that the knowledge module delivers reliable and consistent outcomes (**Appendix J.1**).
3. **Validation Under Extreme Data Scarcity**
Tested pseudodata quality with as little as 0.25–0.5% labeled data (~20 points per benchmark) and subsequent ChemBOMAS performance. Results show robust optimization performance even under severe data limitations, alleviating concerns about the need for extensive dataset‑specific tuning (**Appendix J.2**).
4. **Strengthened Statistical Reporting**
All key experiments (baselines, ablations, RAG sensitivity, pseudo‑label bias, and acquisition function selection) were rerun with 10 independent seeds. Reported means, standard deviations, confidence intervals, and significance tests confirm that ChemBOMAS’s improvements in best‑found value, convergence speed, and initialization are statistically significant in nearly all cases (**Tables 2, 4, 7, 9-11**).
5. **Clarified Fair and Transparent Evaluation**
We supplemented the details on unified experiment settings for all baselines and the proposed method to ensure strict fairness. Additionally, we added the global maximum line and initialized prior data for each benchmark in Figure 2 to clarify and highlight ChemBOMAS's superior optimization performance in a fair comparison with baselines.  (**Table 6  and Figure 2**)
6. **Extended Discussion of Generality and Scalability**
Further elaborated on the material‑type task and the wet‑lab campaign (~10⁵ combinations) to underscore the framework’s broad applicability and scalability (**Appendices I & K**).

We believe these revisions and subsequent replies fully address the reviewers’ concerns and hope the paper now aligns with the standards of excellence set by your esteemed conference. Thank you for your time and consideration.

---

### Meta-Review · Area_Chair_j7GY · 2025-12-27

**Summary:**

The paper describes a new LLM-based system for scientific discovery by accelerated Bayesian optimization.  The reviewers expressed the following concerns:

1. Lack of LLM regression baselines
2. Lack of clarity regarding optimization results
3. Lack of optimization baselines
4. The design space of BO, in terms of acquisition functions and policies, is explored insufficiently.
5. The performance gap between the proposed method and LLMs trained on labeled data is not too huge
6. Lack of statistical reporting
7. The LLM regressor can introduce bias if mis-calibrated.
8. Regarding RAG partitioning auditability: the retrieval corpus, filtering criteria, and sensitivity of subspace splits to retrieval noise are under-documented.
9. Issues regarding presentation and clarity
10. Lack of experimental rigour
11. The model's performance relies heavily on fine-tuning data volume, suggesting that substantial effort is needed to optimize the model for specific datasets.
12. The integration of knowledge-driven and data-driven modules seem complex and convoluted

**Reviewer Concerns:**

The rebuttal and revised paper addressed most concerns.  The authors did a lot of work that significantly improved the paper.

Reviewer BP3g expressed concerns about a lack of baselines for regression and optimization.  The authors included several new baselines, but did not cite, discuss nor compare the two baselines referenced by the reviewer for optimization:

[8] Song, Xingyou, et al. "Omnipred: Language models as universal regressors." arXiv preprint arXiv:2402.14547 (2024).

[9] Kristiadi, Agustinus, et al. "A sober look at LLMs for material discovery: Are they actually good for Bayesian optimization over molecules?." arXiv preprint arXiv:2402.05015 (2024).

Similarly, the authors added some baselines for regression, but did not cite, discuss nor compare most of the baselines referenced by the reviewer in the revised paper:

[1] Tripp, Austin, Erik Daxberger, and José Miguel Hernández-Lobato. "Sample-efficient optimization in the latent space of deep generative models via weighted retraining." Advances in Neural Information Processing Systems 33 (2020): 11259-11272.

[2] Maus, Natalie, et al. "Local latent space bayesian optimization over structured inputs." Advances in neural information processing systems 35 (2022): 34505-34518.

[3] Lee, Seunghun, et al. "Advancing bayesian optimization via learning correlated latent space." Advances in Neural Information Processing Systems 36 (2023): 48906-48917.

[4] Chu, Jaewon, et al. "Inversion-based latent bayesian optimization." Advances in Neural Information Processing Systems 37 (2024): 68258-68286.

[5] Moss, Henry B., Sebastian W. Ober, and Tom Diethe. "Return of the latent space COWBOYS: Re-thinking the use of VAEs for Bayesian optimisation of structured spaces." arXiv preprint arXiv:2507.03910 (2025).

[6] Grosnit, Antoine, et al. "High-dimensional Bayesian optimisation with variational autoencoders and deep metric learning." arXiv preprint arXiv:2106.03609 (2021).

[7] Lee, Seunghun, et al. "Latent bayesian optimization via autoregressive normalizing flows." arXiv preprint arXiv:2504.14889 (2025).

The authors clarified the optimization results by revising Figure 2 and adding some explanations.

The concerns expressed by reviewer brq8 were well-addressed by adding some experiments and new explanations.

Most of the concerns expressed by reviewer NBfQ were addressed by the rebuttal and the revised paper.  While the authors added the t-test to determine statistical significance, this statistical test assumes that the data is Gaussian, which is rarely the case in practice and cannot be verified most of the time.  The authors should redo the statistical analysis with the Wilcoxon signed rank test since it does not make any distributional assumption.  The authors provided additional explanations and experiments regarding the RAG techniques, but the description remains too high level.

Most of the concerns expressed by reviewer GLbf were addressed by the rebuttal and revised paper.  However, the RAG technique remains too high level and the overall explanation of the system lacks details.

**Reviewer Scores:**

I expect the reviewers to stick with their scores because the rebuttal and the revised paper partly addressed the concerns of the reviewers.  The authors added many new experiments and improved the paper significantly.  However, most of the baselines pointed by the reviewers were not referenced, discussed not compared to.  The description of the RAG approach remains too high level.  The statistical test used assumes that the data is Gaussian, which is unlikely to be the case.  Overall, this is promising engineering work that combines several existing approaches that yield improvements for scientific discovery.  While the work deserves to be published eventually, it will need another round of revisions and reviews.

---

### Decision · Program_Chairs · 2026-01-26

Reject